# Unifying Perspectives: Plausible Counterfactual Explanations on Global, Group-wise, and Local Levels

## Abstract

The growing complexity of AI systems has intensified the need for transparency through Explainable AI (XAI). Counterfactual explanations (CFs) offer actionable "what-if" scenarios on three levels: Local CFs providing instance-specific insights, Global CFs addressing broader trends, and Group-wise CFs (GWCFs) striking a balance and revealing patterns within cohesive groups. Despite the availability of methods for each granularity level, the field lacks a unified method that integrates these complementary approaches. We address this limitation by proposing a gradient-based optimization method for differentiable models that generates Local, Global, and Group-wise Counterfactual Explanations in a unified manner. We especially enhance GWCF generation by combining instance grouping and counterfactual generation into a single efficient process, replacing traditional two-step methods. Moreover, to ensure trustworthiness, we innovatively introduce the integration of plausibility criteria into the GWCF domain, making explanations both valid and realistic. Our results demonstrate the method's effectiveness in balancing validity, proximity, and plausibility while optimizing group granularity, with practical utility validated through practical use cases.

## 1 Introduction

The increasing complexity of AI systems has fueled regulatory and societal demands for transparency, a need addressed by Explainable AI (XAI) (Goodman & Flaxman, 2017; Wachter et al., 2017; Adadi & Berrada, 2018; Samek & Müller, 2019). Among XAI techniques, counterfactual explanations (CFs) are particularly valuable for providing actionable "what-if" scenarios that specify how input feature changes can alter model predictions (Wachter et al., 2017). For example, a CF could show a loan applicant the precise changes needed for loan approval, offering actionable feedback crucial in many fields (Guidotti, 2022).

Counterfactual explanations can be generated at three distinct levels of granularity. The most popular **Local** CFs offer tailored guidance for individual instances but miss broader patterns (Fragkathoulas et al., 2024; Carrizosa et al., 2024). **Global** CFs provide high-level summaries for entire datasets but lack individual specificity (Ramamurthy et al., 2020; Plumb et al., 2020). Bridging this gap, **group-wise** counterfactual explanations (GWCFs) explain cohesive data subsets, revealing shared patterns while maintaining actionable insights, which is crucial for fairness and policy-making in sensitive domains (Carrizosa et al., 2024; Kanamori et al., 2022; Warren et al., 2024). A detailed comparison of these approaches is illustrated in Figure 1 and discussed in Appendix A.

Despite their promise, existing GWCF methods face significant challenges. Most rely on a two-step process of first clustering data and then generating CFs (or vice versa), which is inefficient and dependent on clustering parameterization (Kavouras et al., 2024; Artelt & Gregoriades, 2024). Furthermore, ensuring the *plausibility* of CFs—that is, their alignment with the data distribution and real-world constraints—remains a key challenge, as unrealistic explanations undermine trust and actionability (Artelt & Hammer, 2020).

To address these challenges, we propose a unified framework for generating local, group-wise, and global counterfactuals, as illustrated in Figure 1. Our end-to-end, gradient-based method simultaneously optimizes instance grouping and counterfactual generation, eliminating the inefficient two-step

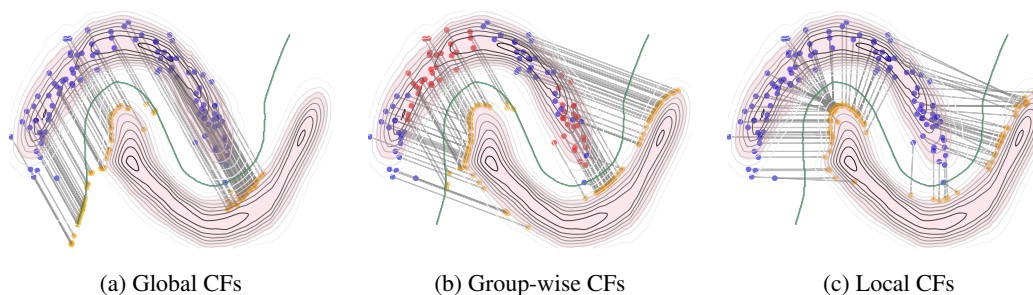

(a) Global CFs        (b) Group-wise CFs        (c) Local CFs

Figure 1: The figure illustrates three types of explanations generated by our approach: (a) *global CFs*, identifying a single direction of change applicable to the entire dataset; (b) *group-wise CFs*, providing vectors of change for specific groups, distinguished by colors (red, blue); and (c) *local counterfactual explanations*, offering instance-specific shift vectors, minimal changes required to modify individual predictions. Decision boundary (green line) and density threshold contours.

process common in prior work. It can dynamically generate explanations for a varying number of groups, automatically balancing a number through regularization. By formulating this as a single optimization problem, our method efficiently produces CFs at any desired granularity. Crucially, we introduce a probabilistic plausibility criterion, using normalizing flows for density estimation (Rezende & Mohamed, 2015), to ensure that explanations are not only valid but also realistic and actionable.

In summary, our key contributions are:

- A novel unified approach for generating CFs at local, group-wise, and global levels, dynamically adapting to user needs and automatically balancing groups diversity and granularity, leveraging gradient-based optimization.
- A significant advancement in GWCFs generation through end-to-end optimization that unifies group discovery and counterfactual generation while introducing probabilistic plausibility constraints in this domain.
- An experimental evaluation and real-world use case analysis demonstrating our approach's performance, providing the effective balance between validity, proximity, plausibility, and the number of shifting vectors.

## 2 RELATED WORKS

**Local Counterfactual Explanations** Local CFs identify minimal feature changes to alter a model's prediction for a single instance (Wachter et al., 2017). While early methods were often heuristic-based, subsequent work has introduced more sophisticated techniques, including gradient-based optimization, generative models, and contrastive explanations, to improve CF quality and diversity (Dhurandhar et al., 2018; Russell, 2019; Kanamori et al., 2020; Mothilal et al., 2020; Guidotti, 2022). However, ensuring the plausibility and actionability of these explanations remains an ongoing challenge (Keane et al., 2021).

**Global and Group-wise Counterfactual Explanations** Global and group-wise CFs extend explanations beyond single instances to entire datasets or cohesive subgroups. Global approaches seek a single or a few explanations for all instances, using techniques like feature space translations (Plumb et al., 2020), actionable rule sets (Rawal & Lakkaraju, 2020; Ley et al., 2022), or scalable vector-based methods (Ley et al., 2023). Group-wise methods provide more granular insights. Some approaches partition the input space using tree structures to assign collective actions (Ramamurthy et al., 2020; Kanamori et al., 2022; Bewley et al., 2024). Others follow a two-step process, first generating local CFs and then clustering them to find group-level explanations (Kavouras et al., 2024; Artelt & Gregoriades, 2024). These two-step methods, however, can be inefficient and sensitive to clustering parameters.

**Plausible Counterfactual Explanations** Plausibility ensures that a CF resides in a high-density region of the data manifold, making it realistic and trustworthy. Various techniques have been proposed to enforce this, such as imposing density constraints using Gaussian Mixture Models (Artelt & Hammer, 2020) or normalizing flows (Wielopolski et al., 2024). Other approaches leverage causal constraints (Mahajan et al., 2019) or generative models like VAEs to learn the data manifold and generate plausible CFs from it (Pawelczyk et al., 2020). A comprehensive survey by Karimi et al. (2022) details the challenges and opportunities in this area.

## 3 BACKGROUND

**Counterfactual Explanations** Following Wachter et al. (2017), a local counterfactual explanation finds a new instance $\mathbf{x}' \in \mathbb{R}^D$ for an original instance $\mathbf{x}_0 \in \mathbb{R}^D$ such that the prediction of a model $h$ changes to a desired class $y'$, i.e., $h(\mathbf{x}') = y'$. The instance $\mathbf{x}'$ is typically found by solving the optimization problem:

$$\arg \min_{\mathbf{x}' \in \mathbb{R}^D} d(\mathbf{x}_0, \mathbf{x}') + \lambda \cdot \ell(h(\mathbf{x}'), y'). \tag{1}$$

The function $\ell(\cdot, \cdot)$ refers to a loss function tailored for classification tasks such as the 0-1 loss or cross-entropy. On the other hand, $d(\cdot, \cdot)$ quantifies the distance between the original input $\mathbf{x}_0$ and its counterfactual counterpart $\mathbf{x}'$, employing metrics like the L1 (Manhattan) or L2 (Euclidean) distances to evaluate the deviation. The parameter $\lambda \geq 0$ plays a pivotal role in regulating the trade-off, ensuring that the counterfactual explanation remains sufficiently close to the original instance while altering the prediction outcome as intended.

**Plausible Counterfactual Explanations** To ensure realism, Artelt & Hammer (2020) introduced a plausibility constraint, requiring the counterfactual $\mathbf{x}'$ to lie in a high-density region of the data distribution $p(\mathbf{x}|y')$ for the target class. The optimization problem becomes:

$$\arg \min_{\mathbf{x}' \in \mathbb{R}^D} d(\mathbf{x}_0, \mathbf{x}') + \lambda \cdot \ell(h(\mathbf{x}'), y') \tag{2a}$$

$$\text{s.t.} \quad \delta \leq p(\mathbf{x}'|y'), \tag{2b}$$

where $p(\mathbf{x}'|y')$ denotes conditional probability of the counterfactual explanation $\mathbf{x}'$ under desired target class value $y'$ and $\delta$ represents the density threshold.

**Global and Group-wise Counterfactual Explanations** Global and group-wise explanations extend the local concept by applying a shared change vector $\mathbf{d}$ to a set of instances. For a global explanation, a single vector $\mathbf{d}$ is applied to all instances. For group-wise explanations, different vectors are found for different subgroups of the data. The counterfactual for an instance $\mathbf{x}_0$ is then generated by a simple update:

$$\mathbf{x}' = \mathbf{x}_0 + \mathbf{d}, \tag{3}$$

where $\mathbf{d}$ is the shift vector of size $D$, which remains invariant across all observations within the same class or group.

In contrast to the standard formulation, GLOBE-CE (Ley et al., 2023) introduces a scaling factor, $k$, specific to each observation, allowing for individual adjustments to the magnitude of the shift:

$$\mathbf{x}' = \mathbf{x}_0 + k \cdot \mathbf{d}. \tag{4}$$

## 4 METHOD

### 4.1 GLOBAL COUNTERFACTUAL EXPLANATIONS

The base problem of global counterfactual explanation assumes finding the global shifting vector $\mathbf{d}$ of size $D$. In order to solve that problem using optimization techniques, we can define the problem in the following way:

$$\arg \min_{\mathbf{d}} d^G(\mathbf{X}_0, \mathbf{X}') + \lambda \cdot \ell^G(h(\mathbf{X}'), y'), \tag{5}$$

where $\mathbf{X}_0 = [\mathbf{x}_{1,0}, \ldots, \mathbf{x}_{1,N}]^{\mathrm{T}}$ represents the matrix storing the initial input $N$ examples, $\mathbf{X}' = [\mathbf{x}_{1,0} + \mathbf{d}, \ldots, \mathbf{x}_{1,N} + \mathbf{d}]^{\mathrm{T}}$ represent the extracted conterfactuals, after shifting the input examples

with vector $\mathbf{d}$. Formally, $\mathbf{X}' = \mathbf{X}_0 + \mathbf{D}$, where $\mathbf{D} = \mathbf{1}_N \cdot \mathbf{d}^{\mathrm{T}}$ and $\mathbf{1}_N$ represents $N$-dimesional vector containing ones. We define a global distance as $d^G(\mathbf{X}_0, \mathbf{X}') = \sum_{n=1}^{N} d(\mathbf{x}_{n,0}, \mathbf{x}'_n)$, and global classification loss as an aggregation of the components: $\ell^G(h(\mathbf{X}'), y') = \sum_{n=1}^{N} \ell(h(\mathbf{x}'_n), y')$.

Extracting a single direction vector $\mathbf{d}$ can be inefficient due to the dispersed initial positions $\mathbf{X}_0$ and, as discussed by Kanamori et al. (2022), it strictly depends on the farthest observation. Therefore, following the GLOBE-CE (Ley et al., 2023), we incorporate additional magnitude components and represent the counterfactuals as:

$$\mathbf{X}'_K = \mathbf{X}_0 + \mathbf{KD}, \tag{6}$$

where $\mathbf{K}$ is the diagonal matrix of magnitudes on the diagonal, i.e., $\mathbf{K} = \mathrm{diag}(k_1, \dots, k_N)$. In order to ensure non-negative values of magnitudes, we represent them as $k_i = \exp(h_i)$. This formulation extends the classical vector-based update rule given by eq. equation 4 to the matrix notation. In order to extract the counterfactuals, we simply include $\mathbf{X}_0$ in eq. equation 5 and optimize $\mathbf{K}$ together with $\mathbf{d}$.

## 4.2 GROUP-WISE COUNTERFACTUAL EXPLANATIONS

Incorporating magnitude components into the global counterfactual problem enhances the shifting options during counterfactual calculation, yet the direction remains uniform across all observations. To address this, we propose a novel method that automatically identifies groups represented by local shifting vectors with varying magnitudes. This approach restricts the number of desired shifting components to these identified groups. The formula for extracting group-wise counterfactuals is defined as:

$$\mathbf{X}'_{GW} = \mathbf{X}_0 + \mathbf{KSD}_{GW}, \tag{7}$$

where $\mathbf{D}_{GW}$ is a matrix of size $K \times D$, $K$ is the number of base shifting vectors and $D$ is the dimesionality of the data. $\mathbf{S}$ is a sparse selection matrix of size $N \times K$, where $s_{n,k} \in \{0, 1\}$ and $\sum_{k=1}^{K} s_{n,k} = 1$ for each of the considered rows. Practically, the operation selects one of the base shifting vectors $\mathbf{d}_k$, where $\mathbf{D} = [\mathbf{d}_1, \dots, \mathbf{d}_K]^{\mathrm{T}}$, scaled by components $k_n$ located on diagonal of matrix $\mathbf{K}$. We aim to optimize the selection matrix $\mathbf{S}$ together with base vectors $\mathbf{D}_{GW}$ and magnitude components $\mathbf{K}$ using the gradient-based approach. Optimizing binary $\mathbf{S}$ directly is challenging due to the type of data and the given constraints. Therefore, we replace the $\mathbf{S}$ with the probability matrix $\mathbf{P}$, where the rows $\mathbf{p}_{n,\bullet}$ represent the values of Sparsemax (Martins & Astudillo, 2016) activation function:

$$\mathbf{p}_{n,\bullet} = \arg\min_{\mathbf{p} \in \Delta} ||\mathbf{p} - \mathbf{b}_{n,\bullet}||^2, \tag{8}$$

where $\Delta = \{\mathbf{p} \in \mathbb{R}^K : \mathbf{1}_K^{\mathrm{T}} \mathbf{p} = 1, \mathbf{p} \geq \mathbf{0}_K\}$ and $\mathbf{b}_{n,\bullet}$ is $n$-th row of $\mathbf{B}$, which is the real-valued auxiliary matrix that is used to model rows of $\mathbf{S}$ as one-hot binary vectors. Practically, each row of the matrix $\mathbf{P}$ represents a multinomial distribution, and matrix $\mathbf{B}$ is optimized in the gradient-based framework.

The objective for extracting group-wise counterfactuals is as follows:

$$\arg\min_{\mathbf{K}, \mathbf{B}, \mathbf{D}_{GW}} \quad d^G(\mathbf{X}_0, \mathbf{X}'_{GW}) + \lambda \cdot \ell^G(h(\mathbf{X}'_{GW}), y') + \\ + \lambda_s \cdot \ell_s(\mathbf{B}) + \lambda_k \cdot \ell_k(\mathbf{B}), \tag{9}$$

where $\ell_s(\mathbf{B})$ and $\ell_k(\mathbf{B})$ are entropy-based regularisers applied to preserve sparsity of matrix $\mathbf{P}$, and $\lambda_s$ and $\lambda_k$ are regularisation hyperparameters. The regularizer $\ell_s(\mathbf{B})$ is encouraging assignment to one group for each of the raw vectors $\mathbf{p}_{n,\bullet}$:

$$\ell_s(\mathbf{B}) = -\sum_{n=1}^{N} \sum_{k=1}^{K} p_{n,k} \cdot \log p_{n,k}. \tag{10}$$

The second regularisation component is responsible for reducing the number of groups extracted during counterfactual optimization:

$$\ell_k(\mathbf{B}) = -\sum_{k=1}^{K} p_k \cdot \log p_k, \tag{11}$$

where $p_k = \frac{\sum_{n=1}^{N} p_{k,n}}{\sum_{k=1}^{K} \sum_{n=1}^{N} p_{k,n}}$.

The problem formulation provided by eq. equation 7 and equation 9 represents the unified framework for counterfactual explanations. If the number of base shifting vectors in matrix $\mathbf{D}_{GW}$ is equal to the number of examples ($K = N$), $\mathbf{S} = \mathbf{K} = \mathbb{I}$, and $\lambda_k = \lambda_s = 0$, the problem statements refer to standard formulation of local explanations. In the case where $\mathbf{D}_{GW} = \mathbf{D}$, $\mathbf{S} = \mathbf{K} = \mathbb{I}$, and $\lambda_s = \lambda_k = 0$, the statement pertains to standard global counterfactual explanations. When $\mathbf{K} \neq \mathbb{I}$, it is equivalent to the formulation given in Eq. 6, i.e., GCFs with magnitude. In other cases ($1 < K < N$), the problem is formulated as a group-wise explanation case. In this setting, we can disable automatic group detection ($\lambda_k = 0$) and instead prioritize manual control over the automatic number of group formations ($\lambda_k > 0$). This latter configuration will be our primary focus for GWCFs.

### 4.3 PLAUSIBLE COUNTERFACTUAL EXPLANATIONS AT ALL LEVELS

The plausibility is an important aspect of generating relevant counterfactuals. In this paper, we focus on density-based problem formulation, where the extracted example should satisfy the condition of preserving the density function value on a given threshold level (see eq. equation 2b): $\delta \leq p(\mathbf{x}'|y')$. Moreover, we utilize a specific form of classification loss that enables a balance between the plausibility and validity of the extracted examples.

The general criterion for extracting plausible group-wise counterfactuals can be formulated as follows:

$$\arg \min_{\mathbf{K},\mathbf{B},\mathbf{D}_{GW}} \quad d^G(\mathbf{X}_0, \mathbf{X}'_{GW}) + \lambda \cdot \ell^G(h(\mathbf{X}'_{GW}), y') +$$
$$+ \lambda_p \cdot \ell_p(\mathbf{X}'_{GW}, y') + \lambda_s \cdot \ell_s(\mathbf{B}) + \lambda_k \cdot \ell_k(\mathbf{B}), \tag{12}$$

where the loss component $\ell_p(\mathbf{X}'_{GW}, y')$ controls probabilistic plausibility constraint ($\delta \leq p(\mathbf{x}'|y')$) and is defined as:

$$\ell_p(\mathbf{X}'_{GW}, y') = \sum_{n=1}^{N} \max\Big(\delta - p(\mathbf{x}'_{GW,n}|y'), 0\Big), \tag{13}$$

where $\mathbf{x}'_{GW,n}$ is $n$-th counterfactual example stored in rows of $\mathbf{X}'_{GW} = [\mathbf{x}'_{GW,1}, \dots, \mathbf{x}'_{GW,N}]^{\mathrm{T}}$ and $\delta$ is the density threshold defined by the user depending on the desired level of plausibility.

Various approaches, like Kernel Density Estimation (KDE) or Gaussian Mixture Model (GMM) can be used to model conditional density function $p(\mathbf{x}'_{GW,n}|y')$. In this work, we follow Wielopolski et al. (2024) and use a conditional normalizing flow model (Rezende & Mohamed, 2015) to estimate the density. Compared to standard methods, like KDE or GMM, normalizing flows do not assume a particular parametrized form of density function and can be successively applied for high-dimensional data. Compared to other generative models, normalizing flows enables the calculation of density function directly using the change of variable formula and can be trained via direct negative log-likelihood (NLL) optimization. A detailed description of how to model and train normalizing flows is provided in Appendix B. Having the trained discriminative model $p_d(y'|\mathbf{x}'_{GW,n})$ and generative normalizing flow $p(\mathbf{x}'_{GW,n}|y')$ the set of conterfacuals $\mathbf{X}'_{GW}$ is estimated using a standard gradient-based approach.

### 4.4 VALIDITY LOSS COMPONENT

The application of the cross-entropy classification loss $\ell^G(h(\mathbf{X}'_{GW}), y')$ in eq. equation 12 constantly encourages 100% confidence of the discriminative model, which may have some negative impact on balancing other components in aggregated loss. In order to eliminate this limitation, we replace $\ell^G(h(\mathbf{X}'_{GW}), y')$ with validity loss based on Wielopolski et al. (2024):

$$\ell_v(h(\mathbf{X}'_{GW}), y') =$$
$$\sum_{n=1}^{N} \max\Big(\max_{y \neq y'} p_d(y|\mathbf{x}'_{GW,n}) + \epsilon - p_d(y'|\mathbf{x}'_{GW,n}), 0\Big). \tag{14}$$

This enforces that $p_d(y'|\mathbf{x}'_{GW,n})$ will be higher than the most probable class among the remaining classes by the $\epsilon$ margin. Using our criterion, the model can focus more on producing closer and more plausible counterfactuals.

### 4.5 GROUP DIVERSITY REGULARIZATION

During optimization, the algorithm may converge towards proposing similar groups, overly capturing fine details. To ensure diversity among the base shifting vectors in $\mathbf{D}_{GW}$, we introduce a determinant-based regularization term that encourages linear independence and broad representation. The penalty is defined as:

$$\ell_d(\mathbf{D}_{GW}) = -\log\det(\mathbf{D}_{GW}\mathbf{D}_{GW}^{\mathrm{T}} + \epsilon\mathbf{I}), \tag{15}$$

where $\epsilon$ is a small positive constant that ensures numerical stability by preventing the determinant from becoming zero.

The optimization objective from Eq. equation 12 is updated to include the diversity term:

$$\arg\min_{\mathbf{K},\mathbf{B},\mathbf{D}_{GW}} \quad d^G(\mathbf{X}_0, \mathbf{X}'_{GW}) + \lambda \cdot \ell_v(h(\mathbf{X}'_{GW}), y') +$$
$$+ \lambda_p \cdot \ell_p(\mathbf{X}'_{GW}, y') + \lambda_s \cdot \ell_s(\mathbf{B}) + \tag{16}$$
$$+ \lambda_k \cdot \ell_k(\mathbf{B}) + \lambda_d \cdot \ell_d(\mathbf{D}_{GW}),$$

This term maximizes the volume spanned by the group shifting vectors, promoting distinct and diverse groups of counterfactual explanations.

Based on this, we conducted an ablation study on each component and their combinations (see Appendix I) and selected hyperparameters based on our findings to ensure optimal performance. We initialize with $K = N$ base shifting vectors and assign the highest weight, $\lambda = 10^5$, to emphasize validity. For plausibility, group sparsity, and number-of-groups regularization, we use equal weights $\lambda_p = \lambda_s = \lambda_k = 10^4$, reflecting their comparable importance in balancing realistic counterfactuals with group number. Finally, to ensure diversity among the group shifting vectors while allowing other constraints to dominate, we set $\lambda_d = 10^1$. Furthermore, we used the first quartile of the probabilities of the observed train set as the probability threshold $\delta$.

## 5 EXPERIMENTS

In this section, we evaluate the performance of our method in global, group-wise, and local configurations using various datasets and metrics. The experiments benchmark our approach against state-of-the-art methods, highlighting its strengths and providing insights into its unified capabilities. To further illustrate the practical value of our method, we analyze the created groups in two use cases, demonstrating its ability to generate actionable and interpretable insights. The code for these experiments is publicly available on GitHub[1]. Detailed results and additional evaluations are provided in Appendix J.

### 5.1 COMPARATIVE EXPERIMENTS

**Datasets** We conducted experiments on six datasets that cover diverse domains and challenges and are frequently used as benchmarks in the counterfactual explanation literature. The datasets include: three for tabular data binary classification (*Law*, *HELOC*, *Moons*); two for tabular data multiclass classification (*Blobs*, *Wine*); and one image dataset with multiple classes (*Digits*). The sizes of these datasets range from 178 samples (*Wine*) to 10,459 samples (*HELOC*), while feature dimensionality spans from 2 features (*Moons*, *Blobs*) to 64 features (*Digits*), ensuring robustness across different scales and complexities. Detailed descriptions of these datasets are available in Appendix E.1.

**Classification Models** For classification models, we trained a 2-layer *Multilayer Perceptron* (MLP) to test non-linear deep neural network configurations. For completeness, we provide additional comparative results using the LR model in Appendix J.4. Detailed descriptions of both model architectures are provided in Appendix E.2.

**Metrics** We evaluated counterfactual explanations using three key metrics: *Validity*, which measures the success of CFs in altering the model's predictions; *Proximity*, calculated as the $L2$ distance between the original instance and the CFs; *Plausibility*, assessed through the Isolation Forest metric

---

[1]Will be added in camera-ready version.

Table 1: Comparative analysis: Global Methods.

| DATASET | METHOD | COVERAGE↑ | VALID.↑ | L2↓ | ISOFOREST↑ | TIME(S)↓ |
|---|---|---|---|---|---|---|
| BLOBS | GLOBE-CE | **1.00 ± 0.00** | 0.99 ± 0.01 | **0.25 ± 0.04** | −0.06 ± 0.03 | **0.66 ± 0.03** |
| | GLANCE | **1.00 ± 0.00** | **1.00 ± 0.00** | 0.42 ± 0.01 | 0.01 ± 0.00 | 43.30 ± 9.72 |
| | OUR$_{global}$ | **1.00 ± 0.00** | **1.00 ± 0.00** | 0.48 ± 0.01 | **0.03 ± 0.00** | 7.89 ± 0.86 |
| DIGITS | GLOBE-CE | **1.00 ± 0.00** | 0.00 ± 0.00 | - | - | **0.95 ± 0.08** |
| | GLANCE | **1.00 ± 0.00** | 0.30 ± 0.07 | **11.24 ± 0.70** | 0.09 ± 0.01 | 678.36 ± 29.07 |
| | OUR$_{global}$ | **1.00 ± 0.00** | **1.00 ± 0.00** | 17.08 ± 0.54 | **0.10 ± 0.00** | 31.48 ± 5.28 |
| HELOC | GLOBE-CE | **1.00 ± 0.00** | **1.00 ± 0.00** | 0.52 ± 0.03 | 0.03 ± 0.01 | **2.02 ± 0.18** |
| | GLANCE | **1.00 ± 0.00** | 0.97 ± 0.01 | 0.68 ± 0.07 | −0.01 ± 0.02 | 99.89 ± 44.14 |
| | OUR$_{global}$ | **1.00 ± 0.00** | **1.00 ± 0.00** | **0.36 ± 0.02** | **0.06 ± 0.00** | 32.47 ± 10.01 |
| LAW | GLOBE-CE | **1.00 ± 0.00** | **1.00 ± 0.00** | **0.22 ± 0.02** | **0.01 ± 0.01** | **0.81 ± 0.02** |
| | GLANCE | **1.00 ± 0.00** | 0.97 ± 0.01 | 0.45 ± 0.02 | −0.04 ± 0.01 | 90.81 ± 9.03 |
| | OUR$_{global}$ | **1.00 ± 0.00** | **1.00 ± 0.00** | 0.38 ± 0.01 | **0.01 ± 0.00** | 13.44 ± 3.11 |
| MOONS | GLOBE-CE | **1.00 ± 0.00** | **1.00 ± 0.00** | **0.30 ± 0.03** | −0.06 ± 0.01 | **0.65 ± 0.01** |
| | GLANCE | **1.00 ± 0.00** | 0.68 ± 0.05 | 0.39 ± 0.02 | −0.02 ± 0.01 | 77.97 ± 9.11 |
| | OUR$_{global}$ | **1.00 ± 0.00** | 0.91 ± 0.12 | 0.45 ± 0.04 | **−0.01 ± 0.01** | 9.55 ± 1.37 |
| WINE | GLOBE-CE | **1.00 ± 0.00** | **1.00 ± 0.00** | **0.73 ± 0.20** | 0.04 ± 0.02 | **0.39 ± 0.01** |
| | GLANCE | **1.00 ± 0.00** | 0.57 ± 0.17 | 0.46 ± 0.07 | **0.06 ± 0.01** | 5.82 ± 3.10 |
| | OUR$_{global}$ | **1.00 ± 0.00** | **1.00 ± 0.00** | **0.73 ± 0.07** | **0.06 ± 0.01** | 5.73 ± 0.89 |

Table 2: Comparative analysis: Group-Wise Methods.

| DATASET | METHOD | GROUPS | COVERAGE↑ | VALID.↑ | L2↓ | ISOFOREST↑ | TIME(S)↓ |
|---|---|---|---|---|---|---|---|
| BLOBS | EA | 3.60 ± 1.67 | **1.00 ± 0.00** | **1.00 ± 0.00** | 1.00 ± 0.00 | −0.16 ± 0.00 | 95.38 ± 40.81 |
| | GLANCE | 2.00 ± 0.00 | **1.00 ± 0.00** | 0.96 ± 0.03 | 0.56 ± 0.02 | −0.10 ± 0.01 | 49.07 ± 3.9 |
| | TCREx | 2.40 ± 0.55 | **1.00 ± 0.00** | **1.00 ± 0.00** | 0.00 ± 0.00 | 0.02 ± 0.00 | **0.00 ± 0.00** |
| | OUR$_{group}$ | 1.60 ± 0.49 | **1.00 ± 0.00** | **1.00 ± 0.00** | 0.46 ± 0.01 | **0.03 ± 0.00** | 14.55 ± 2.51 |
| DIGITS | EA | 4.00 ± 0.00 | 0.00 ± 0.00 | − | − | − | 972.35 ± 62.15 |
| | GLANCE | 4.00 ± 0.00 | **1.00 ± 0.00** | **1.00 ± 0.00** | 2.01 ± 0.18 | −0.08 ± 0.01 | 761.25 ± 75.97 |
| | TCREx | 91.00 ± 50.76 | **1.00 ± 0.00** | **1.00 ± 0.00** | **0.15 ± 0.06** | 0.09 ± 0.00 | **13.37 ± 5.26** |
| | OUR$_{group}$ | 2.80 ± 1.83 | **1.00 ± 0.00** | **1.00 ± 0.00** | 16.35 ± 1.25 | **0.10 ± 0.00** | 102.23 ± 13.14 |
| HELOC | EA | 4.60 ± 1.14 | **1.00 ± 0.00** | **1.00 ± 0.00** | 1.90 ± 0.09 | −0.02 ± 0.03 | 338.84 ± 43.44 |
| | GLANCE | 10.00 ± 0.00 | **1.00 ± 0.00** | 0.95 ± 0.01 | 1.00 ± 0.07 | −0.01 ± 0.01 | 116.31 ± 16.93 |
| | TCREx | 26.80 ± 21.02 | **1.00 ± 0.00** | 0.94 ± 0.07 | **0.07 ± 0.05** | **0.05 ± 0.00** | **0.13 ± 0.07** |
| | OUR$_{group}$ | 16.80 ± 2.56 | **1.00 ± 0.00** | **1.00 ± 0.00** | 0.48 ± 0.06 | 0.02 ± 0.01 | 169.58 ± 24.21 |
| LAW | EA | 4.40 ± 1.95 | **1.00 ± 0.00** | **1.00 ± 0.00** | 1.13 ± 0.07 | −0.12 ± 0.01 | 121.26 ± 44.08 |
| | GLANCE | 2.00 ± 0.00 | **1.00 ± 0.00** | 0.95 ± 0.03 | 0.53 ± 0.05 | −0.05 ± 0.02 | 96.32 ± 15.61 |
| | TCREx | 5.00 ± 2.00 | **1.00 ± 0.00** | **1.00 ± 0.00** | **0.11 ± 0.09** | 0.03 ± 0.00 | **0.00 ± 0.00** |
| | OUR$_{group}$ | 4.40 ± 1.36 | **1.00 ± 0.00** | **1.00 ± 0.00** | 0.36 ± 0.02 | **0.04 ± 0.01** | 77.31 ± 60.42 |
| MOONS | EA | 5.20 ± 2.05 | **1.00 ± 0.00** | **1.00 ± 0.00** | 1.03 ± 0.00 | −0.14 ± 0.01 | 131.36 ± 50.25 |
| | GLANCE | 3.00 ± 0.00 | **1.00 ± 0.00** | 0.84 ± 0.14 | 0.53 ± 0.03 | −0.02 ± 0.02 | 91.44 ± 6.34 |
| | TCREx | 6.00 ± 0.00 | **1.00 ± 0.00** | 0.83 ± 0.15 | **0.10 ± 0.05** | 0.00 ± 0.01 | **0.00 ± 0.00** |
| | OUR$_{group}$ | 10.80 ± 0.98 | **1.00 ± 0.00** | **1.00 ± 0.00** | 0.46 ± 0.04 | **0.02 ± 0.00** | 42.47 ± 25.88 |
| WINE | EA | 1.00 ± 0.00 | **1.00 ± 0.00** | **1.00 ± 0.00** | 1.39 ± 0.26 | −0.03 ± 0.03 | 16.66 ± 0.50 |
| | GLANCE | 2.00 ± 0.00 | **1.00 ± 0.00** | 0.84 ± 0.10 | 0.70 ± 0.09 | 0.05 ± 0.01 | 7.2 ± 3.48 |
| | TCREx | 15.40 ± 11.28 | **1.00 ± 0.00** | **1.00 ± 0.00** | **0.09 ± 0.15** | 0.05 ± 0.01 | **0.00 ± 0.00** |
| | OUR$_{group}$ | 1.00 ± 0.00 | **1.00 ± 0.00** | **1.00 ± 0.00** | 0.81 ± 0.07 | **0.07 ± 0.01** | 32.41 ± 23.19 |

(Liu et al., 2012) to evaluate whether the CFs are realistic with respect to the target class distribution. The extended evaluation within more metrics is available in Appendix J.4. For methods that produce CFs via tree structures, we calculate these metrics by first applying each instance leaf-specific action to generate its counterfactual, then evaluating the metrics individually before aggregating across the dataset.

**Baselines** We benchmarked our method against various approaches across local, global, and group-wise configurations to ensure a comprehensive comparison of effectiveness and applicability at different levels of explanation. **For the global configuration**, we compared against GLOBE-CE (Ley et al., 2023) and GLANCE (Kavouras et al., 2024) in its global option (with only one group), as these represent state-of-the-art global CF methods, providing robust baselines for evaluating global coherence and plausibility. **For group-wise counterfactual explanations**, we evaluated our method against GLANCE, EA (Artelt & Gregoriades, 2024), and T-CREx (Bewley et al., 2024), which are designed to produce coherent and interpretable group-wise CFs. **For the local configuration**, we compared against several methods: the foundational gradient-based CF method by Wachter et al.

Table 3: Comparative analysis: Local Methods.

| DATASET | METHOD | COVERAGE↑ | VALID.↑ | L2↓ | ISOFOREST↑ | TIME(S)↓ |
|---------|--------|-----------|---------|-----|-----------|----------|
| BLOBS | DiCE | **1.00 ± 0.00** | **1.00 ± 0.00** | 0.51 ± 0.03 | −0.1 ± 0.00 | 8.15 ± 5.24 |
| | WACH | 0.99 ± 0.03 | **1.00 ± 0.00** | **0.23 ± 0.01** | −0.04 ± 0.00 | **0.22 ± 0.05** |
| | CCHVAE | **1.00 ± 0.00** | **1.00 ± 0.00** | 0.37 ± 0.05 | −0.06 ± 0.01 | 2.15 ± 0.62 |
| | PPCEF | **1.00 ± 0.00** | **1.00 ± 0.00** | 0.47 ± 0.01 | **0.04 ± 0.00** | 19.55 ± 0.30 |
| | OUR$_{local}$ | **1.00 ± 0.00** | **1.00 ± 0.00** | 0.39 ± 0.01 | 0.03 ± 0.00 | 6.20 ± 0.20 |
| DIGITS | DiCE | **1.00 ± 0.00** | **1.00 ± 0.00** | 23.77 ± 0.99 | 0.03 ± 0.01 | 162.88 ± 15.52 |
| | WACH | **1.00 ± 0.00** | **1.00 ± 0.00** | **2.10 ± 0.44** | 0.09 ± 0.00 | 16.41 ± 0.62 |
| | CCHVAE | **1.00 ± 0.00** | **1.00 ± 0.00** | 2.19 ± 0.24 | 0.04 ± 0.01 | **3.38 ± 0.52** |
| | PPCEF | **1.00 ± 0.00** | **1.00 ± 0.00** | 11.42 ± 0.05 | 0.10 ± 0.01 | 25.09 ± 0.40 |
| | OUR$_{local}$ | **1.00 ± 0.00** | **1.00 ± 0.00** | 11.41 ± 0.51 | **0.11 ± 0.00** | 18.58 ± 0.68 |
| HELOC | DiCE | **1.00 ± 0.00** | **1.00 ± 0.00** | 1.00 ± 0.06 | −0.01 ± 0.00 | 230.85 ± 26.00 |
| | WACH | **1.00 ± 0.00** | **1.00 ± 0.00** | **0.16 ± 0.02** | 0.06 ± 0.00 | 33.88 ± 4.98 |
| | CCHVAE | **1.00 ± 0.00** | **1.00 ± 0.00** | 0.59 ± 0.02 | **0.11 ± 0.00** | **14.60 ± 3.83** |
| | PPCEF | **1.00 ± 0.00** | 0.98 ± 0.02 | 0.42 ± 0.02 | 0.07 ± 0.00 | 24.31 ± 4.52 |
| | OUR$_{local}$ | **1.00 ± 0.00** | **1.00 ± 0.00** | 0.47 ± 0.01 | 0.08 ± 0.00 | 20.21 ± 2.02 |
| LAW | DiCE | **1.00 ± 0.00** | **1.00 ± 0.00** | 0.52 ± 0.01 | −0.05 ± 0.00 | 43.82 ± 9.62 |
| | WACH | 0.97 ± 0.05 | 1.00 ± 0.01 | **0.16 ± 0.01** | 0.05 ± 0.00 | 21.66 ± 3.91 |
| | CCHVAE | **1.00 ± 0.00** | **1.00 ± 0.00** | 0.31 ± 0.01 | **0.09 ± 0.01** | **0.28 ± 0.17** |
| | PPCEF | **1.00 ± 0.00** | 0.95 ± 0.01 | 0.32 ± 0.02 | 0.06 ± 0.00 | 20.63 ± 1.08 |
| | OUR$_{local}$ | **1.00 ± 0.00** | **1.00 ± 0.00** | 0.32 ± 0.00 | 0.05 ± 0.00 | 7.80 ± 0.29 |
| MOONS | DiCE | **1.00 ± 0.00** | **1.00 ± 0.00** | 0.55 ± 0.01 | −0.04 ± 0.01 | 17.85 ± 6.64 |
| | WACH | 0.97 ± 0.06 | **1.00 ± 0.00** | **0.16 ± 0.01** | −0.00 ± 0.00 | 0.23 ± 0.05 |
| | CCHVAE | **1.00 ± 0.00** | **1.00 ± 0.00** | 0.28 ± 0.01 | 0.02 ± 0.01 | **0.10 ± 0.04** |
| | PPCEF | **1.00 ± 0.00** | 0.98 ± 0.01 | 0.34 ± 0.04 | **0.03 ± 0.01** | 20.44 ± 1.75 |
| | OUR$_{local}$ | **1.00 ± 0.00** | **1.00 ± 0.00** | 0.30 ± 0.01 | **0.03 ± 0.00** | 7.32 ± 0.22 |
| WINE | DiCE | **1.00 ± 0.00** | **1.00 ± 0.00** | 0.72 ± 0.08 | 0.03 ± 0.01 | 0.70 ± 0.05 |
| | WACH | **1.00 ± 0.00** | **1.00 ± 0.00** | **0.43 ± 0.08** | 0.03 ± 0.02 | 0.10 ± 0.02 |
| | CCHVAE | **1.00 ± 0.00** | **1.00 ± 0.00** | 0.79 ± 0.05 | **0.09 ± 0.00** | **0.02 ± 0.00** |
| | PPCEF | **1.00 ± 0.00** | **1.00 ± 0.00** | 0.66 ± 0.05 | 0.07 ± 0.01 | 12.41 ± 0.52 |
| | OUR$_{local}$ | **1.00 ± 0.00** | **1.00 ± 0.00** | 0.69 ± 0.07 | 0.05 ± 0.01 | 5.49 ± 0.32 |

(2017) (Wach), which serves as a widely recognized baseline; PPCEF (Wielopolski et al., 2024), which employs a similar approach using normalizing flow models for plausibility and is particularly relevant as as our local configuration mathematically reduces to PPCEF's formulation when setting specific parameters ($K = N$, $\mathbf{S} = \mathbf{K} = \mathbb{I}$, $\lambda_s = \lambda_k = \lambda_d = 0$), as detailed in Appendix I.3; CCH-VAE (Pawelczyk et al., 2020), which also focuses on plausibility through generative modeling; and DiCE (Mothilal et al., 2020), which is used by both GLANCE and EA for prior clustering, making it a relevant comparison for local CFs.

**Experiment Results** The results are reported in Tables 1, 2, and 3, presenting comparative performance across six datasets with mean values and standard deviations over multiple runs. To validate the robustness of observed differences, we conducted Friedman tests across all configurations, which revealed statistically significant differences among methods for all evaluated metrics ($p < 0.05$); detailed statistical analysis is provided in Appendix J.5. Our proposed method consistently outperformed baseline approaches across all granularity levels: global, group-wise, and local.

**In the global configuration** (Table 1), our framework achieved perfect or near-perfect validity across all datasets except Moons, substantially outperforming GLOBE-CE, which achieved 0.00 validity on Digits. GLANCE showed lower validity on multiple datasets. For plausibility, OUR$_{global}$ consistently achieved the highest scores, demonstrating superior data manifold alignment compared to baselines, which often produced negative scores. Post-hoc pairwise comparisons confirmed these improvements are statistically significant (OUR vs. GLANCE: $p < 0.001$ for both validity and plausibility). Regarding proximity, our method achieved the best performance on Heloc and competitive results elsewhere, balancing minimal feature changes with plausibility. Notably, OUR$_{global}$ was significantly faster than GLANCE while maintaining superior quality metrics.

**For group-wise counterfactuals** (Table 2), our approach identified compact, interpretable group structures while maintaining high validity and plausibility. On all datasets, OUR$_{group}$ achieved perfect validity scores (1.00), matching or exceeding all baselines. For plausibility, OUR$_{group}$ consistently achieved positive IsoForest scores, statistically significantly outperforming EA and GLANCE ($p < 0.001$), while matching the performance of TCREx, yet identifying substantially fewer groups. For instance, on Digits, OUR$_{group}$ identified $2.80 \pm 1.83$ groups versus TCREx's $91.00 \pm 50.76$. The proximity scores demonstrate that our counterfactuals required reasonable feature changes while en-

suring realistic outcomes. An ablation study on the number of groups (see Appendix G) confirms that while increasing groups improves plausibility, the benefits plateau, validating our regularization approach.

**At the local level** (Table 3), all methods achieved perfect or near-perfect coverage and validity, making plausibility the key differentiator. $OUR_{local}$ significantly surpassed DiCE and Wach in plausibility across all datasets ($p < 0.001$). When compared to PPCEF and CCHVAE, our approach achieved statistically comparable performance (no significant differences, $p > 0.05$), demonstrating that our unified framework matches specialized plausibility-focused methods. While CCHVAE demonstrated strong plausibility and often the fastest execution time, and PPCEF showed comparable plausibility on several datasets, $OUR_{local}$ maintained consistently high plausibility across all datasets with reasonable computational efficiency. Our method generates CFs balancing validity, proximity, and plausibility. Proximity alone is insufficient as CFs must also be realistic (plausible). Methods achieving the lowest proximity often generate unrealistic examples that cross the decision boundary minimally but lie in low-density regions.

Overall, the evaluation demonstrates that our method excels in validity and plausibility across all granularities, with statistically significant improvements confirmed through rigorous testing. It maintains competitive proximity scores, effectively balancing plausibility and actionability. Furthermore, our group-wise approach, integrating probabilistic plausibility criteria, enhances performance by consistently achieving plausible results while maintaining reasonable proximity. This highlights an effective trade-off between plausibility and distance, showcasing the practical utility and effectiveness of our unified framework.

## 5.2 CASE STUDY 1: CREDIT SCORING WITH HELOC DATASET

The dataset comprises HELOC credit line applications aimed at predicting whether applicants will repay their credit lines within two years. We selected five financial indicators (*Number of Satisfactory Trades*, *Net Fraction of Revolving Burden*, *Net Fraction of Installment Burden*, *Number of Revolving Trades with Balance*, *Number of Installment Trades with Balance*) for their potential to enable rapid behavioral adjustments. By allowing the selection of only a subset of variables, enforcing monotonicity constraints where features can change in only one direction, and specifying feature ranges, our method ensures actionability by focusing on financially adjustable features within realistic limits. Implementation details are provided in Appendix C. Specifically, we applied the following constraints: Number of Satisfactory Trades can only increase (reflecting improved credit standing), Net Fraction of Revolving Burden and Net Fraction of Installment Burden can only decrease (indicating reduced debt utilization), Number of Revolving Trades with Balance can only decrease (showing debt consolidation), while Number of Installment Trades with Balance can both increase or decrease (allowing flexibility in loan management strategies). These indicators facilitate immediate changes, such as simulating the effects of a rejected credit scenario. Our method generated CFs, optimizing them into six groups. The proposed actions are illustrated in Figure 4. The results reveal diverse group-specific recommendations. Although some groups prioritize increasing satisfactory trades, others focus on reducing revolving burdens or trades. In addition, the groups differ significantly in size, which highlights potential for subgroup analysis. A detailed interpretation is provided in Appendix J.2.

## 5.3 CASE STUDY 2: HANDWRITTEN DIGIT TRANSFORMATIONS WITH DIGITS DATASET

Figure 2 demonstrates our method's application to the Digits dataset, presenting group-wise counterfactual explanations for two cases. In Figure 2a, the origin class is 9, transitioning to the desired class 0. In Figure 2b, the origin class is 6, transitioning to the desired class 3. Our method clusters instances into three groups, ensuring that instances within the same group require similar modifications to achieve their counterfactuals.

In Figure 2a, the first group demands substantial changes, as shown by prominent shifts in the change vector, while the third group requires fewer adjustments, indicating an easier path to the desired class. This variation underscores our method's ability to differentiate the effort required for different groups to reach the target class. Figure 2b highlights that the third group uniquely requires a subtraction in the lower-right corner, while the first and second groups do not exhibit

(a) Counterfactual explanations for class 9 with the desired class 0.

(b) Counterfactual explanation for class 6 with desired class 3.

Figure 2: CFs for different digit pairs, showing the transformation process between different digit classes. Each row represents a distinct group. Original images are on the left, shifting vectors are in the middle column, and CFs are on the right. Red pixels in the shifting vector indicate subtracted values, while blue pixels indicate added values.

significant changes in this region. This distinction demonstrates how our method tailors group-specific counterfactuals based on structural and feature differences.

These findings confirm our method's capability to produce interpretable and group-specific counterfactual explanations for image data, offering insights into the transformations needed to achieve GWCFs for diverse instance groups.

## 6 CONCLUSIONS

In this work, we introduced a unified method for generating counterfactual explanations at the local, group-wise, and global levels. Our approach dynamically adapts to different levels of granularity, eliminating the need for separate clustering and counterfactual generation steps. By formulating a counterfactual search as a single optimization task, we efficiently generate explanations that balance validity, proximity, and plausibility while optimizing group granularity. Additionally, we integrate probabilistic plausibility constraints within global and group-wise counterfactual explanations, ensuring that generated recourse suggestions remain realistic and actionable. The experimental results demonstrate the effectiveness of our approach across multiple datasets and classification models. In particular, we showed that our group-wise method produces a relatively small number of meaningful and interpretable groups, capturing distinct patterns within the data. Compared to state-of-the-art methods, our framework achieves superior validity while maintaining competitive plausibility and proximity. This method provides a valuable tool for enhancing transparency, accountability, and trust in machine learning by offering a comprehensive understanding of model behavior. It supports informed decision-making and advances research in model debugging and decision support systems.

## REPRODUCIBILITY STATEMENT

To ensure reproducibility of our results, we provide comprehensive implementation details and experimental configurations throughout this work. The complete source code for our unified counterfactual explanation framework will be made publicly available on GitHub upon acceptance. Our mathematical formulation is fully specified in Section 4.2, including all loss components, regularization terms, and optimization objectives. Detailed experimental protocols are described in Section 5, with comprehensive hyperparameter settings, baseline comparisons, and evaluation metrics. Complete dataset descriptions, model architectures, and training procedures are provided in Appendix E.1 and E.2. The computational environment and resource requirements are documented in Appendix E.3. All experimental results include mean values and standard deviations across five-fold cross-validation, with detailed numerical results presented in Appendix J. Our ablation studies (Appendix I) provide thorough analysis of individual components, enabling researchers to understand the contribution of each element. The normalizing flow implementation for plausibility estimation is detailed in Appendix B, and actionability constraints are fully specified in Appendix C.

## ETHICS STATEMENT

We acknowledge and adhere to the ICLR Code of Ethics in all aspects of this research, which aims to contribute positively to society by advancing AI transparency and interpretability through improved counterfactual explanations. Our method is designed to make AI systems more accountable and trustworthy, supporting fairer decision-making across local, group-wise, and global explanation levels. We use only publicly available datasets following established privacy-preserving practices, with no collection of new personal data or re-identification attempts. We acknowledge potential dual-use concerns where explanation techniques could be misused to game AI systems, emphasizing the need for responsible deployment with appropriate governance frameworks, particularly in high-stakes domains. We provide comprehensive disclosure of our method capabilities and limitations and we remain committed to the responsible development of explainable AI techniques.

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

## A    COMPARISON OF COUNTERFACTUAL EXPLANATION TYPES

| Aspect | Local | Global | Group-Wise |
|---|---|---|---|
| **Specificity** | High | Low | Moderate |
| **Scalability** | Low (instance-specific) | High | Moderate-high |
| **Fairness Analysis** | Limited | Weak | Strong |
| **Actionability** | High (per instance) | Low | High (per group) |
| **Interpretability** | Complex for stakeholders | Abstract | Balanced |
| **Privacy Concerns** | Higher risk (individuals) | Minimal | Minimal |

Table 4: Comparison of Local, Global, and Group-Wise Counterfactual Explanations

Table 4 provides a detailed comparison of the three primary types of counterfactual explanations: Local, Global, and Group-Wise. It highlights their respective strengths, limitations, and potential use cases. This comparison builds on the frameworks and analyses presented in related works (Wachter et al., 2017; Artelt & Hammer, 2020; Karimi et al., 2022; Guidotti, 2022; Ley et al., 2022; Kavouras et al., 2024; Artelt & Gregoriades, 2024)

## B    DENSITY ESTIMATIONS USING NORMALIZING FLOWS

Normalizing Flows have gained significant traction in generative modeling due to their flexibility and the straightforward training process through direct negative log-likelihood (NLL) optimization. This flexibility is rooted in the change-of-variable technique, which maps a latent variable $\mathbf{z}$ with a known prior distribution $p(\mathbf{z})$ to an observed variable $\mathbf{x}$ with an unknown distribution. This mapping is achieved through a series of invertible (parametric) functions: $\mathbf{x} = \mathbf{f}_K \circ \cdots \circ \mathbf{f}_1(\mathbf{z}, y)$. Given a known prior $p(\mathbf{z})$ for $\mathbf{z}$, the conditional log-likelihood for $\mathbf{x}$ is formulated as:

$$\log \hat{p}_F(\mathbf{x}|y) = \log p(\mathbf{z}) - \sum_{k=1}^{K} \log \left| \det \frac{\partial \mathbf{f}_k}{\partial \mathbf{z}_{k-1}} \right|, \tag{17}$$

where $\mathbf{z} = \mathbf{f}_1^{-1} \circ \cdots \circ \mathbf{f}_K^{-1}(\mathbf{x}, y)$ is a result of the invertible mapping. A key challenge in normalizing flows is the choice of the invertible functions $\mathbf{f}_K, \ldots, \mathbf{f}_1$. Several solutions have been proposed in the literature to address this issue with notable approaches, including NICE (Dinh et al., 2015), RealNVP (Dinh et al., 2017), and MAF (Papamakarios et al., 2017).

For a given training set $\mathcal{D} = \{(\mathbf{x}_n, h(\mathbf{x}_n))\}_{n=1}^{N}$ we simply train the conditional normalizing flow by minimizing negative log-likelihood:

$$Q = -\sum_{n=1}^{N} \log \hat{p}_F(\mathbf{x}_n|y_n), \tag{18}$$

where $\log \hat{p}_F(\mathbf{x}_n|y_n)$ is defined by eq. equation 17. The model is trained using a gradient-based approach applied to the flow parameters stored in $\mathbf{f}_k$ functions.

## C    SATISFYING ACTIONABILITY CONTRAINT

In our work we enforce actionability constraint by controlling the direction of the gradient. Specifically, before applying each gradient step, the sign of the gradient is checked to determine whether it is positive or negative. For features such as age, where changes are only allowed in one direction (e.g., increasing but not decreasing), the gradient is modified accordingly. Additionally, certain features may be completely non-actionable, such as demographic characteristics (e.g., race, gender) or historical records, which cannot be modified under any circumstances and must remain fixed during counterfactual generation. The new gradient value is computed as:

$$\frac{\partial \mathcal{L}}{\partial x_i}^{constrained} = \begin{cases} 0, & \text{if } x_i \in \mathcal{F}_{\text{non-decrease}} \text{ and } \frac{\partial \mathcal{L}}{\partial x_i} < 0, \\ 0, & \text{if } x_i \in \mathcal{F}_{\text{non-increase}} \text{ and } \frac{\partial \mathcal{L}}{\partial x_i} > 0, \\ 0, & \text{if } x_i \in \mathcal{F}_{\text{immutable}}, \\ \frac{\partial \mathcal{L}}{\partial x_i}, & \text{otherwise}, \end{cases} \quad (19)$$

where $\frac{\partial \mathcal{L}}{\partial x_i}$ represents the gradient value with respect to the $i$-th variable, $\mathcal{F}_{\text{non-decrease}}$ denotes the set of features subject to non-decreasing monotonicity constraints, indicating that these variables can only exhibit increases (e.g., age). $\mathcal{F}_{\text{non-increase}}$ is the set of features governed by non-increasing monotonicity constraints, signifying that these variables may only be decreased. $\mathcal{F}_{\text{immutable}}$ is the set of features that must remain invariant.

## D  LIMITATIONS

An inherent limitation in our methodology arises from the reliance on gradient-based optimization techniques within the data space. This approach requires the use of differentiable discriminative models and, consequently, does not support categorical variables. Nonetheless, the landscape of contemporary modeling techniques largely mitigates this constraint, as many modern models are differentiable or can be adapted to include differentiable components. This integration capacity ensures that our method remains applicable across various settings. While our method generates plausible counterfactuals that lie in dense regions of the data manifold, which may naturally exhibit greater stability under perturbations, we do not provide formal robustness guarantees.

## E  EXPERIMENT DETAILS

### E.1  DATASETS

Table 5: Dataset Characteristics and Model Performances. This table provides an overview of the datasets used in our experiments, including the number of samples ($N$), number of features ($D$), number of classes ($C$), accuracy of Logistic Regression (LR Acc.), Multi-Layer Perceptron (MLP Acc.), and the log density of the Masked Autoregressive Flow (MAF Log Dens.).

| DATASET | $N$ | $D$ | $C$ | LR ACC. | MLP ACC. | MAF LOG DENS. |
|---------|-----|-----|-----|---------|----------|---------------|
| MOONS | 1,024 | 2 | 2 | 0.90 | 0.99 | 1.44 |
| LAW | 2,220 | 3 | 2 | 0.75 | 0.79 | 1.54 |
| HELOC | 10,459 | 23 | 2 | 0.74 | 0.75 | 32.72 |
| WINE | 178 | 13 | 3 | 0.97 | 0.98 | 9.25 |
| BLOBS | 1,500 | 2 | 3 | 1.00 | 1.00 | 2.59 |
| DIGITS | 5,620 | 64 | 10 | 0.96 | 0.98 | -93.32 |

In Table 5, we provide detailed descriptions of the datasets utilized in our study: Moons[2], Law[3], Heloc[4], Wine[5], Blobs[6] and Digits[7]. The **Moons** dataset is an artificially generated set comprising two interleaving half-circles. It includes a standard deviation of Gaussian noise set at 0.01. The **Law** dataset (Wightman, 1998) originates from the Law School Admissions Council (LSAC) and is

---

[2]https://scikit-learn.org/1.6/modules/generated/sklearn.datasets.make_moons.html

[3]https://www.kaggle.com/datasets/danofer/law-school-admissions-bar-passage

[4]https://community.fico.com/s/explainable-machine-learning-challenge

[5]https://archive.ics.uci.edu/dataset/109/wine

[6]https://scikit-learn.org/1.6/modules/generated/sklearn.datasets.make_blobs.html

[7]https://archive.ics.uci.edu/dataset/80/optical+recognition+of+handwritten+digits

referred to in the literature as the Law School Admissions dataset. For our analysis, we selected the three features most correlated with the target variable: entrance exam scores (LSAT), grade-point average (GPA), and first-year average grade (FYA). The **Heloc** dataset (FICO, 2018), initially utilized in the 'FICO xML Challenge', consists of Home Equity Line of Credit (HELOC) applications submitted by real homeowners. This dataset contains numeric features summarizing information from applicants' credit reports. The primary objective is to predict whether the applicant will repay their HELOC account within a two-year period. This prediction is instrumental in determining the applicant's qualification for a line of credit. The **Wine** dataset (Aeberhard & Forina, 1992) comprises chemical analysis results for wines originating from the same region in Italy, produced from three distinct cultivars. This analysis quantified 13 different constituents present in each of the three wine varieties. The **Blobs** dataset is an artificially generated isotropic Gaussian blobs, characterized by equal variance. The **Digits** dataset (Alpaydin & Kaynak, 1998) is utilized for the optical recognition of handwritten digits. It consists of 32x32 bitmap images that are segmented into non-overlapping 4x4 blocks. Within each block, the count of 'on' pixels is recorded, resulting in an 8x8 input matrix. Each element of this matrix is an integer between 0 and 16.

### E.2 CLASSIFICATION MODELS

We used Logistic Regression (LR) and a Multilayer Perceptron (MLP) with two dense layers of 256 neurons each and ReLU activation. Both models utilized a softmax activation function in the output layer and were trained to minimize the cross-entropy loss function for up to 1000 epochs with an early stopping. These configurations ensured efficient training and robust evaluation across linear and non-linear settings.

### E.3 COMPUTATIONAL RESOURCES

In experiments, we used Python as the main programming language (Van Rossum & Drake Jr, 1995). Python with an open-source machine learning library PyTorch (Paszke et al., 2019) forms the backbone of our computational environment. We employed a batch-based gradient optimization method, which proved highly efficient by enabling the processing of complete test sets in a single batch. The experiments were executed on an M1 Apple Silicon CPU with 16GB of RAM, a configuration that provided enough computational power and speed to meet the demands of our algorithm.

## F GROUP DIVERSITY REGULARIZATION ABLATION STUDY

We conducted an ablation study to evaluate the effect of the group diversity regularization term by varying the weight parameter $\lambda_d$. All other parameters were fixed according to our base settings: $\lambda = 10^5$, $\lambda_p = 10^4$, $\lambda_s = 10^4$, and $\lambda_k = 10^3$. The evaluation was based on four key metrics. Validity was assessed by measuring the success rate of generating CFs that led to the desired class. Proximity was quantified using the $L_2$ distance between the original instances and their CFs. Plausibility was determined through the log density of the normalizing flow model, which evaluates the alignment of CFs with the data distribution. Diversity was analyzed using two metrics: the minimum pairwise cosine similarity among group shifting vectors and the mean distance of these vectors to their centroid.

The results presented in Table 6 demonstrated that setting $\lambda_d$ to lower or zero values led to highly similar group shifting vectors, as indicated by near-zero cosine similarity and smaller centroid distances. Increasing $\lambda_d$ enhanced diversity by producing less similar and more dispersed group shifting vectors, while maintaining plausibility and proximity.

Table 6: Impact of Group Diversity Regularization ($\lambda_d$) on our method performance.

| $\lambda_d$ | VALIDITY | PROXIMITY | PLAUSIBILITY | MIN PAIRWISE COSINE SIM. | MEAN CENTROID DISTANCE |
|---|---|---|---|---|---|
| 0.00 | $1.00 \pm 0.00$ | $0.49 \pm 0.04$ | $1.71 \pm 0.06$ | $0.00 \pm 0.00$ | $0.38 \pm 0.23$ |
| $10^{-1}$ | $1.00 \pm 0.00$ | $0.49 \pm 0.04$ | $1.70 \pm 0.06$ | $0.00 \pm 0.00$ | $0.36 \pm 0.23$ |
| $10^2$ | $1.00 \pm 0.00$ | $0.50 \pm 0.04$ | $1.72 \pm 0.06$ | $0.28 \pm 0.18$ | $4.31 \pm 0.35$ |
| $10^3$ | $1.00 \pm 0.00$ | $0.50 \pm 0.03$ | $1.70 \pm 0.04$ | $0.55 \pm 0.22$ | $4.73 \pm 0.56$ |

# G  NUMBER OF GROUPS ABLATION STUDY

We conducted an ablation study to investigate the impact of the number of groups on our method's performance across various metrics. The ablation study was performed using Logistic Regression (LR) and the HELOC dataset. By varying the number of groups from 2 to 10 while keeping all other hyperparameters fixed (using our base configuration: $\lambda = 10^5$, $\lambda_p = 10^4$, $\lambda_s = 10^4$, $\lambda_k = 10^4$, $\lambda_d = 10^1$), we analyzed the trade-offs between model complexity and performance.

Table 7: Impact of the Number of Groups on Method Performance. The table shows how varying the number of groups affects validity, proximity, plausibility metrics, and group diversity.

| GROUPS | VALIDITY↑ | L2↓ | ISOFOREST↑ | LOG DENSITY↑ | PROB. PLAUSIBILITY↑ | MIN PAIRWISE COSINE SIM. |
|--------|-----------|-----|-----------|--------------|---------------------|--------------------------|
| 2 | 0.98 | 0.37 | 0.06 | 30.15 | 0.51 | 7.72 |
| 3 | 0.99 | 0.39 | 0.06 | 30.41 | 0.54 | 2.04 |
| 4 | 0.98 | 0.38 | 0.07 | 31.06 | 0.58 | 0.54 |
| 5 | 0.99 | 0.38 | 0.07 | 31.27 | 0.59 | 0.26 |
| 6 | 0.99 | 0.39 | 0.07 | 31.08 | 0.60 | 0.20 |
| 7 | 0.99 | 0.39 | 0.07 | 31.80 | 0.62 | 0.17 |
| 8 | 0.99 | 0.40 | 0.07 | 31.47 | 0.63 | 0.14 |
| 9 | 0.99 | 0.38 | 0.07 | 31.85 | 0.64 | 0.17 |
| 10 | 0.99 | 0.38 | 0.07 | 32.07 | 0.65 | 0.14 |

The results presented in Table 7 demonstrate several key insights about the relationship between the number of groups and performance metrics:

**Validity** remains consistently high regardless of the number of groups, indicating that our method reliably generates valid counterfactuals across different group configurations.

**Probabilistic Plausibility** shows a clear positive correlation with the number of groups, increasing monotonically from 0.51 with 2 groups to 0.65 with 10 groups. This improvement suggests that more groups allow for better local approximations of the target distribution, enabling the generation of more plausible counterfactual explanations that better align with the data distribution.

**Group Diversity**, measured by the minimum pairwise cosine similarity, exhibits the biggest change. The similarity drops sharply from 7.72 (2 groups) to 2.04 (3 groups), then continues decreasing to stabilize around 0.14-0.17 for 7-10 groups. This pattern indicates that the largest gains in group diversity occur when moving from 2 to 7 groups, with minimal improvements beyond that point.

**Proximity** remains relatively stable across all configurations, suggesting that the number of groups does not significantly impact the distance between original instances and their counterfactuals.

These findings confirm that, while more groups can improve certain metrics, particularly probabilistic plausibility and group diversity, the benefits plateau after approximately 7 groups. This insight supports our adaptive approach that automatically determines the appropriate number of groups based on the specific dataset characteristics, balancing group diversity with performance.

# H  GPU ACCELERATION ABLATION STUDY

We conducted an ablation study comparing execution times between CPU and GPU implementations for our gradient-based optimization framework. While our main experiments used CPU for consistency with baselines, our approach is naturally compatible with GPU acceleration due to its gradient-based nature. All experiments were performed using 5-fold cross-validation to ensure robustness of timing measurements.

Tables 8 and 9 present execution times (in seconds) for our method on the HELOC dataset under global and group-wise configurations.

The results demonstrate that GPU acceleration provides significant performance improvements, particularly for group-wise configurations. While global settings (Table 8) show modest speedups (approximately 1.5x for LR), group-wise settings (Table 9) achieved dramatic improvements with 12.4x speedup for LR (from 230.07s to 18.48s) and 7.6x for MLP (from 237.69s to 31.43s). The standard

Table 8: Comparison of CPU vs. GPU Execution Times (seconds) for Global Settings on HELOC Dataset

| Model | CPU | GPU |
|---|---|---|
| LR | $27.45 \pm 3.58$ | $18.60 \pm 1.25$ |
| MLP | $32.47 \pm 4.01$ | $31.69 \pm 2.74$ |

Table 9: Comparison of CPU vs. GPU Execution Times (seconds) for Group-wise Settings on HELOC Dataset

| Model | CPU | GPU |
|---|---|---|
| LR | $230.07 \pm 21.10$ | $18.48 \pm 1.53$ |
| MLP | $237.69 \pm 30.88$ | $31.43 \pm 3.27$ |

deviations across the 5-fold cross-validation indicate that these performance improvements are consistent and reliable.

This ablation study further validates our choice of a gradient-based optimization framework, as it not only provides effective solutions for generating valid, plausible, and proximate counterfactual explanations but also leverages modern computational architectures to deliver substantial efficiency gains.

## I  HYPERPAPARAMETER VALUES ABLATION STUDY

To systematically evaluate the role of each loss term, we designed a series of experiments summarized in Table 10. The table combines three categories of settings: (i) **Individual Component Analysis** (E1–E5), where each term is activated independently to isolate its contribution, (ii) **Incremental Component Addition** (E6–E9), where loss terms are introduced step by step to observe cumulative effects, and (iii) **Alternative Configurations** (E10–E14), which test different weighting strategies. The corresponding quantitative results are presented separately in Table 11.

### I.1  KEY FINDINGS

**Individual Components (E1–E5).**  Validity-only (E1) reaches full validity but produces distant, implausible counterfactuals. Plausibility-only (E2) pulls counterfactuals closest to the source (L2≈0.18) with much higher plausibility, but validity collapses. Regularizers applied in isolation (E3–E5) fail to produce meaningful counterfactuals without the validity term, confirming their auxiliary nature.

**Incremental Additions (E6–E9).** Starting from validity+plausibility (E6) sharply improves plausibility and proximity over E1 while keeping validity at 1.00. Turning on the group-count regularizer (E7) and then adding sparsity (E8–E9) keeps validity high and nudges proximity slightly down (to ≈0.47–0.48) at the expense of a modest plausibility drop.

**Alternative Configurations (E10–E14).** All-nonzero weights with large magnitudes (E10) stay in the same proximity band as E8–E9 (≈0.47–0.48) with plausibility around 0.07. Mid-scale weights (E11–E13) show a gradual proximity/plausibility trade-off: as sparsity increases from E11 to E13, proximity slightly worsens (L2 rising from 0.49 to 0.50) while plausibility decreases modestly (from 0.07 to 0.06), with validity remaining perfect across all configurations. The lowest all-nonzero setting (E14) remains valid but is farthest from the source points and least plausible among the non-degenerate settings.

### I.2  CRITICAL TRADE-OFFS

Two central trade-offs emerge. First, **proximity vs. plausibility**: optimizing purely for plausibility (E2) yields the closest counterfactuals but breaks validity, while balancing both terms (E6, E9)

achieves practical usability. Second, **group constraints vs. proximity**: introducing group-based regularization (E7–E8) systematically increases the L2 distance, as counterfactuals must satisfy additional structural requirements.

Table 10: Experimental design for the ablation study. The table summarizes all configurations E1–E14, grouped into three categories: Individual Component Analysis (E1–E5), Incremental Component Addition (E6–E9), and Alternative Configurations (E10–E14). Each row specifies the weighting of the loss components: validity ($\lambda$), plausibility ($\lambda_p$), group sparsity ($\lambda_s$), number-of-groups regularization ($\lambda_k$), and diversity ($\lambda_d$). The rationale column provides the motivation for each setup.

| Exp. ID | $\lambda$ | $\lambda_p$ | $\lambda_s$ | $\lambda_k$ | $(\lambda_d)$ | Rationale |
|---|---|---|---|---|---|---|
| E1 | $10^5$ | 0 | 0 | 0 | 0 | Validity impact alone |
| E2 | 0 | $10^5$ | 0 | 0 | 0 | Plausibility impact alone |
| E3 | 0 | 0 | $10^5$ | 0 | 0 | Group sparsity impact alone |
| E4 | 0 | 0 | 0 | $10^5$ | 0 | Number-of-groups regularization alone |
| E5 | 0 | 0 | 0 | 0 | $10^5$ | Diversity regularization alone |
| E6 | $10^5$ | $10^4$ | 0 | 0 | 0 | Validity + Plausibility |
| E7 | $10^5$ | $10^4$ | 0 | $10^4$ | 0 | Add group-count regularization to E6 |
| E8 | $10^5$ | $10^4$ | $10^2$ | $10^4$ | $10^1$ | Turn on sparsity with small weight |
| E9 | $10^5$ | $10^4$ | $10^3$ | $10^4$ | $10^1$ | Increase sparsity while keeping validity/plausibility high |
| E10 | $10^5$ | $10^4$ | $10^4$ | $10^4$ | $10^1$ | All active with largest shared weights |
| E11 | $10^5$ | $10^3$ | $10^2$ | $10^3$ | $10^1$ | Mid-scale weights, lower sparsity |
| E12 | $10^5$ | $10^3$ | $10^3$ | $10^3$ | $10^1$ | Mid-scale balanced weights |
| E13 | $10^5$ | $10^3$ | $10^4$ | $10^3$ | $10^1$ | Mid-scale plausibility/group, stronger sparsity |
| E14 | $10^5$ | $10^2$ | $10^2$ | $10^2$ | $10^1$ | Lowest all-on configuration (closest to "light" regularization) |

Table 11: Complete Ablation Study Results across configurations E1–E14.

| Exp. ID | Validity↑ | Proximity (L2)↓ | IsoForest↑ | Log Density↑ | Prob. Plausibility↑ | Group Num.↓ |
|---|---|---|---|---|---|---|
| E1 | $1.00\pm0.00$ | $1.01\pm0.07$ | $-0.04\pm0.00$ | $-76.44\pm21.71$ | $0.01\pm0.01$ | $510.20\pm48.30$ |
| E2 | $0.05\pm0.00$ | $0.18\pm0.02$ | $0.07\pm0.00$ | $32.10\pm0.59$ | $0.29\pm0.07$ | $1022.00\pm35.38$ |
| E3 | — | — | — | — | — | — |
| E4 | — | — | — | — | — | — |
| E5 | — | — | — | — | — | — |
| E6 | $1.00\pm0.00$ | $0.50\pm0.06$ | $0.02\pm0.01$ | $14.60\pm2.24$ | $0.09\pm0.01$ | $327.40\pm47.62$ |
| E7 | $1.00\pm0.00$ | $0.47\pm0.06$ | $0.02\pm0.01$ | $11.18\pm2.73$ | $0.07\pm0.01$ | $16.80\pm2.79$ |
| E8 | $1.00\pm0.00$ | $0.48\pm0.06$ | $0.02\pm0.01$ | $11.12\pm2.75$ | $0.07\pm0.01$ | $16.60\pm2.87$ |
| E9 | $1.00\pm0.00$ | $0.48\pm0.06$ | $0.02\pm0.01$ | $11.67\pm2.75$ | $0.07\pm0.01$ | $16.80\pm2.71$ |
| E10 | $1.00\pm0.00$ | $0.48\pm0.06$ | $0.02\pm0.01$ | $11.69\pm3.01$ | $0.07\pm0.01$ | $16.80\pm2.56$ |
| E11 | $1.00\pm0.00$ | $0.49\pm0.06$ | $0.02\pm0.01$ | $10.62\pm2.88$ | $0.07\pm0.01$ | $87.40\pm6.65$ |
| E12 | $1.00\pm0.00$ | $0.50\pm0.06$ | $0.02\pm0.01$ | $10.03\pm3.09$ | $0.06\pm0.01$ | $37.80\pm6.01$ |
| E13 | $1.00\pm0.00$ | $0.50\pm0.06$ | $0.02\pm0.01$ | $9.96\pm3.28$ | $0.06\pm0.01$ | $41.80\pm4.62$ |
| E14 | $1.00\pm0.00$ | $0.54\pm0.05$ | $0.01\pm0.01$ | $5.99\pm2.40$ | $0.04\pm0.01$ | $52.00\pm5.90$ |

### I.3 Local Configuration Special Case

We provide explicit clarification on how our framework relates to PPCEF in the local setting. With specific parameter configuration ($K = N$, $\mathbf{S} = \mathbf{K} = \mathbb{I}$, $\lambda_s = \lambda_k = \lambda_d = 0$), our unified framework mathematically reduces to $N$ independent PPCEF Wielopolski et al. (2024) optimizations:

- When $K = N$: each instance has its own dedicated shifting vector in $\mathbf{D}_{GW} \in \mathbb{R}^{N \times D}$
- When $\mathbf{S} = \mathbb{I}_{N \times N}$: each instance selects only its own corresponding vector (no grouping)
- When $\mathbf{K} = \mathbb{I}_{N \times N}$: all magnitude scalers equal 1 (no scaling)

Under these conditions, our general group-wise formulation (Eq. 7) simplifies to:

$$\mathbf{X}'_{GW} = \mathbf{X}_0 + \mathbb{I} \cdot \mathbb{I} \cdot \mathbf{D}_{GW} = \mathbf{X}_0 + \mathbf{D}_{GW}$$

For each instance $n$: $\mathbf{x}'_n = \mathbf{x}_{n,0} + \mathbf{d}_n$, where $\mathbf{d}_n$ is the $n$-th row of $\mathbf{D}_{GW}$ (an independent, instance-specific shift vector).

The optimization objective decouples into $N$ independent problems:

$$\arg \min_{\mathbf{d}_n} \quad d(\mathbf{x}_{n,0}, \mathbf{x}'_n) + \lambda \cdot \ell_v(h(\mathbf{x}'_n), y'_n) + \lambda_p \cdot \ell_p(\mathbf{x}'_n, y'_n)$$

This is precisely PPCEF's formulation (Eq. 6 from Wielopolski et al. (2024)). Thus, our local configuration essentially generalizes PPCEF approach within our broader unified architecture.

## J ADDITIONAL RESULTS

### J.1 METHODS VISUALIZATION

This section provides an in-depth analysis of the methods, focusing on two main aspects: the variation in resulting explanations across global, group-wise, and local contexts, and the visual assessment of plausibility for our method compared to reference methods, as illustrated in Figure 3. Initial observations (blue and red dots) and final counterfactual explanations (orange dots) transition across the Multilayer Perceptron decision boundary (green line) into a probabilistically plausible region (red area), where the density satisfies plausibility thresholds.

For the reference methods, all produce valid counterfactuals, but with varying degrees of plausibility. The **GLOBE-CE** method generates counterfactual explanations just over the decision boundary, resulting in highly implausible outcomes. The **GLANCE** method achieves some plausible counterfactuals but struggles to balance group granularity with plausibility effectively. The **DiCE** method produces counterfactuals that are often significantly distant from the initial observations, reducing their practical relevance.

Our method, when configured globally, also struggles to produce fully plausible results but tends to prioritize a global shifting vector that maximizes plausibility for as many instances as possible. In the group configuration, our method successfully clusters distant instances into the same group, generating valid and plausible counterfactuals. Both the group-wise and local configurations demonstrate the ability to produce counterfactuals that are both valid and plausible, offering a balanced approach to explanation generation.

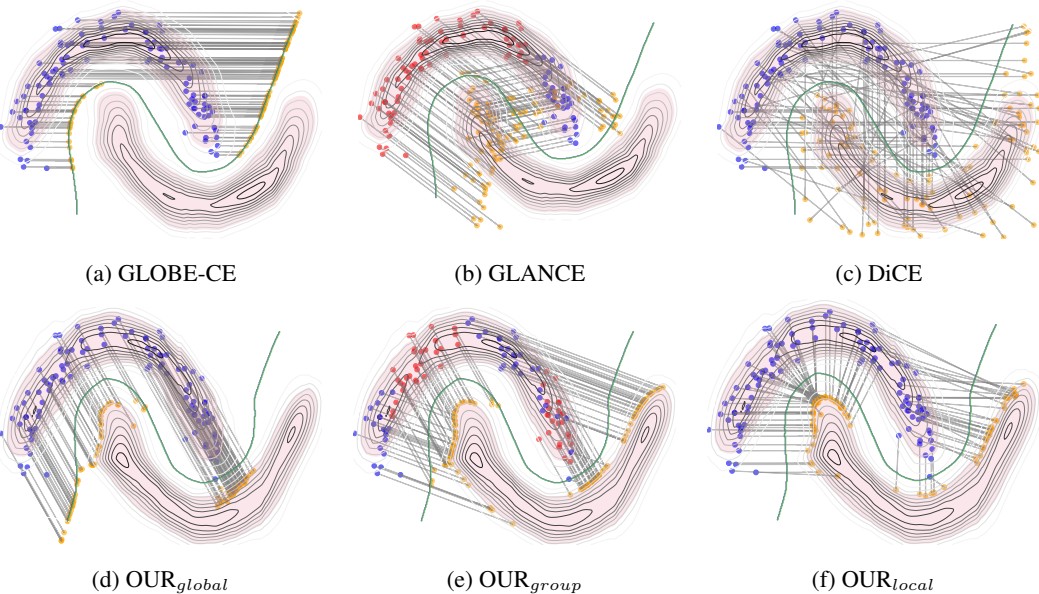

(a) GLOBE-CE        (b) GLANCE        (c) DiCE

(d) OUR$_{global}$        (e) OUR$_{group}$        (f) OUR$_{local}$

Figure 3: Visual comparison of the efficacy of various baseline counterfactual explanation methods with our method in traversing the decision boundary of a MLP model.

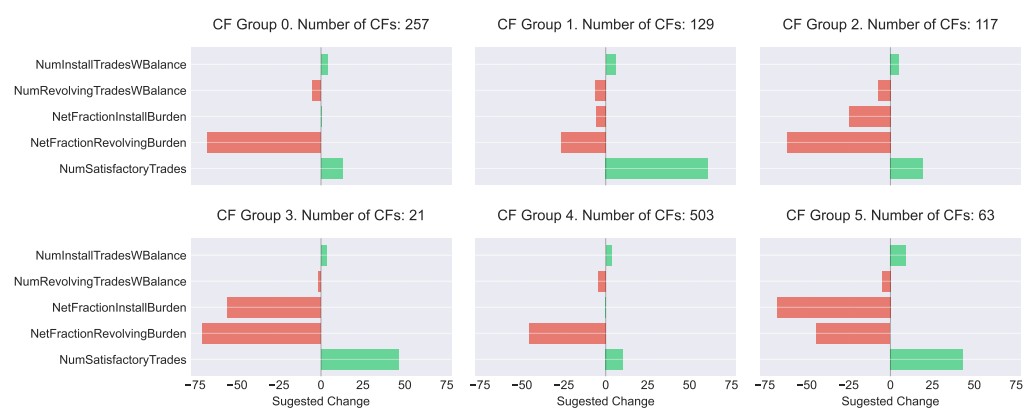

Figure 4: The figure illustrates group-wise counterfactual explanations generated using our method on the HELOC dataset with an MLP model. Each subplot highlights group-specific recommendations for financial adjustments, showing the mean change for selected financial indicators normalized over the average magnitude of changes. For each group, the number of instances is also provided.

## J.2 Case Study 1: Credit Scoring with HELOC Dataset

This subsection presents a detailed interpretation of the practical use case illustrated in Figure 4. We carefully selected features based on their varying degrees of actionability and impact on credit assessment, prioritizing those that individuals could realistically modify through specific financial behaviors. The selected actionable features include:

- **Number of Satisfactory Trades** – Represents successfully completed credit engagements with good standing. This feature can only increase through maintaining existing accounts and establishing new ones over time.

- **Net Fraction of Revolving Burden** – The ratio of revolving credit utilized to the total credit limit. This highly actionable feature can be changed quickly and should decrease to improve outcomes, as lower utilization is generally preferred by lenders.

- **Net Fraction of Installment Burden** – The proportion of the installment debt relative to the original loan amount. This feature requires additional payments to decrease the burden through accelerated repayment.

- **Number of Revolving Trades with Balance** – Tracks ongoing revolving credit accounts with outstanding balances. This highly actionable feature can be decreased by completely paying off certain revolving accounts.

- **Number of Installment Trades with Balance** – Tracks ongoing installment credit accounts with outstanding balances. This feature can either increase (by taking on new loans) or decrease (by paying off existing loans).

This selection of features is particularly effective for counterfactual explanations because it provides a balanced approach to credit improvement. It combines both adjusting revolving burden and credit-building strategies (increasing satisfactory trades). Additionally, it addresses multiple dimensions that influence credit decisions by incorporating credit history depth, utilization rates, and account management practices across both revolving and installment credit types. For each group shown in Figure 4, we propose interpretations from the perspective of a user applying our method.

**Group 0** For individuals in this category, it is advisable to significantly decrease the **Net Fraction of Revolving Burden** while moderately increasing the **Number of Satisfactory Trades**. Minor adjustments include increasing the **Number of Installment Trades with Balance** and reducing the **Number of Revolving Trades with Balance**. This group likely has established credit but is overextended on revolving credit, necessitating debt reduction to enhance their creditworthiness.

**Group 1**   For this group, the primary strategy involves substantially increasing the **Number of Satisfactory Trades** while moderately reducing the **Net Fraction of Revolving Burden**. These individuals should make minor improvements by slightly decreasing the **Net Fraction of Installment Burden** and the **Number of Revolving Trades with Balance**, with a small increase in the **Number of Installment Trades with Balance**. This suggests consumers with thin credit profiles who need both credit-building and utilization management.

**Group 2**   Members of this group should focus on decreasing both the **Net Fraction of Revolving Burden** and the **Net Fraction of Installment Burden** substantially. They should moderately increase the **Number of Satisfactory Trades** while slightly increasing the **Number of Installment Trades with Balance** and reducing the **Number of Revolving Trades with Balance**. This indicates consumers who are overextended across multiple credit products and need comprehensive debt reduction.

**Group 3**   Representing the smallest segment, these individuals require the most extensive changes: significant decreases in both the **Net Fraction of Revolving Burden** and the **Net Fraction of Installment Burden**, coupled with a substantial increase in the **Number of Satisfactory Trades**. Minor adjustments include slightly increasing the **Number of Installment Trades with Balance** and reducing the **Number of Revolving Trades with Balance**. This suggests severely overleveraged borrowers requiring comprehensive credit rehabilitation.

**Group 4**   As the largest group, explanations include moderately decreasing the **Net Fraction of Revolving Burden** while making minor improvements to other factors: slight increases in both the **Number of Satisfactory Trades** and the **Number of Installment Trades with Balance**, with a small reduction in the **Number of Revolving Trades with Balance**. This represents "typical" consumers who primarily need to address revolving debt utilization with minimal other adjustments.

**Group 5**   In this group, the explanation suggests substantial increases in the **Number of Satisfactory Trades** and moderate increases in the **Number of Installment Trades with Balance**. Significant decreases are needed in both the **Net Fraction of Revolving Burden** and the **Net Fraction of Installment Burden**, with minor reductions in the **Number of Revolving Trades with Balance**. This approach requires comprehensive credit improvement across all dimensions.

Across nearly all groups, enhancing the **Number of Satisfactory Trades** emerges as a critical factor in credit approval decisions. Reducing the **Net Fraction of Revolving Burden** is consistently beneficial across all groups, while the importance of managing the **Net Fraction of Installment Burden** varies significantly between segments. Most groups benefit from minor adjustments to account composition, with careful balance between revolving and installment credit products.

### J.3   Case Study 2: Handwritten Digit Transformations with Digits Dataset

Figure 5 illustrates these findings in the context of digit transformations. The rows compare counterfactual explanations with and without plausibility optimization for three digit instance pairs (9 to 0, 6 to 3, and 7 to 1). Without plausibility, our group-wise method partitions the data into two coarse groups, while incorporating plausibility refines the explanations into three distinct and interpretable clusters. This added granularity demonstrates the advantage of plausibility optimization in creating realistic and practical CFs.

In summary, incorporating probabilistic plausibility criteria yields outcomes that are less prone to outliers, potentially enhancing end-user usability. Moreover, within the framework of methods optimizing plausibility, we achieve results of comparable quality to the local counterfactual method, albeit with fewer shifting vectors.

### J.4   Extended Quantitative Evaluation

This section presents a comprehensive evaluation of our method compared to baseline counterfactual explanation techniques. All results are averaged over five cross-validation folds, with mean values and standard deviations reported in six detailed tables that fall into two categories:

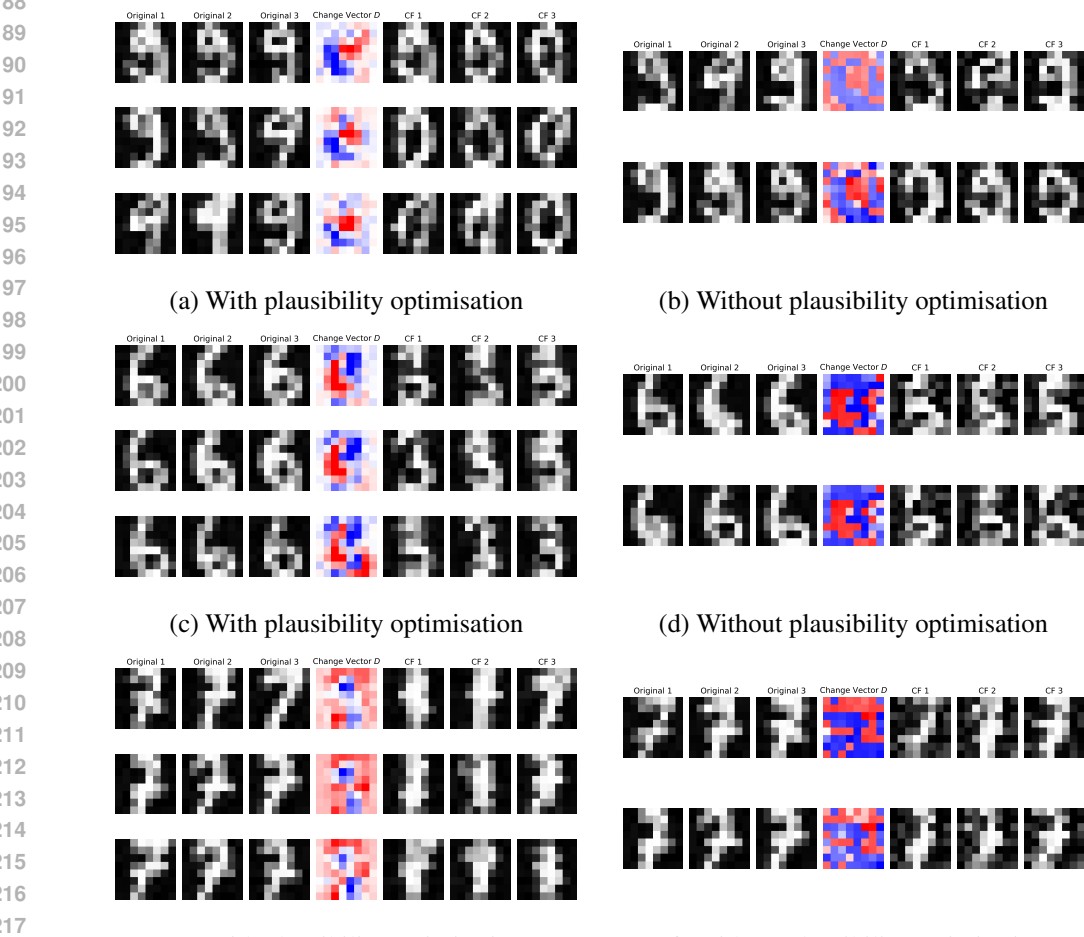

(a) With plausibility optimisation

(b) Without plausibility optimisation

(c) With plausibility optimisation

(d) Without plausibility optimisation

(e) With plausibility optimisation

(f) Without plausibility optimisation

Figure 5: Comparison of group-wise counterfactual explanations with and without plausibility optimisation for different digit pairs. Each pair of columns represents counterfactual explanations for a specific digit transformation (e.g., 9 to 0, 6 to 3, and 7 to 1). Without plausibility optimisation, the method clusters the problem into two groups. With plausibility optimisation, the method refines the counterfactuals into three distinct groups, ensuring more interpretable and realistic transformations. Original images are on the left, shifting vectors are in the middle column, and counterfactuals are on the right for each method. Red pixels in the shifting vector indicate subtracted values, while blue pixels indicate added values.

**Base Metrics Tables** (Tables 12, 14, and 16) contain the primary metrics calculation reported in Tables 1, 2, and 3, including execution times. **Plausibility and Cost Metrics Tables** (Tables 13, 15, and 17) provide additional metrics for a more thorough assessment of counterfactual plausibility and action cost. Following Guidotti (2022), we employ a comprehensive evaluation framework with these metrics:

**Base Metrics:**

- *Validity (Valid.)*: Success rate of counterfactuals in changing model predictions.

- *Proximity (L2)*: L2 distance between original and counterfactual instances.

- *Isolation Forest (IsoForest)*: Lower scores indicate more anomalous counterfactuals.

**Additional Plausibility Metrics:**

- *Local Outlier Factor (LOF)*: Higher values indicate more anomalous counterfactuals.

- *Log Density (Log. Dens.)*: Higher values indicate stronger alignment between counterfactuals and the target class distribution, as measured by a normalizing flow model.
- *Probabilistic Plausibility (Prob. Plaus.)*: Higher values indicate more counterfactuals satisfying Eq. equation 2b.

**Additional Cost Metric:**

- *Cost*: We adopt a cost metric proposed by Ley et al. (2023). Features are divided into 10 equal-sized bins where changing a feature value incurs a cost equal to the number of bin boundaries crossed.

For group-wise and global methods, we additionally report *Coverage* (percentage of instances with valid counterfactuals), while for group-wise methods, we also include the final number of identified groups (*Groups*).

We also conducted comparative analyses with additional baseline methods: AReS by Rawal & Lakkaraju (2020) and the method by Artelt & Hammer (2020) (Artelt). These methods were excluded from the main paper due to compatibility limitations: AReS does not support datasets with fewer than 3 features, while Artelt's method works exclusively with Logistic Regression models, making it impossible to evaluate with Multilayer Perceptron classifiers.

Tables 12 and 13 compare global CF methods. Our method consistently achieves perfect validity across nearly all datasets, whereas GLOBE-CE and GLANCE struggle particularly with the Digits dataset. Additionally, our method demonstrates superior probabilistic plausibility and notably higher Log Density scores, indicating better alignment with the target class distribution. While GLANCE often requires significantly longer execution times, our method maintains efficiency without compromising performance.

Tables 14 and 15 evaluate group-wise CF methods. Our approach shows strong adaptability across datasets, achieving perfect coverage and validity on all datasets. In contrast, EA completely fails with the Digits dataset, and both EA and GLANCE generally produce counterfactuals with substantially lower plausibility. Our method intelligently identifies an appropriate number of groups based on dataset characteristics, while maintaining consistently superior probabilistic plausibility scores compared to baselines. T-CREx, while efficient in execution time, produces much larger numbers of groups, which makes interpretation more difficult.

Tables 16 and 17 present results for local CF methods, comparing DiCE, Wachter (Wach), and CCHVAE with our approach. While all methods achieve high validity, our method consistently demonstrates perfect probabilistic plausibility while maintaining competitive L2 proximity. DiCE typically produces the least plausible counterfactuals, particularly with complex datasets, as evidenced by substantially negative Log Density values. CCHVAE performs well on some metrics but falls short on plausibility for datasets like Blobs and Moons. Our method balances execution time, proximity, and plausibility more effectively than competing approaches across all tested datasets and model types.

### J.5    STATISTICAL SIGNIFICANCE ANALYSIS

To assess the statistical significance of performance differences across methods, we applied the Friedman test, a non-parametric statistical test suitable for comparing multiple related samples. We performed separate Friedman tests for each metric within each configuration (global, group-wise, and local), with a significance level of $\alpha = 0.05$. For metrics where the Friedman test indicated significant differences ($p < 0.05$), we conducted post-hoc pairwise comparisons using the Wilcoxon signed-rank test with Bonferroni correction.

Table 18 presents the Friedman test results. All 14 metrics demonstrated statistically significant differences, providing strong evidence that the choice of counterfactual generation method substantially impacts performance.

**Discussion of Statistical Results**    The Friedman test results provide compelling validation for our experimental findings. In the global configuration, highly significant differences were observed across all metrics, with particularly strong evidence for validity ($p < 0.001$) and plausibility

Table 12: Comparative analysis of our method in **global configuration** with other CF methods across various datasets and classification models. Values are averaged over five cross-validation folds.

| | METHOD | VALID.↑ | L2↓ | IsoForest↑ | TIME(S)↓ |
|---|---|---|---|---|---|
| | | MLP | | | |
| BLOBS | GLOBE-CE | $0.99 \pm 0.01$ | $\mathbf{0.25 \pm 0.04}$ | $-0.06 \pm 0.03$ | $\mathbf{0.66 \pm 0.03}$ |
| | GLOBALGLANCE | $\mathbf{1.00 \pm 0.00}$ | $0.42 \pm 0.01$ | $0.01 \pm 0.00$ | $43.30 \pm 9.72$ |
| | OUR$_{global}$ | $\mathbf{1.00 \pm 0.00}$ | $0.48 \pm 0.01$ | $\mathbf{0.03 \pm 0.00}$ | $7.89 \pm 0.86$ |
| DIGITS | GLOBE-CE | $0.00 \pm 0.00$ | - | - | $\mathbf{0.95 \pm 0.08}$ |
| | GLOBALGLANCE | $0.30 \pm 0.07$ | $11.24 \pm 0.70$ | $0.09 \pm 0.01$ | $678.36 \pm 29.07$ |
| | OUR$_{global}$ | $\mathbf{1.00 \pm 0.00}$ | $\mathbf{17.08 \pm 0.54}$ | $\mathbf{0.1 \pm 0.00}$ | $31.48 \pm 5.28$ |
| HELOC | ARES | $0.28 \pm 0.06$ | $0.68 \pm 0.16$ | $0.02 \pm 0.02$ | $13.25 \pm 1.79$ |
| | GLOBE-CE | $\mathbf{1.00 \pm 0.00}$ | $0.52 \pm 0.03$ | $0.03 \pm 0.01$ | $\mathbf{2.02 \pm 0.18}$ |
| | GLOBALGLANCE | $0.97 \pm 0.01$ | $0.68 \pm 0.07$ | $-0.01 \pm 0.02$ | $99.89 \pm 44.14$ |
| | OUR$_{global}$ | $\mathbf{1.00 \pm 0.00}$ | $\mathbf{0.36 \pm 0.02}$ | $\mathbf{0.06 \pm 0.00}$ | $32.47 \pm 10.01$ |
| LAW | GLOBE-CE | $\mathbf{1.00 \pm 0.00}$ | $\mathbf{0.22 \pm 0.02}$ | $\mathbf{0.01 \pm 0.01}$ | $\mathbf{0.81 \pm 0.02}$ |
| | GLOBALGLANCE | $0.97 \pm 0.01$ | $0.45 \pm 0.02$ | $-0.04 \pm 0.01$ | $90.81 \pm 9.03$ |
| | OUR$_{global}$ | $\mathbf{1.00 \pm 0.00}$ | $0.38 \pm 0.01$ | $\mathbf{0.01 \pm 0.00}$ | $13.44 \pm 3.11$ |
| MOONS | GLOBE-CE | $\mathbf{1.00 \pm 0.00}$ | $\mathbf{0.30 \pm 0.03}$ | $-0.06 \pm 0.01$ | $\mathbf{0.65 \pm 0.01}$ |
| | GLOBALGLANCE | $0.68 \pm 0.05$ | $0.39 \pm 0.02$ | $-0.02 \pm 0.01$ | $77.97 \pm 9.11$ |
| | OUR$_{global}$ | $0.91 \pm 0.12$ | $0.45 \pm 0.04$ | $\mathbf{-0.01 \pm 0.01}$ | $9.55 \pm 1.37$ |
| WINE | GLOBE-CE | $\mathbf{1.00 \pm 0.00}$ | $\mathbf{0.73 \pm 0.20}$ | $0.04 \pm 0.02$ | $0.39 \pm 0.01$ |
| | GLOBALGLANCE | $0.57 \pm 0.17$ | $0.46 \pm 0.07$ | $0.06 \pm 0.01$ | $\mathbf{5.82 \pm 3.10}$ |
| | OUR$_{global}$ | $\mathbf{1.00 \pm 0.00}$ | $\mathbf{0.73 \pm 0.07}$ | $\mathbf{0.06 \pm 0.01}$ | $5.73 \pm 0.89$ |
| | | LR | | | |
| BLOBS | GLOBE-CE | $\mathbf{1.00 \pm 0.00}$ | $\mathbf{0.29 \pm 0.02}$ | $-0.08 \pm 0.00$ | $\mathbf{0.22 \pm 0.01}$ |
| | GLOBALGLANCE | $\mathbf{1.00 \pm 0.00}$ | $0.42 \pm 0.01$ | $0.02 \pm 0.00$ | $38.36 \pm 10.34$ |
| | OUR$_{global}$ | $\mathbf{1.00 \pm 0.00}$ | $0.5 \pm 0.02$ | $\mathbf{0.02 \pm 0.00}$ | $7.93 \pm 1.05$ |
| DIGITS | GLOBE-CE | $0.00 \pm 0.00$ | - | - | $\mathbf{0.16 \pm 0.01}$ |
| | GLOBALGLANCE | $0.50 \pm 0.11$ | $10.94 \pm 1.04$ | $0.09 \pm 0.00$ | $534.20 \pm 40.88$ |
| | OUR$_{global}$ | $\mathbf{1.00 \pm 0.00}$ | $\mathbf{15.61 \pm 0.47}$ | $\mathbf{0.1 \pm 0.00}$ | $34.46 \pm 8.66$ |
| HELOC | ARES | $0.18 \pm 0.13$ | $0.50 \pm 0.23$ | $0.03 \pm 0.02$ | $14.53 \pm 1.62$ |
| | GLOBE-CE | $\mathbf{1.00 \pm 0.00}$ | $\mathbf{0.32 \pm 0.05}$ | $0.05 \pm 0.01$ | $\mathbf{0.45 \pm 0.05}$ |
| | GLOBALGLANCE | $0.97 \pm 0.02$ | $0.61 \pm 0.06$ | $-0.00 \pm 0.02$ | $61.63 \pm 11.58$ |
| | OUR$_{global}$ | $\mathbf{1.00 \pm 0.00}$ | $0.33 \pm 0.03$ | $\mathbf{0.06 \pm 0.00}$ | $27.45 \pm 11.74$ |
| LAW | GLOBE-CE | $\mathbf{1.00 \pm 0.00}$ | $\mathbf{0.19 \pm 0.01}$ | $\mathbf{0.02 \pm 0.00}$ | $\mathbf{0.24 \pm 0.01}$ |
| | GLOBALGLANCE | $0.98 \pm 0.01$ | $0.47 \pm 0.04$ | $-0.05 \pm 0.01$ | $83.25 \pm 19.79$ |
| | OUR$_{global}$ | $\mathbf{1.00 \pm 0.00}$ | $0.39 \pm 0.02$ | $0.01 \pm 0.00$ | $12.71 \pm 2.75$ |
| MOONS | GLOBE-CE | $\mathbf{1.00 \pm 0.00}$ | $\mathbf{0.28 \pm 0.01}$ | $-0.01 \pm 0.01$ | $\mathbf{0.22 \pm 0.01}$ |
| | GLOBALGLANCE | $\mathbf{1.00 \pm 0.01}$ | $0.53 \pm 0.03$ | $-0.04 \pm 0.01$ | $67.90 \pm 11.41$ |
| | OUR$_{global}$ | $\mathbf{1.00 \pm 0.00}$ | $0.46 \pm 0.06$ | $\mathbf{0.00 \pm 0.01}$ | $11.95 \pm 2.41$ |
| WINE | GLOBE-CE | $\mathbf{1.00 \pm 0.00}$ | $\mathbf{0.73 \pm 0.17}$ | $0.03 \pm 0.02$ | $\mathbf{0.20 \pm 0.00}$ |
| | GLOBALGLANCE | $0.60 \pm 0.12$ | $0.47 \pm 0.05$ | $0.06 \pm 0.01$ | $2.77 \pm 1.14$ |
| | OUR$_{global}$ | $\mathbf{1.00 \pm 0.00}$ | $0.76 \pm 0.05$ | $\mathbf{0.06 \pm 0.01}$ | $6.07 \pm 0.27$ |

($p < 0.001$). Post-hoc pairwise comparisons revealed that our method significantly outperforms GLANCE in validity ($p < 0.001$) and achieves superior plausibility compared to both GLOBE-CE and GLANCE (both $p < 0.001$).

For the group-wise configuration, all five metrics demonstrated extremely strong significance ($p < 10^{-7}$), with L2 distance showing the strongest differentiation among methods ($p < 10^{-14}$). Post-hoc analysis confirmed that our method achieves significantly better plausibility than EA and GLANCE while maintaining competitive validity.

In the local configuration, all metrics again showed highly significant differences ($p < 10^{-8}$). Notably, our method demonstrated comparable plausibility to specialized plausibility-focused methods (PPCEF and CCHVAE) with no significant differences observed ($p = 0.064$ and $p = 0.984$, respectively), while significantly outperforming DiCE ($p < 0.001$). These results confirm that the performance gains of our unified approach are statistically robust and not due to random variation.

Table 13: Additional comparative plausibility and cost analysis of our method in **global configuration** with other CF methods across various datasets and classification models. Values are averaged over five cross-validation folds.

| | METHOD | PROB. PLAUS.↑ | LOG DENS.↑ | LOF↓ | COST↓ |
|---|---|---|---|---|---|
| | MLP | | | | |
| BLOBS | GLOBE-CE | $0.00 \pm 0.00$ | $-4.57 \pm 1.67$ | $2.04 \pm 0.18$ | $\mathbf{1.97 \pm 1.53}$ |
| | GLOBALGLANCE | $0.00 \pm 0.00$ | $-49.99 \pm 14.79$ | $1.11 \pm 0.02$ | $5.78 \pm 0.26$ |
| | $\text{OUR}_{global}$ | $\mathbf{0.92 \pm 0.03}$ | $\mathbf{2.89 \pm 0.1}$ | $\mathbf{1.04 \pm 0.01}$ | $6.65 \pm 0.66$ |
| DIGITS | GLOBE-CE | - | - | - | - |
| | GLOBALGLANCE | $0.00 \pm 0.00$ | $-285.44 \pm 21.71$ | $1.31 \pm 0.03$ | $\mathbf{27.45 \pm 1.99}$ |
| | $\text{OUR}_{global}$ | $\mathbf{0.72 \pm 0.09}$ | $\mathbf{-99.42 \pm 0.61}$ | $\mathbf{1.09 \pm 0.01}$ | $49.27 \pm 8.59$ |
| HELOC | ARES | $0.18 \pm 0.14$ | $19.60 \pm 14.31$ | $1.23 \pm 0.09$ | $13.42 \pm 3.24$ |
| | GLOBE-CE | $0.17 \pm 0.02$ | $-17.27 \pm 47.94$ | $1.47 \pm 0.09$ | $\mathbf{4.03 \pm 4.20}$ |
| | GLOBALGLANCE | $0.00 \pm 0.00$ | $-2.43 \pm 9.38$ | $1.67 \pm 0.10$ | $13.48 \pm 1.94$ |
| | $\text{OUR}_{global}$ | $\mathbf{0.46 \pm 0.01}$ | $\mathbf{29.25 \pm 0.4}$ | $\mathbf{1.15 \pm 0.01}$ | $10.75 \pm 4.96$ |
| LAW | GLOBE-CE | $0.37 \pm 0.05$ | $-14.5 \pm 28.64$ | $1.24 \pm 0.09$ | $\mathbf{2.22 \pm 1.79}$ |
| | GLOBALGLANCE | $0.34 \pm 0.10$ | $-0.26 \pm 0.61$ | $1.22 \pm 0.03$ | $6.00 \pm 0.41$ |
| | $\text{OUR}_{global}$ | $\mathbf{0.79 \pm 0.02}$ | $\mathbf{1.5 \pm 0.05}$ | $\mathbf{1.09 \pm 0.01}$ | $6.35 \pm 2.02$ |
| MOONS | GLOBE-CE | $0.00 \pm 0.00$ | $-17.53 \pm 10.28$ | $2.36 \pm 0.08$ | $\mathbf{3.07 \pm 1.84}$ |
| | GLOBALGLANCE | $0.30 \pm 0.07$ | $-2.04 \pm 0.64$ | $1.63 \pm 0.12$ | $5.19 \pm 0.61$ |
| | $\text{OUR}_{global}$ | $\mathbf{0.63 \pm 0.06}$ | $\mathbf{-0.33 \pm 0.9}$ | $1.48 \pm 0.18$ | $5.92 \pm 2.04$ |
| WINE | GLOBE-CE | $0.00 \pm 0.00$ | $-14.74 \pm 16.35$ | $1.86 \pm 0.6$ | $\mathbf{2.69 \pm 4.07}$ |
| | GLOBALGLANCE | $0.00 \pm 0.00$ | $-64.51 \pm 60.94$ | $1.20 \pm 0.04$ | $9.32 \pm 0.75$ |
| | $\text{OUR}_{global}$ | $\mathbf{0.95 \pm 0.11}$ | $\mathbf{7.78 \pm 0.18}$ | $\mathbf{1.09 \pm 0.03}$ | $21.40 \pm 4.06$ |
| | LR | | | | |
| BLOBS | GLOBE-CE | $0.00 \pm 0.00$ | $-6.03 \pm 0.76$ | $2.22 \pm 0.18$ | $\mathbf{2.45 \pm 1.34}$ |
| | GLOBALGLANCE | $0.00 \pm 0.00$ | $-69.32 \pm 21.46$ | $1.11 \pm 0.01$ | $6.07 \pm 0.18$ |
| | $\text{OUR}_{global}$ | $\mathbf{0.92 \pm 0.03}$ | $\mathbf{2.83 \pm 0.12}$ | $\mathbf{1.04 \pm 0.02}$ | $6.83 \pm 0.79$ |
| DIGITS | GLOBE-CE | - | - | - | - |
| | GLOBALGLANCE | $0.00 \pm 0.00$ | $-312.00 \pm 76.17$ | $1.32 \pm 0.04$ | $\mathbf{24.78 \pm 2.82}$ |
| | $\text{OUR}_{global}$ | $\mathbf{0.69 \pm 0.04}$ | $\mathbf{-100.41 \pm 0.31}$ | $\mathbf{1.1 \pm 0.01}$ | $45.70 \pm 9.08$ |
| HELOC | ARES | $0.07 \pm 0.14$ | $-49.16 \pm 97.83$ | $1.67 \pm 0.52$ | $10.12 \pm 0.10$ |
| | GLOBE-CE | $0.13 \pm 0.04$ | $-21.66 \pm 30.5$ | $1.4 \pm 0.11$ | $\mathbf{3.91 \pm 2.73}$ |
| | GLOBALGLANCE | $0.00 \pm 0.00$ | $-15.95 \pm 23.40$ | $1.70 \pm 0.14$ | $10.30 \pm 0.51$ |
| | $\text{OUR}_{global}$ | $\mathbf{0.46 \pm 0.02}$ | $\mathbf{29.93 \pm 0.61}$ | $\mathbf{1.14 \pm 0.01}$ | $10.09 \pm 4.47$ |
| LAW | GLOBE-CE | $0.4 \pm 0.04$ | $0.10 \pm 0.17$ | $1.14 \pm 0.01$ | $\mathbf{2.00 \pm 1.45}$ |
| | GLOBALGLANCE | $0.25 \pm 0.13$ | $-1.39 \pm 1.37$ | $1.32 \pm 0.07$ | $6.05 \pm 0.40$ |
| | $\text{OUR}_{global}$ | $\mathbf{0.82 \pm 0.01}$ | $\mathbf{1.57 \pm 0.12}$ | $\mathbf{1.07 \pm 0.01}$ | $6.70 \pm 2.07$ |
| MOONS | GLOBE-CE | $0.05 \pm 0.1$ | $-0.67 \pm 0.34$ | $\mathbf{1.32 \pm 0.03}$ | $\mathbf{2.84 \pm 1.54}$ |
| | GLOBALGLANCE | $0.25 \pm 0.08$ | $-17.44 \pm 12.46$ | $1.92 \pm 0.08$ | $6.53 \pm 0.15$ |
| | $\text{OUR}_{global}$ | $\mathbf{0.59 \pm 0.21}$ | $\mathbf{0.89 \pm 0.14}$ | $1.17 \pm 0.03$ | $6.93 \pm 2.08$ |
| WINE | GLOBE-CE | $0.06 \pm 0.05$ | $-15.72 \pm 16.9$ | $1.63 \pm 0.24$ | $\mathbf{8.14 \pm 10.54}$ |
| | GLOBALGLANCE | $0.00 \pm 0.00$ | $-249.34 \pm 343.47$ | $1.17 \pm 0.05$ | $9.53 \pm 0.65$ |
| | $\text{OUR}_{global}$ | $\mathbf{0.95 \pm 0.11}$ | $\mathbf{7.75 \pm 0.68}$ | $\mathbf{1.11 \pm 0.05}$ | $22.36 \pm 4.95$ |

Table 14: Comparative analysis of our method in **group-wise configuration** with other CF methods across various datasets and classification models. Values are averaged over five cross-validation folds.

| DATASET | METHOD | GROUPS | COVERAGE↑ | VALID.↑ | L2↓ | ISOFOREST↑ | TIME(S)↓ |
|---|---|---|---|---|---|---|---|
| | | | | MLP | | | |
| BLOBS | EA | 3.60 ± 1.67 | **1.00 ± 0.00** | **1.00 ± 0.00** | 1.00 ± 0.00 | −0.16 ± 0.00 | 95.38 ± 40.81 |
| | GLANCE | 2.00 ± 0.00 | **1.00 ± 0.00** | 0.96 ± 0.03 | 0.56 ± 0.02 | −0.10 ± 0.01 | 49.07 ± 3.9 |
| | TCREx | 2.40 ± 0.55 | **1.00 ± 0.00** | **1.00 ± 0.00** | 0.00 ± 0.00 | 0.02 ± 0.00 | **0.00 ± 0.00** |
| | OUR$_{group}$ | 1.60 ± 0.49 | **1.00 ± 0.00** | **1.00 ± 0.00** | **0.46 ± 0.01** | **0.03 ± 0.00** | 14.55 ± 2.51 |
| DIGITS | EA | 4.00 ± 0.00 | 0.00 ± 0.00 | − | − | − | 972.35 ± 62.15 |
| | GLANCE | 4.00 ± 0.00 | 1.00 ± 0.00 | 1.00 ± 0.00 | 2.01 ± 0.18 | −0.08 ± 0.01 | 761.25 ± 75.97 |
| | TCREx | 91.00 ± 50.76 | 1.00 ± 0.00 | 1.00 ± 0.00 | **0.15 ± 0.06** | 0.09 ± 0.00 | **13.37 ± 5.26** |
| | OUR$_{group}$ | 2.80 ± 1.83 | 1.00 ± 0.00 | 1.00 ± 0.00 | 16.35 ± 1.25 | **0.10 ± 0.00** | 102.23 ± 13.14 |
| HELOC | EA | 4.60 ± 1.14 | **1.00 ± 0.00** | **1.00 ± 0.00** | 1.90 ± 0.09 | −0.02 ± 0.03 | 338.84 ± 43.44 |
| | GLANCE | 10.00 ± 0.00 | **1.00 ± 0.00** | 0.95 ± 0.01 | 1.00 ± 0.07 | −0.01 ± 0.01 | 116.31 ± 16.93 |
| | TCREx | 26.80 ± 21.02 | **1.00 ± 0.00** | 0.94 ± 0.07 | **0.07 ± 0.05** | 0.05 ± 0.00 | **0.13 ± 0.07** |
| | OUR$_{group}$ | 16.80 ± 2.56 | **1.00 ± 0.00** | **1.00 ± 0.00** | 0.48 ± 0.06 | 0.02 ± 0.01 | 169.58 ± 24.21 |
| LAW | EA | 4.40 ± 1.95 | **1.00 ± 0.00** | **1.00 ± 0.00** | 1.13 ± 0.07 | −0.12 ± 0.01 | 121.26 ± 44.08 |
| | GLANCE | 2.00 ± 0.00 | **1.00 ± 0.00** | 0.95 ± 0.03 | 0.53 ± 0.05 | −0.05 ± 0.02 | 96.32 ± 15.61 |
| | TCREx | 5.00 ± 2.00 | **1.00 ± 0.00** | 0.79 ± 0.29 | **0.11 ± 0.09** | 0.03 ± 0.00 | **0.00 ± 0.00** |
| | OUR$_{group}$ | 4.40 ± 1.36 | **1.00 ± 0.00** | **1.00 ± 0.00** | 0.36 ± 0.02 | **0.04 ± 0.01** | 77.31 ± 60.42 |
| MOONS | EA | 5.20 ± 2.05 | **1.00 ± 0.00** | **1.00 ± 0.00** | 1.03 ± 0.00 | −0.14 ± 0.01 | 131.36 ± 50.25 |
| | GLANCE | 3.00 ± 0.00 | **1.00 ± 0.00** | 0.84 ± 0.14 | 0.53 ± 0.03 | −0.02 ± 0.02 | 91.44 ± 6.34 |
| | TCREx | 6.00 ± 0.00 | **1.00 ± 0.00** | 0.83 ± 0.15 | **0.10 ± 0.05** | 0.00 ± 0.01 | **0.00 ± 0.00** |
| | OUR$_{group}$ | 10.80 ± 0.98 | **1.00 ± 0.00** | **1.00 ± 0.00** | 0.46 ± 0.04 | **0.02 ± 0.00** | 42.47 ± 25.88 |
| WINE | EA | 1.00 ± 0.00 | **1.00 ± 0.00** | **1.00 ± 0.00** | 1.39 ± 0.26 | −0.03 ± 0.03 | 16.66 ± 0.50 |
| | GLANCE | 2.00 ± 0.00 | **1.00 ± 0.00** | 0.84 ± 0.10 | 0.70 ± 0.09 | 0.05 ± 0.01 | 7.2 ± 3.48 |
| | TCREx | 15.40 ± 11.28 | **1.00 ± 0.00** | **1.00 ± 0.00** | **0.09 ± 0.15** | 0.05 ± 0.01 | **0.00 ± 0.00** |
| | OUR$_{group}$ | 1.00 ± 0.00 | **1.00 ± 0.00** | **1.00 ± 0.00** | 0.81 ± 0.07 | **0.07 ± 0.01** | 32.41 ± 23.19 |
| | | | | LR | | | |
| BLOBS | EA | 3.60 ± 1.67 | **1.00 ± 0.00** | **1.00 ± 0.00** | 1.00 ± 0.00 | −0.16 ± 0.00 | 90.42 ± 39.69 |
| | GLANCE | 2.00 ± 0.00 | **1.00 ± 0.00** | 0.94 ± 0.04 | 0.55 ± 0.03 | −0.07 ± 0.03 | 37.93 ± 8.77 |
| | TCREx | 2.40 ± 0.55 | **1.00 ± 0.00** | **1.00 ± 0.00** | **0.00 ± 0.00** | 0.02 ± 0.00 | **0.00 ± 0.00** |
| | OUR$_{group}$ | 1.60 ± 0.49 | **1.00 ± 0.00** | **1.00 ± 0.00** | 0.46 ± 0.01 | **0.03 ± 0.00** | 14.55 ± 2.51 |
| DIGITS | EA | 4.00 ± 0.00 | 0.00 ± 0.00 | − | − | − | 895.26 ± 46.34 |
| | GLANCE | 4.00 ± 0.00 | 1.00 ± 0.00 | 0.66 ± 0.11 | 1.69 ± 0.17 | −0.06 ± 0.01 | 605.66 ± 58.69 |
| | TCREx | 101.00 ± 38.21 | 1.00 ± 0.00 | 1.00 ± 0.00 | **0.10 ± 0.09** | 0.09 ± 0.00 | **14.46 ± 6.04** |
| | OUR$_{group}$ | 2.80 ± 1.83 | 1.00 ± 0.00 | 1.00 ± 0.00 | 16.35 ± 1.25 | **0.10 ± 0.00** | 102.23 ± 13.14 |
| HELOC | EA | 5.00 ± 1.58 | 0.98 ± 0.05 | **1.00 ± 0.00** | 1.64 ± 0.15 | 0.01 ± 0.01 | 240.68 ± 34.91 |
| | GLANCE | 10.00 ± 0.00 | **1.00 ± 0.00** | 0.95 ± 0.03 | 0.89 ± 0.11 | 0.00 ± 0.01 | 81.45 ± 9.69 |
| | TCREx | 21.00 ± 3.94 | **1.00 ± 0.00** | **1.00 ± 0.01** | **0.05 ± 0.03** | 0.05 ± 0.00 | **0.11 ± 0.03** |
| | OUR$_{group}$ | 16.80 ± 2.56 | **1.00 ± 0.00** | **1.00 ± 0.00** | 0.48 ± 0.06 | 0.02 ± 0.01 | 169.58 ± 24.21 |
| LAW | EA | 4.6 ± 1.82 | 0.95 ± 0.01 | **1.00 ± 0.00** | 1.06 ± 0.01 | −0.11 ± 0.01 | 127.38 ± 21.35 |
| | GLANCE | 2.00 ± 0.00 | **1.00 ± 0.00** | 0.97 ± 0.03 | 0.53 ± 0.03 | −0.06 ± 0.01 | 95.86 ± 17.19 |
| | TCREx | 6.00 ± 1.22 | **1.00 ± 0.00** | 0.37 ± 0.29 | **0.25 ± 0.08** | 0.01 ± 0.02 | **0.00 ± 0.00** |
| | OUR$_{group}$ | 4.40 ± 1.36 | **1.00 ± 0.00** | **1.00 ± 0.00** | 0.36 ± 0.02 | **0.04 ± 0.01** | 77.31 ± 60.42 |
| MOONS | EA | 3.50 ± 0.71 | **1.00 ± 0.00** | 0.79 ± 0.13 | 1.05 ± 0.03 | −0.13 ± 0.00 | 95.86 ± 17.19 |
| | GLANCE | 3.00 ± 0.00 | **1.00 ± 0.00** | 0.97 ± 0.04 | 0.58 ± 0.02 | −0.04 ± 0.03 | 61.51 ± 9.72 |
| | TCREx | 7.00 ± 1.87 | **1.00 ± 0.00** | 0.91 ± 0.10 | **0.11 ± 0.09** | −0.01 ± 0.02 | **0.00 ± 0.00** |
| | OUR$_{group}$ | 10.80 ± 0.98 | **1.00 ± 0.00** | **1.00 ± 0.00** | 0.46 ± 0.04 | **0.02 ± 0.00** | 42.47 ± 25.88 |
| WINE | EA | 1.00 ± 0.00 | **1.00 ± 0.00** | **1.00 ± 0.00** | 1.6 ± 0.17 | −0.06 ± 0.03 | 17.59 ± 1.74 |
| | GLANCE | 2.00 ± 0.00 | **1.00 ± 0.00** | 0.85 ± 0.12 | 0.62 ± 0.09 | 0.05 ± 0.01 | 4.29 ± 2.31 |
| | TCREx | 17.40 ± 10.67 | **1.00 ± 0.00** | **1.00 ± 0.00** | **0.20 ± 0.10** | 0.05 ± 0.01 | **0.00 ± 0.00** |
| | OUR$_{group}$ | 1.00 ± 0.00 | **1.00 ± 0.00** | **1.00 ± 0.00** | 0.81 ± 0.07 | **0.07 ± 0.01** | 32.41 ± 23.19 |

Table 15: Additional comparative plausibility and cost analysis of our method in **group-wise configuration** with other CF methods across various datasets and classification models. Values are averaged over five cross-validation folds.

| DATASET | METHOD | GROUPS | PROB. PLAUS.↑ | LOG DENS.↑ | LOF↓ | COST↓ |
|---|---|---|---|---|---|---|
| | | | MLP | | | |
| BLOBS | EA | $3.60 \pm 1.67$ | $0.00 \pm 0.00$ | $-194.1 \pm 109.3$ | $10.96 \pm 0.20$ | $10.25 \pm 0.75$ |
| | GLANCE | $2.00 \pm 0.00$ | $0.02 \pm 0.03$ | $-7.16 \pm 1.92$ | $2.53 \pm 0.36$ | $5.94 \pm 0.40$ |
| | TCREX | $2.40 \pm 0.55$ | $0.00 \pm 0.00$ | $-44.51 \pm 27.94$ | $1.10 \pm 0.02$ | $\mathbf{0.00 \pm 0.00}$ |
| | $\mathrm{OUR}_{group}$ | $1.60 \pm 0.49$ | $\mathbf{0.78 \pm 0.05}$ | $\mathbf{2.98 \pm 0.06}$ | $1.04 \pm 0.01$ | $6.50 \pm 0.17$ |
| DIGITS | EA | $4.00 \pm 0.00$ | $0.00 \pm 0.00$ | $-$ | $-$ | $-$ |
| | GLANCE | $4.00 \pm 0.00$ | $0.00 \pm 0.00$ | $-360 \pm 49$ | $1.64 \pm 0.06$ | $30.44 \pm 5.24$ |
| | TCREX | $91.00 \pm 50.76$ | $0.00 \pm 0.00$ | $-359.28 \pm 8.52$ | $\mathbf{1.08 \pm 0.00}$ | $\mathbf{0.00 \pm 0.00}$ |
| | $\mathrm{OUR}_{group}$ | $2.80 \pm 1.83$ | $\mathbf{0.36 \pm 0.08}$ | $\mathbf{-100.96 \pm 0.97}$ | $1.12 \pm 0.02$ | $48.92 \pm 2.71$ |
| HELOC | EA | $4.60 \pm 1.14$ | $0.00 \pm 0.00$ | $-1631 \pm 2694$ | $3.48 \pm 0.41$ | $55.68 \pm 13.55$ |
| | GLANCE | $10.00 \pm 0.00$ | $0.00 \pm 0.00$ | $-83.00 \pm 52.99$ | $1.97 \pm 0.08$ | $13.52 \pm 2.28$ |
| | TCREX | $26.80 \pm 21.02$ | $0.03 \pm 0.04$ | $-15.54 \pm 23.72$ | $\mathbf{1.11 \pm 0.02}$ | $\mathbf{0.85 \pm 1.01}$ |
| | $\mathrm{OUR}_{group}$ | $16.80 \pm 2.56$ | $\mathbf{0.07 \pm 0.01}$ | $\mathbf{11.72 \pm 3.02}$ | $1.47 \pm 0.05$ | $7.35 \pm 1.19$ |
| LAW | EA | $4.40 \pm 1.95$ | $0.00 \pm 0.00$ | $-748 \pm 884$ | $4.19 \pm 0.20$ | $13.33 \pm 3.42$ |
| | GLANCE | $2.00 \pm 0.00$ | $0.22 \pm 0.14$ | $-2.58 \pm 2.25$ | $1.36 \pm 0.16$ | $5.65 \pm 0.52$ |
| | TCREX | $5.00 \pm 2.00$ | $0.44 \pm 0.25$ | $-2.85 \pm 1.35$ | $\mathbf{1.05 \pm 0.01}$ | $\mathbf{0.62 \pm 0.95}$ |
| | $\mathrm{OUR}_{group}$ | $4.40 \pm 1.36$ | $\mathbf{0.74 \pm 0.03}$ | $\mathbf{2.10 \pm 0.03}$ | $1.05 \pm 0.01$ | $5.88 \pm 0.39$ |
| MOONS | EA | $5.20 \pm 2.05$ | $0.00 \pm 0.00$ | $-1250 \pm 1896$ | $6.17 \pm 0.36$ | $11.89 \pm 3.38$ |
| | GLANCE | $3.00 \pm 0.00$ | $0.27 \pm 0.08$ | $-9.02 \pm 10.34$ | $1.46 \pm 0.29$ | $5.39 \pm 2.05$ |
| | TCREX | $6.00 \pm 0.00$ | $0.27 \pm 0.15$ | $-8.29 \pm 7.90$ | $1.28 \pm 0.05$ | $\mathbf{1.38 \pm 1.41}$ |
| | $\mathrm{OUR}_{group}$ | $10.80 \pm 0.98$ | $\mathbf{0.92 \pm 0.06}$ | $\mathbf{1.76 \pm 0.05}$ | $1.01 \pm 0.01$ | $6.00 \pm 0.62$ |
| WINE | EA | $1.00 \pm 0.00$ | $0.00 \pm 0.00$ | $-48.89 \pm 21.95$ | $2.38 \pm 0.44$ | $20.00 \pm 6.49$ |
| | GLANCE | $2.00 \pm 0.00$ | $0.09 \pm 0.09$ | $-2.63 \pm 5.21$ | $1.16 \pm 0.03$ | $9.46 \pm 1.39$ |
| | TCREX | $15.40 \pm 11.28$ | $0.00 \pm 0.00$ | $-372.30 \pm 669.40$ | $1.10 \pm 0.08$ | $\mathbf{0.07 \pm 0.27}$ |
| | $\mathrm{OUR}_{group}$ | $1.00 \pm 0.00$ | $\mathbf{0.72 \pm 0.18}$ | $\mathbf{8.67 \pm 0.51}$ | $1.06 \pm 0.02$ | $24.05 \pm 1.92$ |
| | | | LR | | | |
| BLOBS | EA | $3.60 \pm 1.67$ | $0.00 \pm 0.00$ | $-141 \pm 28$ | $10.97 \pm 0.21$ | $10.25 \pm 0.75$ |
| | GLANCE | $2.00 \pm 0.00$ | $0.12 \pm 0.13$ | $-2.55 \pm 2.29$ | $1.89 \pm 0.44$ | $6.04 \pm 1.15$ |
| | TCREX | $2.40 \pm 0.55$ | $0.00 \pm 0.00$ | $-45.59 \pm 16.68$ | $1.10 \pm 0.02$ | $\mathbf{0.00 \pm 0.00}$ |
| | $\mathrm{OUR}_{group}$ | $1.60 \pm 0.49$ | $\mathbf{0.78 \pm 0.05}$ | $\mathbf{2.98 \pm 0.06}$ | $1.04 \pm 0.01$ | $6.50 \pm 0.17$ |
| DIGITS | EA | $4.00 \pm 0.00$ | $0.00 \pm 0.00$ | $-$ | $-$ | $-$ |
| | GLANCE | $4.00 \pm 0.00$ | $0.01 \pm 0.01$ | $-485 \pm 42$ | $1.54 \pm 0.10$ | $27.83 \pm 6.16$ |
| | TCREX | $101.00 \pm 38.21$ | $0.00 \pm 0.00$ | $-353.45 \pm 86.64$ | $\mathbf{1.08 \pm 0.00}$ | $\mathbf{0.00 \pm 0.00}$ |
| | $\mathrm{OUR}_{group}$ | $2.80 \pm 1.83$ | $\mathbf{0.36 \pm 0.08}$ | $\mathbf{-100.96 \pm 0.97}$ | $1.12 \pm 0.02$ | $48.92 \pm 2.71$ |
| HELOC | EA | $5.00 \pm 1.58$ | $0.00 \pm 0.00$ | $-2170 \pm 3061$ | $3.09 \pm 0.68$ | $46.13 \pm 9.94$ |
| | GLANCE | $10.00 \pm 0.00$ | $0.00 \pm 0.00$ | $-107 \pm 141$ | $1.98 \pm 0.17$ | $10.66 \pm 0.77$ |
| | TCREX | $21.00 \pm 3.94$ | $0.03 \pm 0.04$ | $-30.15 \pm 44.41$ | $\mathbf{1.10 \pm 0.01}$ | $0.60 \pm 1.00$ |
| | $\mathrm{OUR}_{group}$ | $16.80 \pm 2.56$ | $\mathbf{0.07 \pm 0.01}$ | $\mathbf{11.72 \pm 3.02}$ | $1.47 \pm 0.05$ | $\mathbf{7.35 \pm 1.19}$ |
| LAW | EA | $4.6 \pm 1.82$ | $0.00 \pm 0.00$ | $-63.32 \pm 21.79$ | $4.08 \pm 0.16$ | $13.04 \pm 2.88$ |
| | GLANCE | $2.00 \pm 0.00$ | $0.18 \pm 0.10$ | $-2.56 \pm 1.03$ | $1.40 \pm 0.11$ | $5.84 \pm 0.72$ |
| | TCREX | $6.00 \pm 1.22$ | $0.61 \pm 0.12$ | $0.02 \pm 1.80$ | $\mathbf{1.06 \pm 0.03}$ | $\mathbf{0.67 \pm 1.02}$ |
| | $\mathrm{OUR}_{group}$ | $4.40 \pm 1.36$ | $\mathbf{0.74 \pm 0.03}$ | $\mathbf{2.10 \pm 0.03}$ | $1.05 \pm 0.01$ | $5.88 \pm 0.39$ |
| MOONS | EA | $3.50 \pm 0.71$ | $0.00 \pm 0.00$ | $-92.74 \pm 101$ | $6.37 \pm 0.02$ | $11.74 \pm 2.86$ |
| | GLANCE | $3.00 \pm 0.00$ | $0.29 \pm 0.11$ | $-153 \pm 329$ | $1.77 \pm 0.50$ | $6.77 \pm 1.12$ |
| | TCREX | $7.00 \pm 1.87$ | $0.10 \pm 0.13$ | $-236.58 \pm 237.20$ | $1.14 \pm 0.06$ | $\mathbf{1.40 \pm 1.80}$ |
| | $\mathrm{OUR}_{group}$ | $10.80 \pm 0.98$ | $\mathbf{0.92 \pm 0.06}$ | $\mathbf{1.76 \pm 0.05}$ | $1.01 \pm 0.01$ | $6.00 \pm 0.62$ |
| WINE | EA | $1.00 \pm 0.00$ | $0.00 \pm 0.00$ | $-66.5 \pm 47.9$ | $2.76 \pm 0.38$ | $26.03 \pm 4.93$ |
| | GLANCE | $2.00 \pm 0.00$ | $0.02 \pm 0.04$ | $-3.98 \pm 2.84$ | $1.14 \pm 0.05$ | $9.63 \pm 1.61$ |
| | TCREX | $17.40 \pm 10.67$ | $0.00 \pm 0.00$ | $-629.24 \pm 648.00$ | $1.11 \pm 0.08$ | $\mathbf{1.05 \pm 1.32}$ |
| | $\mathrm{OUR}_{group}$ | $1.00 \pm 0.00$ | $\mathbf{0.72 \pm 0.18}$ | $\mathbf{8.67 \pm 0.51}$ | $1.06 \pm 0.02$ | $24.05 \pm 1.92$ |

Table 16: Comparative analysis of our method in **local configuration** with other local CF methods across various datasets and classification models. Values are averaged over five cross-validation folds.

| DATASET | METHOD | COVERAGE↑ | VALID.↑ | L2↓ | IsoForest↑ | TIME(S)↓ |
|---|---|---|---|---|---|---|
| | | | MLP | | | |
| BLOBS | DiCE | $1.00 \pm 0.00$ | $1.00 \pm 0.00$ | $0.51 \pm 0.03$ | $-0.1 \pm 0.00$ | $8.15 \pm 5.24$ |
| | Wach | $0.99 \pm 0.03$ | $1.00 \pm 0.00$ | $\mathbf{0.23 \pm 0.01}$ | $-0.04 \pm 0.00$ | $0.22 \pm 0.05$ |
| | CCHVAE | $1.00 \pm 0.00$ | $1.00 \pm 0.00$ | $0.37 \pm 0.05$ | $-0.06 \pm 0.01$ | $\mathbf{2.15 \pm 0.62}$ |
| | PPCEF | $1.00 \pm 0.00$ | $1.00 \pm 0.00$ | $0.47 \pm 0.01$ | $\mathbf{0.04 \pm 0.00}$ | $19.55 \pm 0.30$ |
| | OUR$_{local}$ | $1.00 \pm 0.00$ | $1.00 \pm 0.00$ | $0.39 \pm 0.01$ | $0.03 \pm 0.00$ | $6.20 \pm 0.20$ |
| DIGITS | DiCE | $1.00 \pm 0.00$ | $1.00 \pm 0.00$ | $23.77 \pm 0.99$ | $0.03 \pm 0.01$ | $162.88 \pm 15.52$ |
| | Wach | $1.00 \pm 0.00$ | $1.00 \pm 0.00$ | $\mathbf{2.10 \pm 0.44}$ | $0.09 \pm 0.00$ | $16.41 \pm 0.62$ |
| | CCHVAE | $1.00 \pm 0.00$ | $1.00 \pm 0.00$ | $2.19 \pm 0.24$ | $0.04 \pm 0.01$ | $\mathbf{3.38 \pm 0.52}$ |
| | PPCEF | $1.00 \pm 0.00$ | $1.00 \pm 0.00$ | $11.42 \pm 0.05$ | $0.10 \pm 0.01$ | $25.09 \pm 0.40$ |
| | OUR$local$ | $1.00 \pm 0.00$ | $1.00 \pm 0.00$ | $11.41 \pm 0.51$ | $0.11 \pm 0.00$ | $18.58 \pm 0.68$ |
| HELOC | DiCE | $1.00 \pm 0.00$ | $1.00 \pm 0.00$ | $1.00 \pm 0.06$ | $-0.01 \pm 0.00$ | $230.85 \pm 26.00$ |
| | Wach | $1.00 \pm 0.00$ | $1.00 \pm 0.00$ | $\mathbf{0.16 \pm 0.02}$ | $0.06 \pm 0.00$ | $33.88 \pm 4.98$ |
| | CCHVAE | $1.00 \pm 0.00$ | $1.00 \pm 0.00$ | $0.59 \pm 0.02$ | $\mathbf{0.11 \pm 0.00}$ | $\mathbf{14.60 \pm 3.83}$ |
| | PPCEF | $1.00 \pm 0.00$ | $0.98 \pm 0.02$ | $0.42 \pm 0.02$ | $0.07 \pm 0.01$ | $24.31 \pm 4.52$ |
| | OUR$local$ | $1.00 \pm 0.00$ | $1.00 \pm 0.00$ | $0.47 \pm 0.01$ | $0.08 \pm 0.00$ | $20.21 \pm 2.02$ |
| LAW | DiCE | $1.00 \pm 0.00$ | $1.00 \pm 0.00$ | $0.52 \pm 0.01$ | $-0.05 \pm 0.00$ | $43.82 \pm 9.62$ |
| | Wach | $0.97 \pm 0.05$ | $1.00 \pm 0.01$ | $\mathbf{0.16 \pm 0.01}$ | $0.05 \pm 0.00$ | $21.66 \pm 3.91$ |
| | CCHVAE | $1.00 \pm 0.00$ | $1.00 \pm 0.00$ | $0.31 \pm 0.01$ | $\mathbf{0.09 \pm 0.01}$ | $\mathbf{0.28 \pm 0.17}$ |
| | PPCEF | $1.00 \pm 0.00$ | $0.95 \pm 0.01$ | $0.32 \pm 0.02$ | $0.06 \pm 0.00$ | $20.63 \pm 1.08$ |
| | OUR$_{local}$ | $1.00 \pm 0.00$ | $1.00 \pm 0.00$ | $0.32 \pm 0.00$ | $0.05 \pm 0.00$ | $7.80 \pm 0.29$ |
| MOONS | DiCE | $1.00 \pm 0.00$ | $1.00 \pm 0.00$ | $0.55 \pm 0.01$ | $-0.04 \pm 0.01$ | $17.85 \pm 6.64$ |
| | Wach | $0.97 \pm 0.06$ | $1.00 \pm 0.00$ | $\mathbf{0.16 \pm 0.01}$ | $-0.00 \pm 0.00$ | $0.23 \pm 0.05$ |
| | CCHVAE | $1.00 \pm 0.00$ | $1.00 \pm 0.00$ | $0.28 \pm 0.01$ | $0.02 \pm 0.01$ | $\mathbf{0.10 \pm 0.04}$ |
| | PPCEF | $1.00 \pm 0.00$ | $0.98 \pm 0.01$ | $0.34 \pm 0.04$ | $\mathbf{0.03 \pm 0.01}$ | $20.44 \pm 1.75$ |
| | OUR$local$ | $1.00 \pm 0.00$ | $1.00 \pm 0.00$ | $0.30 \pm 0.01$ | $0.03 \pm 0.00$ | $7.32 \pm 0.22$ |
| WINE | DiCE | $1.00 \pm 0.00$ | $1.00 \pm 0.00$ | $0.72 \pm 0.08$ | $0.03 \pm 0.01$ | $0.70 \pm 0.05$ |
| | Wach | $1.00 \pm 0.00$ | $1.00 \pm 0.00$ | $\mathbf{0.43 \pm 0.08}$ | $0.03 \pm 0.02$ | $0.10 \pm 0.02$ |
| | CCHVAE | $1.00 \pm 0.00$ | $1.00 \pm 0.00$ | $0.79 \pm 0.05$ | $\mathbf{0.09 \pm 0.00}$ | $\mathbf{0.02 \pm 0.00}$ |
| | PPCEF | $1.00 \pm 0.00$ | $1.00 \pm 0.00$ | $0.66 \pm 0.05$ | $0.07 \pm 0.01$ | $12.41 \pm 0.52$ |
| | OUR$local$ | $1.00 \pm 0.00$ | $1.00 \pm 0.00$ | $0.69 \pm 0.07$ | $0.05 \pm 0.01$ | $5.49 \pm 0.32$ |
| | | | LR | | | |
| BLOBS | Artelt | $1.00 \pm 0.00$ | $1.00 \pm 0.00$ | $0.33 \pm 0.02$ | $-0.06 \pm 0.00$ | $3.42 \pm 0.90$ |
| | DiCE | $1.00 \pm 0.00$ | $1.00 \pm 0.00$ | $0.49 \pm 0.02$ | $-0.1 \pm 0.01$ | $12.65 \pm 3.59$ |
| | Wach | $0.99 \pm 0.01$ | $1.00 \pm 0.00$ | $\mathbf{0.32 \pm 0.05}$ | $-0.01 \pm 0.02$ | $\mathbf{0.34 \pm 0.02}$ |
| | CCHVAE | $1.00 \pm 0.00$ | $1.00 \pm 0.00$ | $0.33 \pm 0.03$ | $-0.05 \pm 0.01$ | $0.94 \pm 0.33$ |
| | PPCEF | $1.00 \pm 0.00$ | $1.00 \pm 0.00$ | $0.50 \pm 0.04$ | $\mathbf{0.04 \pm 0.01}$ | $3.22 \pm 0.84$ |
| | OUR$_{local}$ | $1.00 \pm 0.00$ | $1.00 \pm 0.00$ | $0.45 \pm 0.04$ | $0.04 \pm 0.01$ | $6.56 \pm 0.24$ |
| DIGITS | Artelt | $1.00 \pm 0.00$ | $1.00 \pm 0.00$ | $19.56 \pm 1.55$ | $0.07 \pm 0.01$ | $27.08 \pm 1.16$ |
| | DiCE | $1.00 \pm 0.00$ | $1.00 \pm 0.00$ | $22.2 \pm 0.71$ | $0.04 \pm 0.01$ | $138.12 \pm 12.88$ |
| | Wach | $1.00 \pm 0.00$ | $1.00 \pm 0.00$ | $2.46 \pm 0.32$ | $0.10 \pm 0.00$ | $9.68 \pm 0.08$ |
| | CCHVAE | $1.00 \pm 0.00$ | $1.00 \pm 0.00$ | $\mathbf{2.07 \pm 0.14}$ | $0.04 \pm 0.01$ | $\mathbf{2.61 \pm 0.45}$ |
| | PPCEF | $1.00 \pm 0.00$ | $1.00 \pm 0.00$ | $10.33 \pm 0.04$ | $0.09 \pm 0.01$ | $8.68 \pm 3.65$ |
| | OUR$local$ | $1.00 \pm 0.00$ | $1.00 \pm 0.00$ | $10.55 \pm 0.48$ | $\mathbf{0.11 \pm 0.00}$ | $17.16 \pm 0.45$ |
| HELOC | DiCE | $1.00 \pm 0.00$ | $0.98 \pm 0.05$ | $0.88 \pm 0.07$ | $0.01 \pm 0.01$ | $175.64 \pm 26.01$ |
| | Wach | $1.00 \pm 0.00$ | $1.00 \pm 0.00$ | $\mathbf{0.15 \pm 0.02}$ | $0.06 \pm 0.00$ | $11.69 \pm 0.32$ |
| | CCHVAE | $1.00 \pm 0.00$ | $1.00 \pm 0.00$ | $0.56 \pm 0.01$ | $\mathbf{0.12 \pm 0.01}$ | $\mathbf{8.29 \pm 3.86}$ |
| | PPCEF | $1.00 \pm 0.00$ | $1.00 \pm 0.00$ | $0.23 \pm 0.01$ | $0.07 \pm 0.00$ | $12.44 \pm 2.36$ |
| | OUR$local$ | $1.00 \pm 0.00$ | $1.00 \pm 0.00$ | $0.44 \pm 0.02$ | $0.08 \pm 0.00$ | $19.36 \pm 3.58$ |
| LAW | Artelt | $1.00 \pm 0.00$ | $1.00 \pm 0.00$ | $0.20 \pm 0.01$ | $0.01 \pm 0.00$ | $11.71 \pm 2.34$ |
| | DiCE | $1.00 \pm 0.00$ | $0.96 \pm 0.09$ | $0.55 \pm 0.06$ | $-0.06 \pm 0.02$ | $43.05 \pm 7.67$ |
| | Wach | $1.00 \pm 0.00$ | $1.00 \pm 0.00$ | $\mathbf{0.19 \pm 0.03}$ | $0.04 \pm 0.00$ | $10.33 \pm 0.42$ |
| | CCHVAE | $1.00 \pm 0.00$ | $1.00 \pm 0.00$ | $0.32 \pm 0.01$ | $\mathbf{0.09 \pm 0.01}$ | $\mathbf{0.12 \pm 0.05}$ |
| | PPCEF | $1.00 \pm 0.00$ | $1.00 \pm 0.00$ | $0.23 \pm 0.01$ | $0.07 \pm 0.00$ | $2.42 \pm 0.10$ |
| | OUR$_{local}$ | $1.00 \pm 0.00$ | $1.00 \pm 0.00$ | $0.34 \pm 0.03$ | $0.04 \pm 0.01$ | $7.65 \pm 0.30$ |
| MOONS | Artelt | $1.00 \pm 0.00$ | $1.00 \pm 0.00$ | $0.29 \pm 0.01$ | $-0.02 \pm 0.01$ | $6.84 \pm 2.25$ |
| | DiCE | $1.00 \pm 0.00$ | $1.00 \pm 0.00$ | $0.62 \pm 0.04$ | $-0.07 \pm 0.01$ | $18.04 \pm 7.50$ |
| | Wach | $0.99 \pm 0.02$ | $1.00 \pm 0.00$ | $\mathbf{0.28 \pm 0.02}$ | $0.00 \pm 0.01$ | $7.50 \pm 6.43$ |
| | CCHVAE | $1.00 \pm 0.00$ | $1.00 \pm 0.00$ | $0.34 \pm 0.02$ | $\mathbf{0.03 \pm 0.01}$ | $\mathbf{0.37 \pm 0.08}$ |
| | PPCEF | $1.00 \pm 0.00$ | $1.00 \pm 0.00$ | $0.36 \pm 0.01$ | $0.03 \pm 0.01$ | $1.85 \pm 0.01$ |
| | OUR$local$ | $1.00 \pm 0.00$ | $1.00 \pm 0.00$ | $0.39 \pm 0.04$ | $0.03 \pm 0.00$ | $6.73 \pm 0.98$ |
| WINE | Artelt | $1.00 \pm 0.00$ | $1.00 \pm 0.00$ | $0.59 \pm 0.07$ | $0.05 \pm 0.01$ | $1.66 \pm 0.85$ |
| | DiCE | $1.00 \pm 0.00$ | $1.00 \pm 0.00$ | $0.78 \pm 0.07$ | $0.02 \pm 0.01$ | $1.18 \pm 1.16$ |
| | Wach | $1.00 \pm 0.00$ | $1.00 \pm 0.00$ | $\mathbf{0.41 \pm 0.07}$ | $0.05 \pm 0.02$ | $0.11 \pm 0.03$ |
| | CCHVAE | $1.00 \pm 0.00$ | $1.00 \pm 0.00$ | $0.81 \pm 0.06$ | $\mathbf{0.09 \pm 0.01}$ | $\mathbf{0.01 \pm 0.00}$ |
| | PPCEF | $1.00 \pm 0.00$ | $1.00 \pm 0.00$ | $0.53 \pm 0.04$ | $\mathbf{0.09 \pm 0.01}$ | $2.03 \pm 0.47$ |
| | OUR$local$ | $1.00 \pm 0.00$ | $1.00 \pm 0.00$ | $0.71 \pm 0.04$ | $0.05 \pm 0.00$ | $5.66 \pm 0.29$ |

Table 17: Additional comparative plausibility and cost analysis of our method in **local configuration** with other local CF methods across various datasets and classification models. Values are averaged over five cross-validation folds.

| Dataset | Method | Prob. Plaus.↑ | Log Dens.↑ | LOF↓ | Cost↓ |
|---|---|---|---|---|---|
| | | | MLP | | |
| Blobs | DiCE | $0.07 \pm 0.02$ | $-6.63 \pm 1.3$ | $2.91 \pm 0.22$ | $5.86 \pm 2.57$ |
| | Wach | $0.00 \pm 0.00$ | $-1.55 \pm 0.53$ | $1.62 \pm 0.10$ | $\mathbf{3.46 \pm 0.88}$ |
| | CCHVAE | $0.00 \pm 0.00$ | $-8.92 \pm 3.11$ | $2.78 \pm 0.28$ | $4.60 \pm 1.52$ |
| | PPCEF | $\mathbf{1.00 \pm 0.00}$ | $\mathbf{2.91 \pm 0.04}$ | $1.03 \pm 0.02$ | $-$ |
| | OUR$_{local}$ | $\mathbf{1.00 \pm 0.00}$ | $2.74 \pm 0.07$ | $1.04 \pm 0.01$ | $5.53 \pm 1.01$ |
| Digits | DiCE | $0.00 \pm 0.00$ | $-596.9 \pm 171.94$ | $1.88 \pm 0.03$ | $36.18 \pm 12.95$ |
| | Wach | $0.01 \pm 0.01$ | $-128.91 \pm 3.72$ | $1.29 \pm 0.02$ | $\mathbf{12.13 \pm 8.92}$ |
| | CCHVAE | $0.09 \pm 0.09$ | $\mathbf{-74.81 \pm 26.92}$ | $\mathbf{1.07 \pm 0.02}$ | $41.62 \pm 6.93$ |
| | PPCEF | $0.98 \pm 0.01$ | $-97.28 \pm 1.36$ | $1.13 \pm 0.01$ | $-$ |
| | OUR$_{local}$ | $\mathbf{1.00 \pm 0.00}$ | $-101.31 \pm 1.26$ | $1.23 \pm 0.02$ | $45.78 \pm 7.83$ |
| Heloc | DiCE | $0.00 \pm 0.00$ | $-35.19 \pm 11.26$ | $2.0 \pm 0.05$ | $15.41 \pm 8.85$ |
| | Wach | $0.24 \pm 0.02$ | $21.30 \pm 1.70$ | $1.13 \pm 0.01$ | $26.80 \pm 4.32$ |
| | CCHVAE | $0.74 \pm 0.23$ | $\mathbf{35.90 \pm 1.46}$ | $\mathbf{1.00 \pm 0.01}$ | $21.14 \pm 6.79$ |
| | PPCEF | $\mathbf{1.00 \pm 0.00}$ | $33.24 \pm 0.46$ | $1.08 \pm 0.01$ | $-$ |
| | OUR$_{local}$ | $\mathbf{1.00 \pm 0.00}$ | $33.36 \pm 0.33$ | $1.09 \pm 0.01$ | $\mathbf{10.75 \pm 4.96}$ |
| Law | DiCE | $0.3 \pm 0.01$ | $-0.8 \pm 0.28$ | $1.32 \pm 0.03$ | $6.45 \pm 2.54$ |
| | Wach | $0.57 \pm 0.08$ | $1.06 \pm 0.24$ | $1.05 \pm 0.00$ | $6.13 \pm 1.95$ |
| | CCHVAE | $\mathbf{1.00 \pm 0.00}$ | $\mathbf{2.65 \pm 0.14}$ | $1.02 \pm 0.02$ | $\mathbf{4.45 \pm 1.98}$ |
| | PPCEF | $\mathbf{1.00 \pm 0.00}$ | $2.04 \pm 0.02$ | $1.03 \pm 0.00$ | $-$ |
| | OUR$_{local}$ | $\mathbf{1.00 \pm 0.00}$ | $2.33 \pm 0.08$ | $1.03 \pm 0.00$ | $5.41 \pm 2.02$ |
| Moons | DiCE | $0.29 \pm 0.05$ | $-3.44 \pm 2.42$ | $1.67 \pm 0.1$ | $6.08 \pm 3.01$ |
| | Wach | $0.00 \pm 0.00$ | $-2.66 \pm 1.06$ | $1.58 \pm 0.06$ | $\mathbf{2.22 \pm 0.83}$ |
| | CCHVAE | $0.00 \pm 0.00$ | $-1.56 \pm 1.01$ | $1.41 \pm 0.13$ | $3.54 \pm 1.09$ |
| | PPCEF | $\mathbf{1.00 \pm 0.00}$ | $1.42 \pm 0.04$ | $\mathbf{1.00 \pm 0.02}$ | $-$ |
| | OUR$_{local}$ | $\mathbf{1.00 \pm 0.00}$ | $\mathbf{1.47 \pm 0.04}$ | $\mathbf{1.00 \pm 0.01}$ | $3.88 \pm 0.71$ |
| Wine | DiCE | $0.03 \pm 0.05$ | $-3.66 \pm 2.97$ | $1.46 \pm 0.09$ | $\mathbf{8.87 \pm 3.00}$ |
| | Wach | $0.09 \pm 0.11$ | $0.22 \pm 2.29$ | $1.36 \pm 0.11$ | $11.10 \pm 7.38$ |
| | CCHVAE | $0.08 \pm 0.17$ | $5.50 \pm 1.62$ | $1.03 \pm 0.02$ | $24.95 \pm 5.26$ |
| | PPCEF | $0.99 \pm 0.01$ | $\mathbf{7.79 \pm 0.59}$ | $\mathbf{1.01 \pm 0.01}$ | $-$ |
| | OUR$_{local}$ | $\mathbf{1.00 \pm 0.00}$ | $7.38 \pm 0.61$ | $1.18 \pm 0.05$ | $21.40 \pm 4.06$ |
| | | | LR | | |
| Blobs | Artelt | $0.00 \pm 0.00$ | $-4.67 \pm 1.29$ | $1.88 \pm 0.31$ | $4.83 \pm 1.37$ |
| | DiCE | $0.05 \pm 0.02$ | $-6.63 \pm 0.86$ | $2.85 \pm 0.11$ | $5.90 \pm 2.49$ |
| | Wach | $0.18 \pm 0.37$ | $1.00 \pm 1.17$ | $1.31 \pm 0.18$ | $\mathbf{4.04 \pm 0.83}$ |
| | CCHVAE | $0.00 \pm 0.00$ | $-6.05 \pm 1.28$ | $2.64 \pm 0.26$ | $4.41 \pm 1.40$ |
| | PPCEF | $\mathbf{1.00 \pm 0.00}$ | $3.00 \pm 0.11$ | $\mathbf{1.01 \pm 0.01}$ | $-$ |
| | OUR$_{local}$ | $\mathbf{1.00 \pm 0.00}$ | $\mathbf{3.01 \pm 0.06}$ | $1.03 \pm 0.01$ | $5.91 \pm 0.89$ |
| Digits | Artelt | $0.00 \pm 0.00$ | $-201.24 \pm 24.49$ | $1.71 \pm 0.12$ | $28.27 \pm 4.98$ |
| | DiCE | $0.00 \pm 0.00$ | $-411.41 \pm 135.26$ | $1.84 \pm 0.02$ | $32.97 \pm 12.83$ |
| | Wach | $0.07 \pm 0.07$ | $-117.81 \pm 1.99$ | $1.24 \pm 0.01$ | $9.85 \pm 3.39$ |
| | CCHVAE | $0.07 \pm 0.07$ | $\mathbf{-69.42 \pm 26.05}$ | $\mathbf{1.07 \pm 0.02}$ | $\mathbf{4.41 \pm 1.40}$ |
| | PPCEF | $\mathbf{1.00 \pm 0.00}$ | $-98.26 \pm 1.69$ | $1.12 \pm 0.01$ | $-$ |
| | OUR$_{local}$ | $\mathbf{1.00 \pm 0.00}$ | $-100.92 \pm 0.69$ | $1.2 \pm 0.00$ | $15.23 \pm 2.51$ |
| Heloc | DiCE | $0.01 \pm 0.01$ | $-43.76 \pm 17.02$ | $1.99 \pm 0.12$ | $11.06 \pm 4.30$ |
| | Wach | $0.20 \pm 0.03$ | $16.87 \pm 3.75$ | $1.14 \pm 0.01$ | $16.98 \pm 2.41$ |
| | CCHVAE | $0.92 \pm 0.08$ | $\mathbf{37.79 \pm 0.96}$ | $1.02 \pm 0.02$ | $22.12 \pm 7.29$ |
| | PPCEF | $\mathbf{1.00 \pm 0.00}$ | $32.34 \pm 0.56$ | $1.07 \pm 0.01$ | $-$ |
| | OUR$_{local}$ | $\mathbf{1.00 \pm 0.00}$ | $33.93 \pm 0.28$ | $1.08 \pm 0.01$ | $\mathbf{10.09 \pm 4.47}$ |
| Law | Artelt | $0.39 \pm 0.04$ | $0.02 \pm 0.17$ | $1.15 \pm 0.01$ | $6.73 \pm 2.33$ |
| | DiCE | $0.19 \pm 0.10$ | $-2.31 \pm 0.76$ | $1.42 \pm 0.07$ | $6.55 \pm 2.49$ |
| | Wach | $0.64 \pm 0.14$ | $1.35 \pm 0.48$ | $1.07 \pm 0.00$ | $6.53 \pm 1.43$ |
| | CCHVAE | $\mathbf{1.00 \pm 0.00}$ | $\mathbf{2.83 \pm 0.12}$ | $1.02 \pm 0.02$ | $\mathbf{4.51 \pm 2.08}$ |
| | PPCEF | $\mathbf{1.00 \pm 0.00}$ | $2.05 \pm 0.02$ | $1.03 \pm 0.00$ | $-$ |
| | OUR$_{local}$ | $\mathbf{1.00 \pm 0.00}$ | $2.18 \pm 0.09$ | $1.04 \pm 0.01$ | $5.81 \pm 1.90$ |
| Moons | Artelt | $0.05 \pm 0.11$ | $-0.74 \pm 0.42$ | $1.32 \pm 0.04$ | $6.04 \pm 1.92$ |
| | DiCE | $0.24 \pm 0.05$ | $-17.28 \pm 20.11$ | $2.04 \pm 0.24$ | $7.17 \pm 2.68$ |
| | Wach | $0.15 \pm 0.14$ | $-0.24 \pm 0.69$ | $1.28 \pm 0.05$ | $5.73 \pm 1.52$ |
| | CCHVAE | $0.00 \pm 0.00$ | $-1.61 \pm 1.06$ | $1.63 \pm 0.06$ | $\mathbf{4.32 \pm 1.65}$ |
| | PPCEF | $\mathbf{1.00 \pm 0.00}$ | $\mathbf{1.69 \pm 0.07}$ | $\mathbf{1.01 \pm 0.02}$ | $-$ |
| | OUR$_{local}$ | $0.88 \pm 0.27$ | $1.27 \pm 0.07$ | $1.08 \pm 0.06$ | $5.15 \pm 1.84$ |
| Wine | Artelt | $0.12 \pm 0.14$ | $-2.97 \pm 2.69$ | $1.45 \pm 0.14$ | $15.26 \pm 3.72$ |
| | DiCE | $0.03 \pm 0.05$ | $-3.63 \pm 2.67$ | $1.48 \pm 0.09$ | $\mathbf{9.53 \pm 3.35}$ |
| | Wach | $0.20 \pm 0.25$ | $2.30 \pm 2.55$ | $1.26 \pm 0.08$ | $10.94 \pm 2.89$ |
| | CCHVAE | $0.11 \pm 0.24$ | $4.66 \pm 2.44$ | $\mathbf{1.05 \pm 0.02}$ | $23.91 \pm 5.49$ |
| | PPCEF | $\mathbf{1.00 \pm 0.00}$ | $\mathbf{7.72 \pm 0.62}$ | $1.12 \pm 0.01$ | $-$ |
| | OUR$_{local}$ | $\mathbf{1.00 \pm 0.00}$ | $7.71 \pm 0.89$ | $1.21 \pm 0.07$ | $22.36 \pm 4.95$ |

Table 18: Friedman test results for statistical significance analysis across all configurations. All metrics show significant differences among methods ($p < 0.05$).

| Configuration | Metric | $p$-value |
|---|---|---|
| Global | Validity | $2.39 \times 10^{-5}$ |
| | L2 Distance | $0.0133$ |
| | Plausibility (IsoForest) | $3.94 \times 10^{-4}$ |
| | Time | $6.05 \times 10^{-13}$ |
| Group-wise | Coverage | $3.28 \times 10^{-8}$ |
| | Validity | $5.53 \times 10^{-11}$ |
| | L2 Distance | $2.65 \times 10^{-15}$ |
| | Plausibility (IsoForest) | $1.51 \times 10^{-11}$ |
| | Time | $4.21 \times 10^{-13}$ |
| Local | Coverage | $6.42 \times 10^{-9}$ |
| | Validity | $6.00 \times 10^{-14}$ |
| | L2 Distance | $2.28 \times 10^{-18}$ |
| | Plausibility (IsoForest) | $5.24 \times 10^{-15}$ |
| | Time | $2.96 \times 10^{-15}$ |

