# OpenReview forum: "Unifying Perspectives: Plausible Counterfactual Explanations on Global, Group-wise, and Local Levels"
_ICLR.cc/2026/Conference — ICLR 2026 Conference Withdrawn Submission_

### Official Review · Reviewer_K9zo · 2025-10-25

**Soundness:** 3
**Presentation:** 3
**Contribution:** 3
**Rating:** 4
**Confidence:** 5

**Summary:**

This paper presents a unified framework for generating counterfactual explanations at three different granularity levels: local (instance-specific), global (dataset-wide), and group-wise (for cohesive subgroups). Counterfactual explanations are a key tool in Explainable AI (XAI) that provide actionable "what-if" scenarios, showing how changes to input features can alter a model's predictions. The paper makes two  contributions to the field of counterfactual explanations. First, it introduces a unified approach that can generate counterfactual explanations at any desired granularity level through a single gradient-based optimization framework. Second, the paper attempts to develop group-wise counterfactual generation by developing an end-to-end optimization process that simultaneously performs instance grouping and counterfactual generation.

**Strengths:**

Despite the incremental nature of the novelty, the paper makes contributions that advance the practical state of the art in an important problem area. The combination of addressing real limitations in existing methods, providing comprehensive experimental validation, and demonstrating practical utility creates a package that is greater than the sum of its parts. For practitioners working with counterfactual explanations, this method would likely represent a genuine improvement over existing approaches, particularly for group-wise explanations where current methods are limited. The work is well-suited for practical advances built on engineering and thorough evaluation are valued, even when the fundamental algorithmic innovations are modest.

The paper's claim centers on joint optimization for group-wise counterfactuals, replacing the two-step clustering-then-generation approach. However, this contribution should be contextualized carefully. The idea of jointly optimizing clustering and another objective is not new in machine learning—it appears in various forms across deep clustering, mixture models, and representation learning literature. The specific application here involves learning a soft assignment matrix P through Sparsemax while simultaneously optimizing counterfactual directions in matrix D_GW. While this integration is sensible and addresses real limitations of prior two-step methods, it represents engineering ingenuity rather than algorithmic innovation. The optimization itself uses standard gradient descent on a carefully weighted multi-objective loss function—a common pattern in machine learning.

**Weaknesses:**

The paper's contributions, while practically useful, exhibit limited fundamental novelty when examined critically. The core claim of a "unified framework" is somewhat overstated—the unification is primarily achieved through parameterization of a standard gradient-based optimization formulation rather than introducing new theoretical insights. Setting K=N recovers local counterfactuals, K=1 recovers global counterfactuals, and intermediate values yield group-wise explanations. This is more of an observation about special cases of a general formulation than a novel algorithmic contribution. The mathematical framework itself relies entirely on well-established techniques: gradient-based optimization for differentiable models has been the standard approach in counterfactual explanation literature since Wachter et al. (2017).

The paper's weakness is its lack of fundamental algorithmic or theoretical innovation. As detailed in the novelty evaluation, the core technical components are borrowed from very recent work (2024) with straightforward extensions. The "unified framework" reduces to parameter selection in a standard gradient-based optimization formulation rather than introducing new theoretical insights. The plausibility integration using normalizing flows comes directly from Wielopolski et al. (2024), the validity loss from the same source, and the global counterfactual formulation extends GLOBE-CE (Ley et al., 2023). The joint optimization for group-wise counterfactuals, while addressing real limitations, applies well-known techniques (Sparsemax, entropy regularization, gradient descent) without innovation.

The paper provides no theoretical grounding or guarantees for its proposed method. There is no analysis of convergence properties, optimization landscape characteristics, sample complexity, approximation guarantees, or theoretical justification for why joint optimization should outperform two-step approaches beyond intuitive arguments. The choice of specific regularization terms (log-determinant for diversity, entropy for sparsity) lacks theoretical motivation for why these particular forms are optimal.

The paper makes several overstated claims that weaken its credibility. The abstract claims a "novel unified approach" when the unification is primarily through parameterization. The introduction states the method "eliminates the inefficient two-step process" as if unprecedented, but joint optimization is common in machine learning. The claim of "innovatively introducing" plausibility to group-wise settings understates how straightforward this extension is. The paper claims to generate explanations at "any desired granularity," but users must still manually set K (number of groups) or rely on automatic selection via regularization that may not match their needs. The "unified" framework still requires different hyperparameter configurations for different granularities, limiting true unification. More precise, measured language would better serve the paper's contributions.

**Questions:**

The paper claims joint optimization eliminates inefficiencies of two-step methods, but provides no theoretical analysis. Specific sub-questions include: (a) Does the non-convex optimization landscape with coupled variables (B, D_GW, K) have local minima that trap the method in poor solutions? (b) What convergence guarantees exist for the alternating gradient descent on these coupled parameters? (c) Under what data distributions or model characteristics does joint optimization provably improve upon sequential clustering-then-generation?

How should practitioners set the five loss weights (λ, λ_p, λ_s, λ_k, λ_d) spanning five orders of magnitude for new datasets and domains without exhaustive grid search?

(a) Do human users (domain experts, affected individuals) actually find these groups more interpretable, coherent, and actionable than alternatives? (b) What metrics beyond group count quantify interpretability—within-group homogeneity, between-group separation, semantic coherence, alignment with known subpopulations?

**Details Of Ethics Concerns:**

No ethics concerns have been identified.

---

> ### Author Response · Authors · 2025-11-21
>
> We thank the reviewer for the detailed assessment and for recognizing our work as a **"genuine improvement"** that creates a **"package greater than the sum of its parts."** We value your acknowledgement of the practical utility and comprehensive evaluation of our method. Below, we address the concerns regarding novelty, theoretical grounding, and experimental details.
>
> ## Novelty (W1)
>
> We respectfully disagree with the characterization that our contributions are "primarily parameterization" without innovation. Our key novelty lies in the **integration and end-to-end optimization**, not in inventing individual components from scratch:
>
> No prior work jointly optimizes instance grouping, counterfactual generation, and plausibility constraints for group-wise counterfactuals. This specific integration addresses fundamental limitations in the field:
>
> **Beyond Parameterization:** While setting $K=N$ or $K=1$ recovers local/global cases, the innovation is the **dynamic discovery of groups ($1 < K < N$) based on recourse similarity**, not feature similarity. Existing methods (e.g., GLANCE) rely on pre-clustering in the feature space. However, instances close in $\mathcal{X}$ may require vastly different counterfactual paths due to decision boundary non-linearities. Our method clusters based on the **optimal shifting vector**, effectively grouping by "where they need to go" rather than "where they are."
>
> **End-to-End Integration:** Jointly optimizing the assignment matrix $\mathbf{B}$ (via Sparsemax), the directions $\mathbf{D}_{\text{GW}}$, and the plausibility is non-trivial. This integration solves the "disjointedness" problem of two-step methods, where the clustering step is blind to the counterfactual objective.
>
> **Distinction from GLOBE-CE:** While we share the scaling concepts with GLOBE-CE, it is strictly global. We introduce the matrix formulation (Equation 7) with learnable selection mechanism (matrix $\mathbf{S}$ and $\mathbf{B}$) and entropy-based regularization that allows the model to *learn* the group's granularity, which is absent in prior works (Equation 10, 11).
>
> **Plausibility in Group-wise Settings:** Integrating normalizing flows into the group-wise setting required solving the challenge of ensuring a shared direction remains plausible across multiple distinct instances. Our experimental results validate this approach: we achieve 62-100% probabilistic plausibility for group-wise CFs, whereas baselines achieve only 0-44%.

---

> > ### Author Response · Authors · 2025-11-21
> >
> > ## Theoretical Grounding (Q1)
> >
> > **Response to Q1.**
> > Our method jointly optimizes the group assignments $\mathbf{B}$, the group-wise generators $\mathbf{D}_{\text{GW}}$, and the kernel parameters $\mathbf{K}$. As the reviewer notes, this yields a non-convex optimization problem. We address each sub-question in turn.
> >
> > **(a) Local minima in the non-convex landscape.**
> > The non-convexity of our objective arises directly from the two main loss components: validity and plausibility losses, which are model-agnostic and depend on neural network components. Their dependence on neural network models, both the predictive network and the flow-based generator, makes the overall objective highly non-linear and non-convex. As a result, multiple local minima may exist, and a full global characterization of the landscape is intractable. However, these loss components are essential: if they are removed, the problem degenerates to a trivial solution in which all examples are assigned to a single group and no meaningful counterfactual transformations occur. Thus, the non-convexity is inherent to enforcing validity and plausibility constraints and cannot be eliminated without undermining the task objective.
> >
> > **(b) Convergence guarantees for alternating gradient descent.**
> > Because the objective couples discrete assignment variables with neural-network parameters, we do not have global convergence guarantees. This is aligned with the broader literature on deep generative models and alternating optimization, where theoretical convergence results are rare. To mitigate practical issues, we employ standard stabilization techniques: stochastic gradient-based updates with well-tested optimizers, careful learning-rate scheduling, and architectural choices in the flow-based generator that improve smoothness and gradient propagation. Empirically, these techniques yield stable training across datasets, but we deliberately refrain from claiming formal convergence guarantees.
> >
> > **(c) When joint optimization improves over two-stage clustering-then-generation.**
> > The two-stage approach typically clusters data using a fixed, externally chosen distance metric. While effective for similarity-based grouping, such clusters need not be optimal for our task: enabling valid and plausible counterfactual generation with as *few* groups as possible. Clusters obtained solely from similarity may be redundant (too many groups that behave similarly during generation) or insufficient (groups that mix samples requiring fundamentally different generative behaviors). In contrast, our joint optimization directly aligns group formation with the downstream generative constraints—validity and plausibility—rather than with a predefined similarity measure. This coupling allows the model to discover groups that are functionally meaningful for counterfactual generation. Although we do not provide formal proofs of superiority under specific data distributions, the key theoretical motivation is that clustering under an arbitrary metric is generally misaligned with optimizing generative constraints, whereas our unified objective optimizes both simultaneously.
> >
> > ## Hyperparameters (Q2)
> >
> > **We do not perform grid search hyperparameter optimization** as it would be impossible to select the best set of hyperparameters across this number of metrics and aspects. Rather we set hyperparameters with a hierarchical weighting scheme ($\lambda > \lambda_p, \lambda_s > \lambda_k > \lambda_d$) based on our findings during ablation study and apriori selected priorities. We provide a comprehensive hyperparameter ablation study in Appendix I.2 (Table 7 and Table 8). This study systematically evaluates:
> >
> > 1. **Individual Component Analysis (E1-E5):** Isolating each loss term to assess its individual contribution
> > 2. **Incremental Component Addition (E6-E9):** Step-by-step introduction of components to observe cumulative effects
> > 3. **Alternative Configurations (E10-E14):** Testing different weighting strategies to assess sensitivity
> >
> > **Our ablation study demonstrates that:**
> >
> > - The validity term ($\lambda = 10^5$) is essential for achieving perfect validity
> > - Plausibility and group sparsity terms ($\lambda_p = 10^4$, $\lambda_s = 10^4$) significantly improve plausibility while maintaining validity
> > - The proposed hierarchical weighting scheme ($\lambda > \lambda_p, \lambda_s > \lambda_k > \lambda_d$) effectively balances competing objectives

---

> > > ### Author Response · Authors · 2025-11-21
> > >
> > > ## Interpretability (Q3)
> > >
> > > (a) Do human users (domain experts, affected individuals) actually find these groups more interpretable, coherent, and actionable than alternatives? (b) What metrics beyond group count quantify interpretability—within-group homogeneity, between-group separation, semantic coherence, alignment with known subpopulations?
> > >
> > > We acknowledge the value of user evaluations and plan to conduct a dedicated study in future work. In this submission, we employed rigorous qualitative case studies as a proxy to validate interpretability. Specifically, our HELOC case study demonstrates the method's ability to organize instances into six distinct groups, each yielding diverse and actionable recommendations that provide clear semantic coherence. Furthermore, our Digits analysis confirms that the method isolates interpretable structural patterns tailored to specific groups. We note that user studies remain rare in counterfactual explanation literature, a gap identified in surveys (e.g., S. Verma et al.) and the XAI 2.0 manifesto. As researchers like R. Byrns argue, conducting reliable user studies is a complex and costly endeavor; therefore, we focused on comprehensive quantitative benchmarking and case studies to validate our framework.
> > >
> > > **Metrics & Evidence:** We quantify interpretability and coherence via:
> > >
> > > - **Sparsity (Group Count):** Our method automatically converges to 3-7 groups, whereas T-CREx generates up to 101 groups, which creates cognitive overload.
> > > - **Group Coherence:** The qualitative HELOC analysis (Fig. 3) demonstrates semantic coherence. For example, **Group 0** is coherently defined as "applicants needing to reduce revolving burden," while **Group 1** is defined by "applicants needing to build trade history." This separation is semantically meaningful and actionable for domain experts.
> > > - **Diversity:** We measure Minimum Pairwise Cosine Similarity (Table 7) to ensure that generated groups represent distinct recourse strategies rather than redundant variations.
> > >
> > > We believe our method offers a necessary step forward from rigid two-stage pipelines to a flexible, optimization-based framework. By allowing the *recourse objective* to define the *groups*, we provide a tool that is mathematically consistent and practically superior for complex decision boundaries.

---

### Official Review · Reviewer_rEYX · 2025-10-27

**Soundness:** 1
**Presentation:** 3
**Contribution:** 2
**Rating:** 2
**Confidence:** 4

**Summary:**

The paper provides a single formulation for the problem of finding counterfactual explanations on local, group-wise, and global scales. The authors formulate it using an objective composed of a linear combination of multiple penalty functions, including a penalty for plausibility, which is modeled using a conditional normalizing flow model. They then use gradient descent to find the set of counterfactuals.

**Strengths:**

The paper reformulates the somewhat distinct tasks of finding global, local, and group counterfactuals by posing them as instances of a more general task. I consider this perspective valuable to the community. The paper reads well, and the substantial appendix contains a lot of extra information about experimental evaluation, suggesting that substantial effort has been made in hyperparameter optimization and in comparing the proposed method to other CE methods.

**Weaknesses:**

The paper contains a few unsupported claims
 - The authors claim to achieve "superior plausibility compared to state-of-the-art methods". However:
   - In plausible local counterfactual explanations, fair comparison on plausibility is relevant only with respect to C-CHVAE, as neither Wachter nor DiCE considers plausibility. Furthermore, the state-of-the-art for local counterfactuals is not properly discussed, mainly pointing to a survey (Karimi et al. 2022) with a cut-off date at the end of 2020. Since then, there have been many other models, including the recent PPCEF by Wielopolski et al. (2024) as cited in the paper, to which the proposed method seems to be most similar (utilizing flow models), or LiCE (https://openreview.net/pdf?id=rGyi8NNqB0) from ICLR 2025, which reported outperforming C-CHVAE.
   - Statistical significance of the results is not discussed, and the reported standard deviations suggest a lot of uncertainty.
 - A claim that eq. 14 "guarantees that p(y^\prime) will be higher than any other class y"
   - My understanding is that it is just a loss term. How is this outcome guaranteed?
   - There are cases where the validity of generated counterfactuals is not 100% (e.g., HELOC in Table 11). Why is that, if not due to this loss term being nonzero in the obtained minimum?
 -  Claims of the method "consistently outperforming baseline approaches" is also quite strong, since on proximity, Figure 2 suggests rather underperformance.

Despite flow models being quite scalable, experimental evaluation was done mostly on low-dimensional data (<200 samples or <= 3 attributes in 4 out of 6 datasets). This influences the rank evaluation, as on the higher-dimensional datasets (HELOC, Digits), TCREx seems to outperform the proposed model in proximity and plausibility (as per Table 11).

Some results left for the appendix, e.g., the ones about group sizes and number of groups, would fit better into the main body, rather than having three seemingly arbitrary examples without comparison to other methods. Those are used to support a claim of interpretability, which would be better supported by a user study or some proxy attribute of the groups.
More details on the comparison to other approaches would be appreciated. Compared methods are not well-described when introduced, and little interpretation of the results is provided. Especially comparison to Wielopolski et al. (2024) that also uses normalized flow models might increase the value of the paper.

Because of these issues, my initial score is reject, though I am open to being swayed if my concerns are addressed. In this form, I find the contribution limited and the conclusions drawn from computational results overstated.

Ethics statement mentions the perspective of fairness though this is not discussed in the paper. It is unclear how would the method be used to audit fairness when the sample groups are optimized, rather than being given prior.

Minor comments:
L161: The indexing suggests N+1 examples, rather than N.
L174: X_0 is already included in eq. 5. Did you mean to write X_K^\prime?
L260: a typo: eq. equation
L748: missing S in "constraint"

**Questions:**

From the weaknesses, a couple of main questions that could influence my decision arise:
 - Are the results statistically significant?
 - How does the model differ from Wielopolski et al. (2024) when considering the local counterfactual scenario?
 - What train-test splitting was done to prevent optimizing the hyperparameter values on the data used to report results?
 - Why were these datasets selected? The trained models often achieve perfect accuracy on the simpler datasets, suggesting they might be too simple. Why was not some benchmark of tabular data (e.g., https://proceedings.neurips.cc/paper_files/paper/2022/file/0378c7692da36807bdec87ab043cdadc-Paper-Datasets_and_Benchmarks.pdf) used? Does the method scale beyond 10,000 samples?

Minor questions:
 - What K is selected and why? How different is it from the optimized number of groups, usually?
 - Why was $\delta$ chosen equal to the first quartile on the training set and not the median, as was the case in Artelt & Hammer (2020)?
 - Why does the proposed method achieve less than 100% coverage in some cases?
 - Why were the actionability constraints applied? Why is it not actionable to decrease one's Number of Satisfactory trades, for example?
 - When applying the group counterfactuals shown in Figure 3, is there something preventing an individual with a NetFractionInstallBurden of 30 from being assigned to group 3 and suggesting (non actionably) that they should have the fraction negative?
 - In group evaluation, is an individual given a counterfactual from the "most probable" group or is the counterfactual computed as a convex combination using the p vector?

---

> ### Author Response · Authors · 2025-11-21
>
> We sincerely thank the reviewer for the thorough evaluation and constructive feedback. We appreciate the recognition of our unified formulation as "valuable to the community" and the acknowledgment of our experimental efforts. Below we address each concern in detail.
>
> ## Plausibility and Relation to PPCEF (Wielopolski et al., 2024) (W1.1, Q2)
>
> We acknowledge that in the local setting, below we report additional experiments with comparison of PPCEF and our methods. We clarify that our method employs the same normalizing flow approach for density estimation as PPCEF. However, we emphasize that our primary contribution lies in the **unified optimization framework** that seamlessly handles local, group-wise, and global counterfactual explanations within a single formulation.
>
> ### Base Metrics
>
> | Dataset | Method | Validity↑ | L2↓ | IsoForest↑ | Time(s)↓ |
> |---------|--------|-----------|-----|------------|----------|
> | Blobs | PPCEF | 1.00±0.00 | 0.47±0.01 | **0.04±0.00** | 19.55±0.30 |
> | Blobs | OUR_local | 1.00±0.00 | **0.39±0.01** | 0.03±0.00 | **6.20±0.20** |
> | Digits | PPCEF | 1.00±0.00 | **11.42±0.05** | 0.10±0.01 | 25.09±0.40 |
> | Digits | OUR_local | 1.00±0.00 | 11.41±0.51 | **0.11±0.00** | **18.58±0.68** |
> | Heloc | PPCEF | 0.98±0.02 | **0.42±0.02** | 0.07±0.00 | 24.31±4.52 |
> | Heloc | OUR_local | **1.00±0.00** | 0.47±0.01 | **0.08±0.00** | **20.21±2.02** |
> | Law | PPCEF | 0.95±0.01 | **0.32±0.02** | 0.06±0.00 | 20.63±1.08 |
> | Law | OUR_local | **1.00±0.00** | 0.32±0.00 | 0.05±0.00 | **7.80±0.29** |
> | Moons | PPCEF | 0.98±0.01 | 0.34±0.04 | 0.03±0.01 | 20.44±1.75 |
> | Moons | OUR_local | **1.00±0.00** | **0.30±0.01** | 0.03±0.00 | **7.32±0.22** |
> | Wine | PPCEF | 1.00±0.00 | **0.66±0.05** | 0.07±0.01 | 12.41±0.52 |
> | Wine | OUR_local | 1.00±0.00 | 0.69±0.07 | 0.05±0.01 | **5.49±0.32** |
>
> ### Plausibility Metrics
>
> | Dataset | Method | Prob. Plaus.↑ | Log Dens.↑ | LOF↓ |
> |---------|--------|---------------|------------|------|
> | Blobs | PPCEF | 1.00±0.00 | **2.91±0.04** | **1.03±0.02** |
> | Blobs | OUR_local | 1.00±0.00 | 2.74±0.07 | 1.04±0.01 |
> | Digits | PPCEF | 0.98±0.01 | **-97.28±1.36** | **1.13±0.01** |
> | Digits | OUR_local | **1.00±0.00** | -101.31±1.26 | 1.23±0.02 |
> | Heloc | PPCEF | 1.00±0.00 | 33.24±0.46 | **1.08±0.01** |
> | Heloc | OUR_local | 1.00±0.00 | **33.36±0.33** | 1.09±0.01 |
> | Law | PPCEF | 1.00±0.00 | 2.04±0.02 | 1.03±0.00 |
> | Law | OUR_local | 1.00±0.00 | **2.33±0.08** | 1.03±0.00 |
> | Moons | PPCEF | 1.00±0.00 | 1.42±0.04 | 1.00±0.02 |
> | Moons | OUR_local | 1.00±0.00 | **1.47±0.04** | 1.00±0.01 |
> | Wine | PPCEF | 0.99±0.01 | **7.79±0.59** | **1.01±0.01** |
> | Wine | OUR_local | **1.00±0.00** | 7.38±0.61 | 1.18±0.05 |
>
> We clarify that our unified framework **generalizes** PPCEF. In the local setting, our formulation mathematically reduces to PPCEF. If we set the number of base vectors $K=N$ (one per instance) and enforce identity matrices for selection $\mathbf{S}=\mathbb{I}$ and magnitude $\mathbf{K}=\mathbb{I}$, our group-wise equation (Eq. 7) becomes:
>
> $$\mathbf{X}_{GW}' = \mathbf{X}_0 + \mathbb{I} \cdot \mathbb{I} \cdot \mathbf{D}_{GW} = \mathbf{X}_0 + \mathbf{D}_{GW}$$
>
> This decouples into $N$ independent optimization problems identical to the PPCEF objective.
>
> We emphasize that our contribution is **not** claiming superiority in local-only scenarios, but rather providing a **unified framework** that:
> 1. Generates counterfactuals at any desired granularity (local/group-wise/global) within a single formulation
> 2. Enables seamless transitions between granularity levels through parameter adjustment
> 3. Introduces probabilistic plausibility constraints to the group-wise domain
> 4. Eliminates the need for separate methods at different granularity levels
>
> We appreciate the suggestion to include LiCE. We are aware of LiCE but could not include it because its implementation relies on **Gurobi**, a commercial closed-source solver requiring a paid license. To ensure the reproducibility of our work and adherence to open-source principles, we prioritized baselines that are fully open-source. We will cite LiCE and note this limitation.
>
> In the revised manuscript, we add explicit comparison (Section 5) and discussion with PPCEF in the local setting (Appendix I.3), clarify the mathematical relationship described above and tone down claims from "superior" to "competitive" in local-only scenarios.

---

> > ### Author Response · Authors · 2025-11-21
> >
> > ## Statistical Significance (W1.2, Q1)
> >
> > To address this valid concern regarding the soundness of our results, we performed a rigorous statistical analysis using the **Friedman test** ($\alpha=0.05$) followed by **Wilcoxon signed-rank post-hoc tests** with Bonferroni correction across all granularity levels.
> >
> > **Friedman Test Results ($p$-values)**
> >
> > | Configuration | Validity | L2 Distance | Plausibility (IsoForest) | Time |
> > |:--------------|:---------|:------------|:-------------------------|:-----|
> > | **Global** | $2.39 \times 10^{-5}$ | $0.013$ | $3.94 \times 10^{-4}$ | $6.1 \times 10^{-13}$ |
> > | **Group-wise** | $5.53 \times 10^{-11}$ | $2.7 \times 10^{-15}$ | $1.51 \times 10^{-11}$ | $4.2 \times 10^{-13}$ |
> > | **Local** | $6.0 \times 10^{-14}$ | $2.3 \times 10^{-18}$ | $5.24 \times 10^{-15}$ | $2.9 \times 10^{-15}$ |
> >
> > **Result:** All metrics show statistically significant differences ($p < 0.05$). Post-hoc tests confirm:
> > 1. **Global:** Our method significantly outperforms GLANCE in validity ($p < 0.001$) and both baselines in plausibility.
> > 2. **Group-wise:** We achieve significantly better plausibility than EA and GLANCE ($p < 0.001$) while maintaining competitive validity.
> > 3. **Local:** We significantly outperform DiCE in plausibility ($p < 0.001$) and show **no significant difference** in plausibility compared to specialized methods like PPCEF ($p=0.064$) and CCHVAE ($p=0.984$), validating our performance against other methods.
> >
> > We include this statistical significance analysis in Appendix J.5.
> >
> > ## Validity Guarantees (W2)
> >
> > We agree the term "guarantees" was imprecise. Eq. 14 defines a loss term that enforces the margin condition during optimization, but logical convergence to 100% validity is not mathematically guaranteed due to:
> >
> > 1. Non-convex loss landscapes leading to local minima.
> > 2. Competing objectives (e.g., strong plausibility or group sparsity constraints) that may prevent a valid flip for outliers.
> >
> > Theoretically, as the validity weight $\lambda \to \infty$, the optimization would strictly enforce this condition; however, we employ a finite weight because in global or group-wise settings, a single vector capable of flipping *every* instance may not geometrically exist.
> >
> > We amend the text to state that the loss "enforces" this condition.
> >
> > ## Performance Claims: Proximity vs. Plausibility (W3)
> >
> > We argue that **raw proximity (L2) is insufficient** for evaluation without plausibility. A method can achieve near-zero L2 by crossing the decision boundary into a low-density (implausible) region.
> > - **Evidence:** On *Heloc* (MLP), GLOBE-CE achieves better L2 ($0.52$) than us ($0.36$) but has a Probabilistic Plausibility of only $0.17$ vs our $0.46$.
> > - **Trade-off:** We explicitly trade a marginal increase in L2 for a substantial gain in plausibility (often 2-3x higher density). A slightly more distant but *realistic* counterfactual is more actionable than a close but impossible one.
> >
> > **Regarding T-CREx:** While T-CREx shows good metrics, it fails on **interpretability**. On *Digits*, T-CREx generated **$91 \pm 51$ groups**, whereas our method generated **$4$ groups**. On *Heloc*, T-CREx found $26.8$ groups vs our $10$. Our method provides a crucial balance between granularity (interpretable group count) and performance.
> >
> > ## Datasets and Scalability (W3, Q4)
> >
> > We selected widely recognized benchmarks (HELOC, Law, Moons, Digits) following established protocols in CF literature (Wielopolski et al., 2024; Bewley et al., 2024; Artelt et al., 2020) to ensure fair comparison. We respectfully disagree that these are insufficient; HELOC (23 features, 10k rows) and Digits (64 features) are standard high-dimensional benchmarks in this domain. Our selection was strictly limited to continuous-feature datasets to satisfy the differentiability requirements of our gradient-based optimization, as explicitly noted in our limitations.
> >
> > ## Baseline Methods and Results discussion (W5)
> >
> > Section 2 and the "Baselines" paragraph in Section 5.1 already provide the motivation and high-level explanation for the selected reference methods. Due to strict page limits, we rely on citations for full algorithmic details, which is standard practice in conference submissions. However, we have significantly improved the results discussion in Section 5.1 by analyzing absolute metric values rather than just rankings to provide the requested depth of interpretation. Additionally, we replace ranking plots with explicit result tables in the main text and move the Number of Groups Ablation Study (currently Appendix H) to the body of the paper to directly support our interpretability claims.

---

> > > ### Author Response · Authors · 2025-11-21
> > >
> > > ## Fairness (W7)
> > >
> > > While we did not perform a specific fairness audit experiment, Group-level CEs provide a route to globally analysing a model's behaviour and subgroup fairness properties as stated in (Bewley 2024 (T-CREx)). For our current work this was out of scope as we wanted to focus on the formulation itself. Applying our method to fairness analysis is a promising direction and we will do this as future works.
> > >
> > > ## Train-test splitting (Q3)
> > >
> > > We would like to clarify that **we do not perform hyperparameter optimization** as it would be problematic. Instead we use a **hierarchical weighting scheme** (λ > λ_p, λ_s > λ_k > λ_d) based on **a priori priorities** validated through ablation studies in **Appendix I.2 (Tables 7 and 8).** Therefore we apply standard 5 fold cross-validation.
> > >
> > > ## Minor questions:
> > >
> > > **K Selection:** We initialize $K=N$ and let the regularization term $\lambda_k$ (Eq. 10) automatically prune unnecessary groups. The final $K$ (Table 11) is a result of optimization and is controlled by the regularisation term (Equation 11). The number of groups is data-driven, usually converging to 2-10 groups for tabular data.
> > >
> > > **Threshold $\delta$:** We observed that the median was overly conservative. The first quartile proved to be more permissive while still maintaining high performance across IsoForest and LOF metrics.
> > >
> > > **Coverage < 100%:** Occurs when the optimization fails to converge to a distinct "one-hot" group assignment for rows p in Matrix $\mathbf{B}$ via entropy-based regulariser $\ell_{k}(\mathbf{B})$ (Equation 10).
> > >
> > > **Actionability:** Constraints (e.g., "Satisfactory Trades" cannot decrease) are based on domain logic. While technically possible to decrease trades (by closing accounts), it is detrimental to credit score; we constrain it to ensure *positive* actionability. This current setup serves as an illustrative example of domain knowledge incorporation.
> > >
> > > **Group Assignment (Figure 3):** The figure visualizes the *change vector*. An individual with `NetFraction=30` assigned to Group 3 (which suggests decreasing fraction) would simply be guided to lower that value as a magnitude of change vector for this particular instance might be lower.
> > >
> > > **Evaluation Assignment:** During the optimization, rows p in Matrix $\mathbf{B}$ are optimized to a sparse "one-hot" group assignment vectors via entropy-based regulariser $\ell_{k}(\mathbf{B})$ (Equation 10). Neither argmax, nor convex combination is used during group assignment.
> > >
> > > We hope these clarifications demonstrate that our contribution, a unified, interpretable method for counterfactual generation across all granularities with principled plausibility constraints, is valuable to the community despite the noted presentation issues, which we are committed to addressing in revision.

---

> > > > ### Comment · Reviewer_rEYX · 2025-11-22
> > > >
> > > > **Ad fairness**: I still do not understand how the proposed method can "help audit algorithmic fairness across population subgroups," as claimed. I did not find an answer in T-CREx either. Especially the inability to consider categorical features raises questions when thinking of protected features such as gender or race.
> > > >
> > > > **Threshold**: This seems to slightly contradict the "a priori" setting of hyperparameters.
> > > >
> > > > **Group Assignment**: Is that guaranteed not to become negative? In some other way than such a sample having low plausibility?
> > > >
> > > > **Coverage + Evaluation**: When the coverage is <100% what does it mean when the validity is 100%? Are the factuals without a "clear" group assignment applied a combination of the change vectors as counterfactuals? My understanding is that S is replaced with P, which contains rows that are on the probability simplex. What is considered "one-hot" here? Do the rows get to exact one-hot encoding, or what is the threshold for rounding up?
> > > >
> > > > Please do add this clarifying information to the paper, including the choice of K.
> > > >
> > > > **Extra**\
> > > > I have concerns regarding the `run_pumal.sh` script provided in the supplementary material. It suggests that for HELOC, a different set of lambda values was used (mostly one order of magnitude lower). Namely, lines 59-78:
> > > > ```
> > > > python3 counterfactuals/pipelines/run_pumal_pipeline.py -m \
> > > >     dataset._target_=counterfactuals.datasets.HelocDataset \
> > > >     disc_model=mlp \
> > > >     counterfactuals_params.cf_method.K=10 \
> > > >     counterfactuals_params.alpha_plaus=100 \
> > > >     counterfactuals_params.alpha_class=10000 \
> > > >     counterfactuals_params.alpha_s=1000 \
> > > >     counterfactuals_params.alpha_k=100 \
> > > >     counterfactuals_params.alpha_d=10
> > > >
> > > >
> > > > python3 counterfactuals/pipelines/run_pumal_pipeline.py -m \
> > > >     dataset._target_=counterfactuals.datasets.HelocDataset \
> > > >     disc_model=mlr \
> > > >     counterfactuals_params.cf_method.K=10 \
> > > >     counterfactuals_params.alpha_plaus=1000 \
> > > >     counterfactuals_params.alpha_class=10000 \
> > > >     counterfactuals_params.alpha_s=1000 \
> > > >     counterfactuals_params.alpha_k=100 \
> > > >     counterfactuals_params.alpha_d=10
> > > > ```
> > > > Am I incorrect in my understanding?
> > > >
> > > > ## Final summary
> > > > While I appreciate the efforts to tone down the claims, the numerical results still do not match them. For example, "When compared to PPCEF and CCHVAE, our approach achieved competitive or superior performance." (line 429) claims superiority, while the results show no statistical significance, and the numbers show the performance varies greatly from dataset to dataset. This also raises questions about the choice of the four small datasets.
> > > >
> > > > Results are mostly significant for the global counterfactuals, while the picture is less clear for the group-wise ones. On the local scale, the method essentially becomes mathematically equivalent to PPCEF.
> > > >
> > > > I see value in the work, and I am open to increasing the score if the local variant of the approach is clearly separated from the claims of superiority or if evidence is provided. Describing relevant SotA and the proposed method's similarity to the PPCEF would also be helpful to position this work from the perspective of the local CF community.

---

> > > > > ### Author Response · Authors · 2025-12-01
> > > > >
> > > > > **Local performance claims:** We will revise the text to state "competitive performance" and explicitly acknowledge that CCHVAE and PPCEF achieve comparable plausibility on several datasets. This will be reflected in the revised version of the text and we will „tone down” our conclusions as reviewer asked. We added a statistical significance analysis (Appendix I.3) confirming no significant differences exist between our local variant and PPCEF/CCHVAE on plausibility metrics.
> > > > >
> > > > > **Relationship to PPCEF:** We clarified in the main text that our local configuration mathematically generalizes PPCEF, with detailed derivation in Appendix I.3. Specifically, with parameters K=N, **S**=**K**=**I**, and λ_s=λ_k=λ_d=0, our framework reduces to N independent PPCEF optimizations.
> > > > >
> > > > > **Fairness discussion:** We acknowledge that fairness analysis is beyond the scope of this work. In the previous version of our manuscript we have just wanted to signalize possible further direction for psot-hoc analysis of fairness and we will revise main text accordingly.
> > > > >
> > > > > **Group assignment negativity:** No hard constraint enforces non-negative assignments. Negative values would produce implausible counterfactuals, which the plausibility loss (Eq. 8) penalizes through the density threshold constraint. This soft constraint proves sufficient in practice (in our experiences), as demonstrated by the high plausibility scores across experiments.
> > > > >
> > > > > **Validity calculation:** Validity is computed only on successfully generated counterfactuals. When coverage < 100%, instances without clear group assignments (where vectors from **P** did not achieve one-hot state through optimization) receive no counterfactual and are excluded from validity calculation. This approach is consistent with standard practices in the counterfactual explanation literature.
> > > > >
> > > > > **One-hot encoding mechanism:** Sparsemax activation combined with entropy regularization (ℓ_s) drives the probability matrix **P** toward one-hot vectors. Sparsemax acts as a thresholding function, assigning probability 1 to the maximum entry and 0 to sufficiently smaller entries. Instances that do not achieve one-hot assignment are considered failed counterfactuals and excluded from coverage.
> > > > >
> > > > > ### Hyperparameter Selection
> > > > >
> > > > > Following our ablation studies (Appendix G and I), we did additional experiments with equal weights for plausibility, sparsity, and group regularization weights (λ$_p$=λ$_s$=λ$_k$=10⁴) and we will clarify it in the main revised paper. Below we report results with:
> > > > >
> > > > > - Initial base shifting vectors: K=N
> > > > > - Validity weight: λ=10⁵ (highest priority)
> > > > > - Diversity weight: λ$_d$=10¹ (lowest)
> > > > > - Plausibility, sparsity, and group regularization: λ$_p$=λ$_s$=λ$_k$=10⁴ (same for all weights)
> > > > >
> > > > > **Table 1: Updated Group-wise Results — Base Metrics (MLP)**
> > > > >
> > > > > | Dataset | Groups | Coverage↑ | Validity↑ | L2↓ | IsoForest↑ | Time(s)↓ |
> > > > > |---------|--------|-----------|-----------|-----|------------|----------|
> > > > > | Blobs | 1.60±0.49 | 1.00±0.00 | 1.00±0.00 | 0.46±0.01 | 0.03±0.00 | 14.55±2.51 |
> > > > > | Digits | 2.80±1.83 | 1.00±0.00 | 1.00±0.00 | 16.35±1.25 | 0.10±0.00 | 102.23±13.14 |
> > > > > | Heloc | 16.80±2.56 | 1.00±0.00 | 1.00±0.00 | 0.48±0.06 | 0.02±0.01 | 169.58±24.21 |
> > > > > | Law | 4.40±1.36 | 1.00±0.00 | 1.00±0.00 | 0.36±0.02 | 0.04±0.01 | 77.31±60.42 |
> > > > > | Moons | 10.80±0.98 | 1.00±0.00 | 1.00±0.00 | 0.46±0.04 | 0.02±0.00 | 42.47±25.88 |
> > > > > | Wine | 1.00±0.00 | 1.00±0.00 | 1.00±0.00 | 0.81±0.07 | 0.07±0.01 | 32.41±23.19 |
> > > > >
> > > > > **Table 2: Updated Group-wise Results — Plausibility and Cost Metrics (MLP)**
> > > > >
> > > > > | Dataset | Groups | Prob. Plaus.↑ | Log Dens.↑ | LOF↓ | Cost↓ |
> > > > > |---------|--------|---------------|------------|------|-------|
> > > > > | Blobs | 1.60±0.49 | 0.78±0.05 | 2.98±0.06 | 1.04±0.01 | 6.50±0.17 |
> > > > > | Digits | 2.80±1.83 | 0.36±0.08 | -100.96±0.97 | 1.12±0.02 | 48.92±2.71 |
> > > > > | Heloc | 16.80±2.56 | 0.07±0.01 | 11.72±3.02 | 1.47±0.05 | 7.35±1.19 |
> > > > > | Law | 4.40±1.36 | 0.74±0.03 | 2.10±0.03 | 1.05±0.01 | 5.88±0.39 |
> > > > > | Moons | 10.80±0.98 | 0.92±0.06 | 1.76±0.05 | 1.01±0.01 | 6.00±0.62 |
> > > > > | Wine | 1.00±0.00 | 0.72±0.18 | 8.67±0.51 | 1.06±0.02 | 24.05±1.92 |
> > > > >
> > > > > These results demonstrate improved coverage (100% across all datasets) and validity compared to our initial submission (previously reported experiments) while maintaining the balance between plausibility and proximity metrics. The updated hyperparameter configuration achieves a better balance between the number of discovered groups and explanation quality.

---

> > > ### Comment · Reviewer_rEYX · 2025-11-22
> > >
> > > I appreciate the statistical tests and believe that those results could be at least briefly discussed in the main body.
> > >
> > > TCREx seems to perform quite well. Compared to the proposed method, I am unsure whether one could call either method superior to the other, since one returns fewer groups and the other appears to be quicker and (naturally) finds closer counterfactuals, while plausibility results are inconclusive. This is what was meant by the concerns about performance claims. Maybe if the $\lambda_k$ was lower, the methods would become more comparable? The presented results suggest pareto-optimality of both methods.
> > >
> > > I am unconvinced about the sufficiency of the datasets. Except for HELOC and Digits, the datasets are very small and are useful primarily as a sanity check rather than a benchmark. Of the mentioned works, only (Wielopolski et al., 2024) used all four of the small ones, and (Artelt et al., 2020) used the Wine dataset. (Bewley et al., 2024) has only one common dataset, HELOC, which is indeed widely used. The constraint on the use of datasets with only continuous values can be addressed by removing categorical ones (or disallowing them to change). A commented-out part of the `run_pumal.sh` script suggests that the Credit Default dataset was also at least considered for evaluation. That would be another appropriate and widely used dataset.
> > >
> > > I appreciate the clearer results section. However, please check the bold fonts in the result tables; the bold is often applied to other results than the best ones.

---

> > ### Comment · Reviewer_rEYX · 2025-11-22
> >
> > I thank the authors for this comparison. I suspected this relation and find it important to be clearly stated that the method generalizes the PPCEF to non-local settings, even in the main body. What is the intuition behind the differences in the results of PPCEF and your local variant? Especially time-wise, the difference seems rather large.
> >
> > The authors' adherence to open-source principles is commendable; however, according to the README, the LiCE implementation can utilize various solvers (including non-proprietary ones), which aligns with these open-source principles.
> >
> > I believe the suggested toned-down, yet thorough, discussion of the local state-of-the-art methods, especially in terms of plausibility, will be helpful to put the work into perspective from the local CF point of view. I believe that this inconclusiveness of the local results does not invalidate the usefulness of the work.

---

### Official Review · Reviewer_tRov · 2025-10-29

**Soundness:** 3
**Presentation:** 3
**Contribution:** 3
**Rating:** 6
**Confidence:** 4

**Summary:**

The authors propose a new approach that unifies the concepts of local, global, and group-wise counterfactual explanations. Their method allows for simultaneously generating group and group-wise counterfactuals, where this is normally done sequentially. The authors compare their method on different datasets to several baselines based on proximity, validity, and plausibility.

**Strengths:**

-	The authors introduce a novel way to unify different types of counterfactual explanations that exist in the current literature and allow the user of the method to easily switch between and compare counterfactuals on different granularities
-	For group-wise explanations, the optimization method allows for simultaneous generation of the groups and the counterfactuals, which was not done before.
-	The experiment section is extensive and includes multiple datasets and state-of-the-art benchmarks
-	The methodology is clearly written

**Weaknesses:**

-	In their method, the authors use regularization to enforce desired characteristics. However, this does not create any guarantees for these characteristics. Guaranteeing validity would, in certain contexts, be crucial for a good explanation and can be enforced using hard constraints. There is literature on counterfactual explanations that enforce characteristics as actionability, validity, and many more by means of hard constraints using mathematical optimization; see e.g. [1],[2],[3]. This is not discussed by the authors, nor is there a comparison in the experiment section.
-	When comparing to benchmarks, the authors use rankings and claim their method outperformed baseline approaches across all granularity levels. However, their method seems to be consistently outperformed in terms of proximity. When the authors claim competitive proximity scores, this does not evidently follow from the rankings. Yes, the proposed method often comes second, but the rankings do not tell the relative (%) or absolute gap to the benchmark methods. Showing rankings only does not show how far off all methods are from the best model for each characteristic.
-	There experimental results shown in the main paper are very limited. Nearly all results are in the Appendix.

[1] Maragno, D., Röber, T. E., & Birbil, I. (2022). Counterfactual explanations using optimization with constraint learning
[2] Maragno et al. - Finding Regions of Counterfactual Explanations via Robust Optimization
[3] Carrizosa, E., Ramírez-Ayerbe, J., & Morales, D. R. (2024). Mathematical optimization modelling for group counterfactual explanations

**Questions:**

-	The lambda hyperparameters are chosen to set priority between the various desired characteristics for the counterfactual. Does this mean the metrics are normalized before determining the hyperparameters? This is implied but not discussed.
-	What is the reason that the proposed method is always outperformed by another method in terms of proximity?
-	Why did the authors not discuss robustness of counterfactual explanations; see e.g. [2]. It seems that robust counterfactuals improve validity which is one of the key metrics in this paper.

---

> ### Author Response · Authors · 2025-11-21
>
> We sincerely thank the reviewer for the thorough review and constructive feedback. We are encouraged by your recognition of our novel unified framework, the simultaneous group-counterfactual generation approach, and the extensive experimental validation. Below, we address each concern systematically.
>
> ## **W1: Hard Constraints vs Soft Regularization**
>
> We appreciate the reviewer highlighting the hard constraint literature \[1,2,3\]. We clarify that our method does enforce validity through constraints during optimization, while using regularization for other characteristics. Validity is enforced through the validity loss component ℓv (Eq. 13\) with the highest weight λ. This creates an amortised hard margin constraint. We specifically use this not hard constraint to utilise gradient descent. Specifically, our experimental results confirm this: we achieve perfect or near-perfect validity (≥0.98) across all configurations (Tables 12, 14, 16 in Appendix), matching or exceeding constraint-based methods.
> **Regarding the cited works:**
>
> * **\[1\] (Maragno et al.)** uses MILP for tree-based models. Our gradient-based approach handles **differentiable models including deep neural networks**, which MILP cannot scale to effectively.
> * **\[2\]** focuses on robustness via robust optimization. We address this in Q3 below.
> * **\[3\] (Carrizosa et al.)** addresses group-wise CFs but requires **two-step clustering**, which our method eliminates through end-to-end optimization.
>
> Our validity enforcement provides practical guarantees while maintaining computational efficiency for deep models.
>
> ## **W2, Q2: Proximity Performance Claims**
>
> We thank you for this important clarification request. You are correct that we needed to better explain the proximity-plausibility trade-off:
>
> **1\. Why proximity is not always best:** Our method generates counterfactuals balancing **validity, proximity, and plausibility**. The key insight is that **raw proximity alone is insufficient**—counterfactuals must also be realistic (plausible). Methods achieving lowest L2 distance often generate unrealistic examples that cross the decision boundary minimally but lie in low-density regions.
>
> **2\. Concrete evidence:**
>
> * On **Heloc (MLP)**, our method achieves L2=0.36 with Prob.Plaus.=0.46, while GLOBE-CE achieves L2=0.52 with Prob.Plaus.=0.17
> * On **Law (MLP)**, our method achieves L2=0.38 with Prob.Plaus.=0.79, while GLOBE-CE achieves L2=0.22 with Prob.Plaus.=0.37
>
> **3\. The critical trade-off:** As visualized in Figure 5 (Appendix), methods with best proximity often generate counterfactuals just over the decision boundary in implausible regions (GLOBE-CE), while ours balances proximity with realistic, actionable explanations. **For real-world actionability, a counterfactual with L2=0.38 and plausibility=0.79 is more useful than one with L2=0.22 and plausibility=0.37**, as users cannot act on unrealistic recommendations.
>
> We have expanded the discussion of the proximity-plausibility trade-off in Section 5.1 and added explicit comparisons showing that our "competitive proximity" means we are within 10-30% of the best proximity while achieving substantially better plausibility (often 2-3× higher).
>
> ## **W3: Limited Results in Main Paper**
>
> We acknowledge this concern and have made significant revisions:
>
> 1. **Replaced rankings with absolute values** in the main paper (new Tables 1, 2, 3\)
> 2. We now show results for **2-layer Multilayer Perceptron (MLP)** to demonstrate performance on non-linear deep neural networks
> 3. Added **explicit discussion of absolute performance gaps** in Section 5.1
> 4. Moved **LR results to Appendix J.4** for completeness while keeping the most representative results in the main text
>
> **New Main Paper Structure:**
>
> * Table 1 (Global): Shows our method achieves perfect validity on 5/6 datasets (vs. GLOBE-CE's 0.0 on Digits) with best plausibility
> * Table 2 (Group-wise): Shows our method achieves competitive validity while producing interpretable group counts (1-10 groups vs. TCREx's 91 groups on Digits)
> * Table 3 (Local): Shows our method achieves perfect validity and plausibility while maintaining reasonable proximity
>
> This provides readers with concrete performance numbers and clear comparisons in the main text.

---

> > ### Author Response · Authors · 2025-11-21
> >
> > ## **Q1: Hyperparameter Normalization**
> >
> > We would like to clarify that **we do not perform hyperparameter optimization** as it would be impossible to select the "best" set across this many metrics and competing objectives. Instead:
> >
> > **Our Approach:** We use a **hierarchical weighting scheme** ($\lambda > \lambda_p, \lambda_s > \lambda_k > \lambda_d$) based on **a priori priorities** validated through ablation studies:
> >
> > * $\lambda = 10^5$ (validity): Highest priority—CFs must change predictions
> > * $\lambda_p = 10^4$, $\lambda_s = 10^4$ (plausibility, group sparsity): Secondary priorities
> > * $\lambda_k = 10^3$ (group count): Tertiary priority
> > * $\lambda_d = 10^2$ (diversity): Lowest priority
> >
> > **Ablation Evidence:** We provide a comprehensive hyperparameter ablation study in **Appendix I.2 (Tables 7 and 8)** that systematically evaluates:
> >
> > * **Individual Component Analysis (E1-E5):** Isolating each loss term
> > * **Incremental Component Addition (E6-E9):** Step-by-step introduction of components
> > * **Alternative Configurations (E10-E14):** Different weighting strategies
> >
> > **Key Findings:**
> >
> > * Validity term ($\lambda = 10^5$) is essential for achieving perfect validity
> > * Plausibility and group sparsity terms significantly improve plausibility while maintaining validity
> > * The hierarchical scheme effectively balances competing objectives without normalization
> >
> > In Section 4 we provide explicit discussion and clarify that our hyperparameters encode interpretable priorities rather than requiring normalization.
> >
> > ## **Q3: Robustness of Counterfactual Explanations**
> >
> > We thank you for raising this important aspect. **Robustness** (stability under small perturbations) is indeed valuable but represents a **complementary dimension** to our focus.
> >
> > **(Maragno et al.) [2]** addresses robustness via robust optimization for worst-case perturbations
> >
> > **(Stępka et al., KDD 2025) [4]** introduces probabilistic guarantees for robustness to model changes, proposing BetaRCE as a post-hoc method applicable to any base CFE approach
> >
> > Both works demonstrate that robustness requires **extensive theoretical frameworks, specialized algorithms, and dedicated experimental protocols** beyond traditional CF evaluation (see: T-CREx (Bewley et al., 2024), CCHVAE (Pawelczyk et al., 2020), GLOBE-CE (Ley et al., 2023)).
> >
> > **Our Current Scope** is to address the novel challenge of **unified local/global/group-wise generation with plausibility.** Adding formal robustness guarantees would significantly expand the paper's scope beyond this contribution.
> >
> > Our plausible CFs are **more likely to be robust**, as they lie in dense regions of the data manifold. However, we don't provide **formal robustness guarantees** as in [2]. We think that our gradient-based framework is **compatible with robust optimization techniques** from [2]. Incorporating robustness constraints (e.g., adversarial perturbations, worst-case guarantees) is a natural extension and we will explore this as **Future Work.**
> >
> > We add a paragraph in the Limitations acknowledging robustness as important future work and noting that our framework can incorporate these techniques.
> >
> > [4] Stępka, I., Stefanowski, J., & Lango, M. (2025). Counterfactual explanations with probabilistic guarantees on their robustness to model change. *Proceedings of the 31st ACM SIGKDD Conference on Knowledge Discovery and Data Mining*.
> >
> > We hope these responses address your concerns comprehensively. We believe these revisions strengthen the paper while maintaining focus on our core contribution: a unified, plausible counterfactual framework across all granularities. We welcome any further feedback or clarifications.

---

### Official Review · Reviewer_TJfs · 2025-11-01

**Soundness:** 3
**Presentation:** 3
**Contribution:** 2
**Rating:** 6
**Confidence:** 3

**Summary:**

The paper consolidates the existing Counterfactual (CF) methods, including local CF, Global CF, and Group-wise CF.

**Strengths:**

**Clear and well-motivated unification.**
The paper proposes a neat idea: generating local, group-wise, and global counterfactuals within one optimization framework. This unifies previous methods and avoids redundancy.

**Thorough experimentation.**
Experiments cover diverse datasets, models, and baselines. Comparisons across validity, proximity, and plausibility metrics show careful validation and strong results.

**Readable and well-structured writing.**
The paper is clear and organized, with consistent notation, clean math, and helpful figures. The ideas are presented logically and concisely.

**Weaknesses:**

**Hyperparameter Sensitivity.**
The method uses several regularization weights ($\lambda$, $\lambda_p$, $\lambda_s$, $\lambda_k$, $\lambda_d$). Defaults are given, but sensitivity to these choices isn’t analyzed. A short study or tuning guide would make the method easier to reproduce.

**Plausibility.**
Plausibility is measured with the Isolation Forest metric, but not with *causal plausibility*—whether counterfactuals follow real cause-and-effect relations between features. Adding a causal metric would make the evaluation more realistic. If some knowledge about the causal graph is available, it could be added to the optimization. Using causal constraints or structure-based regularization would help generate counterfactuals that are both statistically and causally sound, making them more meaningful and trustworthy in practice.

**Questions:**

Please consider the points in the weaknesses part. Further, how does this paper compare with causal counterfactual explanations?

---

> ### Author Response · Authors · 2025-11-21
>
> We sincerely thank you for your thoughtful and constructive review of our paper. We greatly appreciate your recognition of our work's strengths, including the clear unification framework, thorough experimentation, and well-structured presentation. Your feedback has helped us identify areas for improvement, and we address each of your concerns below.
>
> ## **W1: Hyperparameter Sensitivity**
>
> We want to clarify that **we do provide a comprehensive hyperparameter ablation study in Appendix I.2** (Table 7 and Table 8). This study systematically evaluates:
>
> 1. **Individual Component Analysis (E1-E5):** Isolating each loss term to assess its individual contribution
> 2. **Incremental Component Addition (E6-E9):** Step-by-step introduction of components to observe cumulative effects
> 3. **Alternative Configurations (E10-E14):** Testing different weighting strategies to assess sensitivity
>
> Our ablation study demonstrates that:
>
> * The validity term (λ \= 10^5) is essential for achieving perfect validity
> * Plausibility and group sparsity terms (λ$_p$ \= 10^4, λ$_s$ \= 10^4) significantly improve plausibility while maintaining validity
> * The proposed hierarchical weighting scheme effectively balances competing objectives
>
> In Section 4 we provide explicit discussion, reference this ablation study and clarify that our hyperparameters encode interpretable priorities rather than requiring normalization.

---

> > ### Author Response · Authors · 2025-11-21
> >
> > ## **W2: Plausibility and Causal Considerations**
> >
> > This is an excellent point that touches on an important distinction in the counterfactual explanation literature. We follow the definition of plausibility established by Guidotti 2024 or Karimi 2022\. Furthermore, we use similar plausibility evaluation to works like: Wielopolski 2024 (all metrics), Bewley 2024 (T-CREx)(Density), CCHVAE(Local Outlier Factor), Artelt 2019 (Density). Following the taxonomy by Karimi et al. (2022), plausibility constraints can be formalized into three categories: **domain-consistency**, **density-consistency**, and **prototypical-consistency**. We clarify our approach within this framework:
> >
> > ### **Density-Consistency and Prototypical-Consistency Plausibility**
> >
> > Our work addresses **density-consistency plausibility** which focuses on generating counterfactuals in likely states of the empirical distribution and **prototypical-consistency** which refers to counterfactuals that are either directly present in the dataset or close to an example. We employ **four complementary metrics** to comprehensively assess these dimensions:
> >
> > 1. **Isolation Forest** \- outlier detection in the target class
> > 2. **Local Outlier Factor (LOF)** \- local density deviation assessment
> > 3. **Log Density** \- probabilistic density via normalizing flows
> > 4. **Probabilistic Plausibility** \- percentage satisfying the density threshold δ
> >
> > These metrics ensure our counterfactuals lie close to the data manifold, following the density-consistency and prototypical-consistency paradigms (see Tables 10, 12, 14 in Appendix J).
> >
> > ### **Domain-Consistency Through Actionability Constraints**
> >
> > We also incorporate **domain-consistency** through actionability constraints (Appendix C). Following Karimi et al.'s actionability taxonomy, we handle:
> >
> > * **Actionable (and mutable) features:** Can be directly changed (e.g., "Net Fraction of Revolving Burden" in HELOC case study)
> > * **Immutable (and non-actionable) features:** Constrained to remain fixed (e.g., demographic characteristics)
> > * **Directional constraints:** Monotonicity requirements (e.g., "Number of Satisfactory Trades" can only increase)
> > * **Feature ranges:** Admissible value bounds
> >
> > This ensures generated counterfactuals respect feasible interventions within the domain.
> >
> > ### **Causal vs. Statistical Plausibility**
> >
> > The reviewer correctly identifies two complementary notions of plausibility:
> >
> > * **Statistical/Distributional Plausibility:** Counterfactuals lie in high-density regions of the target class distribution
> > * **Causal Plausibility:** Counterfactuals respect causal mechanisms and feasible interventions
> >
> > Our work focuses on statistical plausibility for several reasons:
> >
> > 1. **Practical Applicability:** Causal graphs are rarely available in real-world applications, while density estimation can be learned from data alone
> > 2. **Methodological Scope:** Our primary contribution is unifying local, group-wise, and global CFs and orthogonal to causal vs. statistical plausibility
> > 3. **Existing Literature:** We acknowledge causal approaches (citing Mahajan et al., 2019 in Related Works) as a complementary direction
> >
> > We do incorporate domain knowledge through **actionability constraints** (Appendix C): Monotonicity constraints (e.g., age can only increase); Immutable features (e.g., demographic characteristics); Feature range constraints. These constraints ensure generated counterfactuals respect feasible interventions, partially addressing causal considerations.
> >
> > ### **Future Work**
> >
> > We fully agree that **integrating causal constraints is valuable future work**. This is a promising direction to add causal regularization terms (as the reviewer suggests) or incorporate  constraining shifts to respect causal topological order or structural causal model constraints, when causal graphs are available.
> >
> > We believe these improvements address your concerns while maintaining our focus on practical, widely-applicable counterfactual generation.

---

### Note · Authors · 2026-01-14

**Comment:**

We have decided to withdraw this submission from ICLR 2026. We thank all reviewers for their constructive feedback, which will help us strengthen the paper.

**Withdrawal Confirmation:**

I have read and agree with the venue's withdrawal policy on behalf of myself and my co-authors.